# A review of marine geomorphometry, the quantitative study of the seafloor

Vincent Lecours[1], Margaret F. J. Dolan[2], Aaron Micallef[3], Vanessa L. Lucieer[4]

[1] Marine Geomatics Research Lab, Department of Geography, Memorial University of Newfoundland, St. John's, A1B 3X9, Canada

[2] Geological Survey of Norway, P.O. Box 6315 Sluppen, 7491Trondheim, Norway

[3] Marine Geology and Seafloor Surveying, Department of Geosciences, University of Malta, Msida, MSD 2080, Malta

[4] Institute for Marine and Antarctic Studies, University of Tasmania, Hobart, 7004, Australia

*Correspondence to*: V. Lecours (vlecours@mun.ca)

**Abstract.** Geomorphometry, the science of quantitative terrain characterization, has traditionally focused on the investigation of terrestrial landscapes. However, the dramatic increase in the availability of digital bathymetric data and the increasing ease by which geomorphometry can be investigated using Geographic Information Systems (GIS) and spatial analysis software has prompted interest in employing geomorphometric techniques to investigate the marine environment.

Over the last decade or so, a multitude of geomorphometric techniques (e.g. terrain attributes, feature extraction, automated classification) have been applied to characterize seabed terrain from the coastal zone to the deep sea. Geomorphometric techniques are however not as varied, nor as extensively applied, in marine as they are in terrestrial environments. This is at least partly due to difficulties associated with capturing, classifying, and validating terrain characteristics underwater. There is, nevertheless, much common ground between terrestrial and marine geomorphometry applications and it is important that, in developing marine geomorphometry, we learn from experiences in terrestrial studies. However, not all terrestrial solutions

can be adopted by marine geomorphometric studies since the dynamic, four-dimensional nature of the marine environment causes its own issues throughout the geomorphometry workflow. For instance, issues with underwater positioning, variations in sound velocity in the water column affecting acoustic-based mapping, and our inability to directly observe and measure depth and morphological features on the seafloor are all issues specific to the application of geomorphometry in the marine

environment. Such issues fuel the need for a dedicated scientific effort in marine geomorphometry.

This review aims to highlight the relatively recent growth of marine geomorphometry as a distinct discipline, and offers the first comprehensive overview of marine geomorphometry to date. We address all the five main steps of geomorphometry, from data collection to the application of terrain attributes and features. We focus on how these steps are relevant to marine geomorphometry and also highlight differences and similarities from terrestrial geomorphometry. We conclude with

recommendations and reflections on the future of marine geomorphometry. To ensure that geomorphometry is used and developed to its full potential, there is a need to increase awareness of (1) marine geomorphometry amongst scientists

already engaged in terrestrial geomorphometry, and of (2) geomorphometry as a science amongst marine scientists with a wide range of background and experience.

**Keywords:** Geomorphometry, terrain analysis, marine habitat mapping, marine geomorphology, bathymetry, GIS.

# 1 Introduction

## 1.1 Background

Studies of geomorphology have improved our understanding of many of the Earth's systems and surface processes (Smith et al., 2011; Bishop et al., 2012). Morphology and quantitative measures of topography are considered the most important components of geomorphology because they represent the age and origin of the landscape (Speight, 1974; Minár and Evans, 2008; Bishop et al., 2012). The shapes of the terrestrial landscape are important for many Earth systems across a range of scales. For instance, broad-scale features such as mountains and valleys may dictate weather patterns (Dimri et al., 2013), vegetation and biodiversity patterns (Anderson and Ferree, 2010), and hydrological processes (Iordanishvili, 2000), while fine-scale features such as local slope may influence soil stability (Buscarnera and Di Prisco, 2013) or influence nest-site selection by certain bird species (Whittingham et al., 2002). Overall, topography is known to influence gradients in moisture, energy and nutrients across the landscape (Hengl and MacMillan, 2009). Likewise, the oceans play a fundamental role in the Earth system at multiple scales. Knowledge of seafloor topography is also crucial for many subjects (Smith, 2004). For example, seafloor topography, or bathymetry, influences surface currents (Gille et al., 2004), near-bottom currents (White et al., 2007), and ocean mixing rates (Kunze and Llewellyn Smith, 2004). Lack of knowledge on factors influenced by bathymetry can affect the efficacy of model predictions, for example models of marine species distributions (McArthur et al., 2010), climate (Jayne et al., 2004), or the paths of floating objects like marine debris (Smith and Marks, 2014).

It is commonly stated that 90% of the global ocean is unexplored (e.g. Gjerde, 2006) and that more is known about the surface of Earth's Moon, Mars, Mercury or Venus than about the ocean floor (Sandwell et al., 2002; Smith and Marks, 2014). However, such statements mean little without further specification or elaboration on their real meaning in relation to objectives, data types and spatial resolution. The entire ocean floor has been mapped to a resolution of a few kilometres using satellites, which has created an estimated surface of global bathymetry (Smith and Sandwell, 1994). However, these coarse-resolution data are often inadequate for many scientific, economic, public safety and management purposes. Applications such as tsunami hazard assessment, submarine cable and pipeline route planning, resource exploration, habitat mapping, territorial claims, navigation, and ocean circulation and climate studies all require more reliable, fine-scale bathymetric data (i.e. finer than 5 km) (Sandwell et al. 2002).

Fuelled by advancements in remote sensing and Geographic Information Systems (GIS) (e.g. Grohmann, 2004), the field of geomorphometry has entered a new era in recent decades (Evans and Minár, 2011; Florinsky, 2012). Geomorphometry is defined as the science on which quantitative measurements of terrain morphology are based, with foundations in geosciences, mathematics, and computer sciences (Chorley et al., 1957; Mark, 1975; Pike et al., 2009). It can be divided into

two sub-fields: general geomorphometry (e.g. Minár et al., 2013), and specific geomorphometry (e.g. Drăguţ and Blaschke, 2006). General geomorphometry deals with continuous surfaces in order to extract terrain attributes (e.g. slope, aspect, rugosity), while specific geomorphometry aims at characterizing or extracting discrete landforms (Evans, 1972). The science of geomorphometry, including its theories, methods, algorithms, and tools, was mainly developed and tested on artificial (e.g. Jones, 1998; Qin et al., 2013), terrestrial (e.g. Grohmann, 2015; Rigol-Sanchez et al., 2015) and extra-terrestrial settings

(e.g. Li et al., 2014; Podobnikar and Székely, 2015). These methods are relevant for underwater applications and have been increasingly used in the last decade (Lecours et al., 2015a), but differences in the nature of the input data (e.g. no need to hydrologically correct the surface model, little to no ability to validate measurements on the terrain) can sometimes produce different results than expected from land-based studies, creating the additional need for a dedicated scientific effort in marine geomorphometry. To our knowledge, no review on the state-of-the-art of marine geomorphometry has ever been written.

This contribution aims to raise awareness of the relatively recent field of marine geomorphometry by providing an overview of current practices and application areas and summarising the relevant literature to date. We first discuss the gradual rise in the application and development of marine geomorphometry (Sect. 1.2), from the first marine geophysical applications to the latest developments in marine habitat mapping and geomorphology. The paper then addresses the five main steps of geomorphometry identified by Pike et al. (2009) and adopted by the community (Bishop et al., 2012), with a focus on how

these steps are relevant to marine geomorphometry and different from traditional, terrestrial geomorphometry (Fig. 1). Section 2 reviews the first step of the geomorphometry workflow, which is to sample the surface. The characteristics of bathymetric data collected from four types of remote sensing techniques are described: satellite radar altimetry, optical remote sensing, acoustic remote sensing, and LiDAR (Light Detection And Ranging). Section 3, addressing the step in which we need to generate a surface model from the sampled heights, discusses elements that have implications on how the

seafloor is represented as data, including the interpolation methods used to create models and the spatial scale (i.e. spatial resolution and extent) at which to generate them. Section 4 addresses the pre-processing step that corresponds to making the surface model ready for the next step, surface analysis, which is reviewed in Sect. 5. The pre-processing of the surface model involves the correction for errors, artefacts and erroneous data in the surface model. The analysis of the surface is the core of the geomorphometric workflow and consists in deriving terrain attributes and terrain features (or objects). Finally, the last

step of this workflow is the use of the derived terrain attributes or features for a particular problem or application. The main disciplines in which marine geomorphometry has been applied and developed are examined in Section 6, and we also suggest other fields of research and applications that could benefit from the integration of marine geomorphometry in their practices. We conclude this review with recommendations and reflections on the future of marine geomorphometry.

## 1.2 The rise of marine geomorphometry

The science of geomorphometry has roots in morphography, hypsometry, cartometric, geophysics, and geomorphology (Pike et al., 2009; Evans and Minár, 2011; Evans, 2013). Following the increase in digital terrain models (DTM) availability in the 1960s, the underlying theories and mathematical developments of modern geomorphometry (i.e. based on quantitative measurements rather than qualitative observations derived from DTMs) started to be developed in the early 1970s (e.g. Carson and Kirby, 1972; Evans, 1972; Krcho, 1973; Schabber et al., 1979). These methods and algorithms were slowly automated in the 1980s as computers became more available (e.g. Horn, 1981; Imhof, 1982; Pike, 1988). However, constraints in computing power (Burrough, 1986) delayed the rapid expansion of geomorphometry until the early 1990s (Pike et al., 2009). As mentioned in recent reviews (e.g. Gessler et al., 2009; Evan and Minár, 2011, p. 105), the field of modern geomorphometry is a "young field" that is "still forming, with many concepts, methods and applications."

In the marine environment, early geophysical research that studied the link between the shape of the seafloor and elements such as global tectonics (Parsons and Scalter, 1977; Wessel and Chandler, 2011; references therein) led the way to the more recent marine applications of modern geomorphometry. The first applications of quantitative measurements derived from marine DTMs came from the field of marine geomorphology (e.g. Czarnecki and Bergin, 1986; Shaw and Smith, 1987, 1990; Malinverno, 1990; Goff, 1992, 2001), and was sometimes known as mathematical morphology or geology, or simply seafloor classification (Herzfeld, 1993). Then, the realization that different characteristics of seabed morphology was often linked to species distribution and biodiversity (e.g. Burrows et al., 2003; Giannoulaki et al., 2006), combined with the increase availability of higher-resolution bathymetric data (Smith and McConnaughey, 2016), opened a wide range of possibilities for marine habitat mapping in the mid-2000s (Bakran-Petriocili et al., 2006; Lundblad et al., 2006; Wilson et al., 2007). As discussed in Sect. 6, these two applications are still leading research areas in marine geomorphometry, as new applications slowly emerge. Despite the important amount of literature available on the qualitative description of terrain morphology from DTMs, for instance from the geophysical literature, the current review focusses on modern geomorphometry, i.e. the extraction from depth models of additional quantitative information describing terrain characteristics.

Figure 2 compares the increase in publications in both marine and terrestrial (and potentially extra-terrestrial) geomorphometry over time. The numbers illustrate that marine applications of geomorphometry are more recent and less numerous than their terrestrial counterparts. However, we note that the lower number of published marine applications indicated in Fig. 2 may be biased by the fact that the researched terms (e.g. geomorphometry and terrain analysis) are not always used in marine studies, even where geomorphometric techniques have been employed. Consequently, Figure 2 may be a reflection of how these terms have been adopted rather than a representation of the actual evolution of the practice. For instance, in Harris and Baker (2012a), an authoritative volume in marine habitat mapping, all of the 57 case studies used bathymetry, 33 of them generated slope, 23 of them measured rugosity, and 14 of them calculated a topographic position

index, amongst other terrain attributes (Harris and Baker, 2012b). Despite this high use of general geomorphometry techniques, the term "geomorphometry" was not once used in the 900 pages of the volume, and "terrain analysis" was only mentioned twice.

# 2 Sampling the depth of the seafloor

For centuries, the lead line was the main instrument used to determine the depth of the seafloor, until remote sensing technologies revolutionized the way we could measure bathymetry. This section introduces four types of remote sensing technology that are currently used to collect depth information and found in the geomorphometry literature: satellite radar altimetry, optical remote sensing, acoustic remote sensing, and bathymetric LiDAR. From an application perspective, the survey methodology or methodologies dictate the spatial scale (i.e. resolution and extent) of the final surface model. First,

the fundamental technical limitations of the remote sensing technique that is used to collect bathymetric data will define the scale (resolution) of the surface model (Kenny et al., 2003; Van Rein et al., 2009). For instance, radar altimetry data limits models constructed with them to coarse, usually kilometre-scale resolution, while other methods can achieve up to centimetre-scale resolution models. Second, by defining the distance between the platform and the target, the characteristics of the latter may also influence the scale (extent) of the final model; a remotely sensed image collected from a satellite will

usually have a coarser resolution and cover a larger area than an image collected from an aircraft or an unmanned aerial vehicle. While radar altimetry and acoustic remote sensing are limited to deeper waters and optical remotely sensed and bathymetric LiDAR data are limited to shallower waters, there is some degree of overlap between the depths in which the various methods can be applied. In terms of effort, systematic bathymetric survey could be performed with satellite-based methods within a few years at a global scale (Sandwell et al., 2002), compared to the estimated 600 years (Carron et al.

2001) that it would take using acoustic remote sensing technologies. The different techniques are discussed in the perspective of using the information they collect to generate DTMs and perform geomorphometric analyses. DTMs using bathymetric data are hereafter referred to as Digital Bathymetric Models (DBM) to distinguish them from Digital Elevation Models (DEM), a term usually reserved for terrestrial elevation data. Other techniques can be used to measure depth but are less common in the literature (e.g. ground-penetrating radar, Feurer et al., 2008). More details on the underlying theories of

these four techniques can be found in Appendix 1.

## 2.1 Satellite radar altimetry

In the 1970s, satellite-based radar altimeters were developed (cf. Appendix 1) as a method to study the oceans on a global scale (Douglas et al., 1987), which was a significant improvement over the extent covered by very narrow ship tracks. Consequently, the applications of altimetry-derived bathymetric data are limited to the study of broad-scale patterns,

processes and features as they only provide low resolution estimates of bathymetry (Goff et al., 2004). Technological

constraints and satellite orbits also prevent data collection close to the poles and the coastline (Sandwell et al., 2002). Some authors identified weaknesses in the method and warned that predicted depths from altimetry may not be reliable and should not be used for geodynamics studies (Smith, 1993), navigation, or hazard identification (Smith and Sandwell, 1994). The main advantages of altimetry-derived bathymetry are speed of collection and uniformity of coverage (Mackenzie, 1997).

Two main altimetry-derived datasets are currently used in applications of marine geomorphometry: the General Bathymetric Chart of the Oceans (GEBCO, 2014) and the Shuttle Radar Topography Mapping 30-arc second database (SRTM30_PLUS, Becker et al., 2009). They are both free datasets that combine together elevation and bathymetric data. The bathymetric parts were created by filling the gaps between publicly available datasets from different sources with radar altimetry (Smith and Sandwell, 1994; Becker et al., 2009). These datasets have been used for instance in habitat mapping and predictive

modelling (e.g. Davies et al., 2008; Knudby et al., 2013), conservation (e.g. Ross and Howell, 2013), search and rescue operations (Smith and Marks, 2014), and geomorphology (e.g. Harris et al., 2014). Many works have found these datasets to be too coarse for their purposes (e.g. Davies et al., 2008; McNutt, 2014). For instance, Vierod et al. (2014) stated: "At present, the availability of bathymetric data at a resolution sufficient to inform reliable terrain attribute predictors is a major limitation to the ability of deep-sea species distribution models to make accurate predictions of the distributions of benthic

organisms." For many applications, quality can also be just as important as resolution (cf. Sect. 4 and Sect. 6).

## 2.2 Optical remote sensing

Of the four remote sensing methods presented in this section, optical remote sensing is the least common in the marine geomorphometry literature. However, it presents a cheaper alternative to LiDAR data for collecting depth information in very shallow coastal areas (Su et al., 2014), as satellites can cover large areas in less time (Lafon et al., 2002; Wang and

Philpot, 2007). Two main optical remote sensing groups of methods are usually used to estimate bathymetry from optical remote sensing: one based on the interactions of electromagnetic radiations with water and one based on stereoscopy (see Appendix 1).

Methods based on electromagnetic radiations are limited to shallow waters because of light attenuation within the water column (Jawak et al., 2015). In theory, based on light penetration in coastal waters, depths down to 50 m could be retrieved

(Speight and Henderson, 2010). However, the practical limit varies with local sea conditions. A maximum of 30 m deep is usually achieved when local conditions are exceptionally good (Collet et al., 2000; Jawak et al., 2015), and most often a depth of 15 m is reported as being the performance limit of optical remote sensing for bathymetry retrieval (e.g. Stumpf et al., 2003). Optical methods are sensitive to errors caused by waves, turbidity, sunglint from specular reflection, heterogeneous and complex seafloors, and the presence of shadow that artificially increases depth estimates (Lafon et al.,

2002; Louchard et al., 2003; Holman and Haller, 2013; Eugenio et al., 2015). Some of these elements can be corrected or accounted for. For instance, Knudby et al. (2010) applied 'deglinting' (Hedley et al., 2005; Kay et al., 2009) and water

column corrections (Lyzenga, 1978), in addition to the common geometric and atmospheric corrections, to IKONOS satellite images to create a 4 m resolution DBM from which was derived measures of seafloor rugosity. The authors indicated, however, that noise prevented the use of bathymetry at depths deeper than 15 m as rugosity values were artificially increased.

5  For stereoscopy-based methods, photogrammetry applied to pairs of stereo images can be used to build DBMs in a similar way than what is done on land. Although possible (e.g. Stojic et al., 1998), through-water photogrammetry is challenging due to the need to correct for the air-water interface (Feurer et al., 2008). However, underwater photogrammetry (i.e. active remote sensing) has been successfully applied at a fine scale to reconstruct the digital terrain (e.g. Johnson-Roberson et al., 2010; Kwasnitschka et al., 2013). The work by Friedman et al. (2012) is noteworthy as they derived multi-scale measures of 10  rugosity, slope and aspect from underwater stereo image reconstructions.

## 2.3 Acoustic remote sensing

The development of acoustic technologies has fuelled marine exploration probably more than any other method, by providing reliable swath coverage and relatively high-density data at ever-decreasing price per line kilometre (see Lurton, 2010 for review). Three types of active sonars (Sound Navigation and Ranging) can be used to collect depth estimates and/or 15  backscatter information using sound waves (see Appendix 1): sidescan sonars (SSS; reviewed in Blondel and Murton, 1997), single-beam echosounders (SBES), and multibeam echosounders (MBES; reviewed in de Moustier, 1988). Backscatter data is effective in providing information on seafloor properties (e.g. sediment composition). These tools can be pole-mounted on the side of vessels, or mounted on the hull of vessels, on remotely operated vehicles (ROV), on autonomous underwater vehicles (AUV), or on a towed platform.

20  SSS provide an acoustic image of the seafloor from backscatter measurements that can inform on topographic roughness. They can only provide bathymetric measurements when data from two receiving antennas are combined and principles from interferometry are applied. SSS are a commonly used seafloor technology because they are easy to deploy and cheaper than other acoustic technologies (Harris and Baker, 2012c). The acoustic image quality of SSS images are very high resolution, and characteristics (e.g. length and shape) of the acoustic shadows, which are the areas on a SSS image that have null 25  intensity values because the sound was blocked by an object of feature higher than its surroundings, enable the estimation of the height and size of these objects or features (Blondel and Murton, 1997). Collier and Humber (2012) provide an example of the use of sidescan-derived bathymetry to identify geomorphic features on the seafloor. Some techniques from specific geomorphometry are used on backscatter data to identify specific bedforms or depositional units on the seafloor based on their unique acoustic signature (Greene et al., 1999; Huvenne et al., 2005; Martorelli et al., 2012) and to detect differences in 30  reflectivity and texture patterns on the seafloor (van Lancker et al., 2012). We also recognize the potential to generate higher resolution (centimetre-scale) bathymetric data using modern synthetic aperture sonar systems such as the HISAS 1030

(Kongsberg Maritime, 2015) from a stable AUV platform (e.g. Ludvigsen et al. 2014). There are many potential benefits to this approach although, at present, the processing of bathymetric data is very computationally demanding and therefore best suited to mapping of small areas.

SBES, or fathometers, collect both depth and backscatter data by transmitting a single sound beam at nadir. The mapped extent is thus limited to a single track directly below the supporting platform. Although they remain standard for ships navigation, SBES are less and less used for mapping purposes since MBES became more affordable. However, recent applications can still be found, particularly where compiled SBES data are available; a SBES bathymetric dataset of the English Channel was used by Coggan and Diesing (2012) for the broad-scale analysis of an exposed rock ridge system, by Elvenes et al. (2014) for surficial sediment and habitat mapping, and by James et al. (2012) to identify geomorphic features in a paleo-valley.

MBES provide a relatively fast, high-resolution and wide coverage measurement of the seafloor. They sweep a large swath of the seafloor by emitting a fan of narrow sound beams, and are currently the most efficient and accurate tool available to collect bathymetric and backscatter data (Costa et al., 2009; Schimel et al., 2010a, 2010b). In recent years, advancement in MBES technology has further enhanced a valuable source of seafloor data. These advances have come out of traditional user groups extending the application of the data to meet new requirements and from the motivation of new user groups wanting to employ the technology. This wide ranging and ever growing community of MBES users are adapting and extending the potential of MBES data to address unique applications. MBES users have traditionally included hydrographers, navigators, engineers, marine geologists and military planners, but now we see the extension of the technology to meet the needs of maritime explorers, archaeologists, fisheries biologists, geomorphologists and ecosystem modellers, to name a few. MBES is currently the main source of bathymetric data for applications of marine geomorphometry, although these data are limited in terms of coverage: "Multibeam soundings are the gold standards, but such mapping has been concentrated in coastal zones, along shipping lanes, and in regions harboring hydrocarbon or mineral deposits" (Normile, 2014, p. 964).

## 2.4 Bathymetric LiDAR

Bathymetric LiDAR is an adaptation of the more traditional airborne topographic LiDAR (Guenther et al., 2002, see Appendix 1) and has become increasingly common in the literature in the last two decades (Brock and Purkis, 2009). Recently, they have been combined into topo-bathymetric LiDAR, which are multispectral systems that enable data collection both above land and water; when flying over the water, a green laser – characteristic of bathymetric LiDAR – penetrates the sea surface and collects information on the water column and the seafloor, while the red/infrared laser – characteristic of topographic LiDAR – collects information on the sea surface. LiDAR can also collect intensity values that, like acoustic backscatter, provide information on the characteristics of the seafloor (Costa et al., 2009; Kashani et al., 2015).

Bathymetric LiDAR is the only technique that can collect high-resolution data in very shallow waters, which makes it especially relevant for coastal applications requiring fine-scale data (<1 m resolution) (Brock and Purkis, 2009). The efficiency of bathymetric LiDAR systems is greatly limited by turbidity, wave action, depth (up to 50-70 m in exceptionally good conditions), steep slopes, and rocky substrate (Costa et al., 2009; Chust et al., 2010; Jalali et al., 2015). Current geomorphometric applications on bathymetric LiDAR data are mainly related to the exploration of coastal ecosystems (e.g. Wedding et al., 2008; Zavalas et al., 2014) and geomorphology (e.g. Arifin and Kennedy, 2011; Kennedy et al., 2014), but are likely to extend to other applications such as marine archaeology and natural hazards assessment (e.g. Solsten and Aitken, 2006). In 2015, LiDAR data represented 4.5% of the coastal data collected for the Continually Updated Shoreline Product (CUSP) compiled by the National Oceanic and Atmospheric Administration (NOAA) and the National Geodetic Survey (NGS) of the United States (Graham et al., 2015).

# 3 Generating a surface model from sampled depths

A detailed account of the various approaches to processing (i.e. georeferencing and applying system and environment related corrections) and cleaning of data (i.e. removal of spurious depth soundings) are beyond the scope of this paper and are specific to the sensors used for data acquisition as well as industry and application-related standard practices. In this section, we focus on the interpolation of data and only touch briefly on data cleaning when we present a method where uncertainty algorithms are used to aid data cleaning, and where the interpolation of data is intrinsically linked to the calculation of uncertainty of the bathymetric surface.

By nature, geomorphometric analyses necessitate spatially continuous data, but no remote sensing techniques used to collect depth samples create truly continuous surfaces. Hengl and Evans (2009) identified several techniques used to generate gridded DTMs from height samples for geomorphometric purposes, including inverse distance weighting (IDW), minimum curvature, spline, kriging, polynomial regression, moving average, and many others. The same methods can all be used to generate DBMs from depth samples. For instance, Ezhova et al. (2012) created a DBM from SBES data using the natural neighbour interpolation method, and Ramillien and Cazenave (1997) combined altimetry and ship-based data into a single DBM using bilinear interpolation. More rarely, triangulated irregular networks (TIN) are created from depth samples; for example, Heyman and Kobara (2012) generated a TIN from SBES data, and Foster et al. (2009) computed TINs from SBES and bathymetric LiDAR from which volumetric attributes were computed. The choice of interpolator varies depending on the type of data and the spatial arrangement of the depth samples. For instance, MBES or LiDAR data can collect very dense point clouds that require little interpolation between points, resulting in limited interpolator influence in the final DBM. On the other hand, creating a DBM for a big area from SBES data requires more interpolation as SBES only sample very narrow tracks and have a high density of points along the survey line but no data between the survey lines. This has implications for geomorphometry as the interpolated DBM may miss important geomorphological features (depending on the distance

between the survey lines), similarly to what happens with the interpolation of contour lines (Wise, 1998). Also, some methods (e.g. IDW) do not extrapolate and are hence less accurate in cases of sparse sampling. The choice of an appropriate interpolator to generate a surface model is critical as some interpolators may produce erroneous depth values that do not adequately represent the real bathymetry (Smith and Wessel, 1990).

There are no optimal interpolation methods (Li et al., 2005), and it is well known that each technique has different sensitivity to errors and sample distribution and that the quality of DTMs can be improved when making the appropriate choice of interpolator (Carrara et al., 1997; Hengl and Evans, 2009). For instance, some techniques will consider all samples while other will ignore outliers or smooth out their effect. By being different in nature, sampled depths may not require the same characteristics from an interpolator than sampled elevations; for instance, DBMs do not need to be hydrologically corrected

as drainage analyses are futile underwater. This is why techniques were developed in recent years to address the particular characteristics of depth sampling. Here we examine on such technique, the CUBE (Combined Uncertainty and Bathymetric Estimator) algorithm (Calder, 2003), which accounts for different errors specific to acoustic remote sensing (e.g. geometric and acoustic) and is incorporated in several of the most widely used bathymetric processing software used by the hydrographic survey industry and scientific community. Although not yet universally accepted as data cleaning method by

the hydrographic survey industry and hydrographic agencies, who have a particular need to preserve shoal soundings and comply with strict quality control procedures to ensure safety of navigation, CUBE is widely used and of special interest to more applied bathymetric data users and the related scientific community. According to Schimel et al. (2010b), CUBE could be more appropriate than traditional gridding methods to compute precise bathymetry and associated terrain attributes. CUBE is based on the spatially explicit quantification of the total propagated uncertainty (TPU) for each data point (Calder

and Mayer, 2003), enabling the rejection of samples that are outside a certain uncertainty confidence level (e.g. 95% for Calvert et al., 2015). When creating the DBM, the algorithm provides vertical error estimates and statistically assigns, to each pixel, the most likely depth value based on the uncertainty of each sounding within the pixel (see Dolan and Lucieer, 2014). In several bathymetric processing software offering CUBE, users can visualize not only the most probable depth for each pixel, but also the subsequent most probable depths (e.g. second or third most likely) and select the one they think is the

most appropriate based on their knowledge of an area. For example, this can allow correction for occurrences when the sonar detects fish close to the seafloor instead of the seafloor itself and data cleaning did not appropriately remove these soundings. Figure 3 shows some of the information that can be extracted and visualised from the application of the CUBE algorithm. When interpolating the soundings to create a DBM, the bathymetry and the horizontal and vertical components of uncertainty can be stored in a BASE (Bathymetry Associated with Statistical Errors) surface. The BASE format allows

multi-attributes surface models. CUBE's main inconvenience is that it requires a lot of ancillary data to be collected in order to compute TPU, but it is very reliable in defining the spatial pattern of errors, their importance, and helping to identify their sources (Passalacqua et al., 2015). In addition to the bathymetry, a map of uncertainty can be computed, which can become very important when making decisions using the bathymetry and for onward geomorphometric analysis.

Spatial scale is an important component of the interpolation of depth data to generate DBMs, and differences in sampling characteristics have an impact on the spatial scale of the resulting surface model. Unlike systems used in optical remote sensing, radar altimetry and bathymetric LiDAR, acoustic systems do not sample the seafloor uniformly, which influences the spatial scale of the resulting DBM. The sampling density of these systems is directly dependent on depth, or more specifically on the sensor-to-seafloor distance (Lecours and Devillers, 2015). For instance, as the distance between a MBES and the seafloor increases, it takes longer for the sound to reach the seafloor, the system's footprint and related beam widths increase, leading to a lower sampling density for a greater area sampled at a coarser resolution. Since the seafloor is rarely perfectly flat and at a constant depth, the sampling density is almost never uniform across survey areas, which can make it challenging to determine the appropriate spatial resolution of DBMs for interpolation. Ultimately, the spatial scale of a DBM will be dictated by its intended use (see Sect. 6) which then influences the choice of the data collection method, typically following hydrographic standards (IHO, 2008) which ensure the appropriate data are acquired to ensure safety of navigation. Besides DBMs created directly from one source of survey data, we are increasingly seeing DBMs generated, or pooled together from several surveys and/or sensor technologies (e.g. EMODnet, 2015). These datasets can be a valuable resource but impose additional challenges for DBM creation and geomorphometric analysis (e.g. Sect. 6.4.3).

# 4 Correcting errors and artefacts in digital bathymetric models

In terrestrial applications, it is well known that all DEMs, regardless of the techniques used to collect and generate data, are influenced by uncertainty and errors (Fisher and Tate, 2006; Gessler et al., 2009). This is also true for marine applications, but the properties and dynamic nature of the ocean makes DBMs more prone to errors and artefacts than DEMs (Hughes-Clarke et al., 1996). As illustrated in Fig. 4, this has significant implications for marine geomorphometry, which shows as widely recognized in the terrestrial literature (Florinsky, 1998; Zhou and Liu, 2004; Oksanen and Sarjakoski, 2005) that errors and artefacts in a DTM propagate to and may be amplified in terrain attributes.

As with DEMs (Harrison et al., 2009; Sofia et al., 2013), errors and artefacts in DBMs can be caused by the interpolation method (Erikstad et al., 2013), movement and positioning of the supporting platform (Hughes-Clarke et al., 1996), and a temporal (Lecours and Devillers, 2015) or spatial (Hughes-Clarke, 2003a, 2003b) misalignment between the different elements of the surveying system. Data from radar altimetry are the least sensitive to platform motion (Smith and Sandwell, 1994). However, large artefacts resulting from fine-scale noise in the gravity field (Goff et al., 2004) and the algorithms used to convert gravity data into bathymetric estimates (Dixon et al. 1983; Calmant and Beaudry, 1996) are often characteristic of these data. Similar large linear artefacts can sometimes be found in satellite images (e.g. Klemas, 2011a). The level of error in bathymetric data from optical remote sensing is known to directly depend on water depth as a result of light attenuation in the water column (e.g. Liceaga-Correa and Euan-Avila, 2002). Recent studies (Leon et al., 2013; Hamylton et al., 2015) have demonstrated that the integration of the spatial structure of errors improves bathymetric estimates derived from satellite

images. Data collected with acoustic methods are the most susceptible to artefacts for several reasons. First, they are collected from surface vessels/platforms or underwater vehicles that can be strongly affected by environmental conditions such as waves and wind. Furthermore acoustic waves need to be corrected for sound velocity. Without this correction the data will exhibit artefacts broadly similar to those caused by an inappropriate correction of the atmospheric conditions in
optical remote sensing (Li & Goldstein, 1990). Sound velocity varies with temperature, salinity and pressure and the failure to correct for these variations can induce refraction artefacts in the DBM (Yang et al., 2007). This is particularly challenging as water column properties vary both spatially and temporally, especially in the coastal zone where there is the additional complication of freshwater input from rivers, and are less predictable than atmospheric conditions (Cushman-Roisin and Beckers, 2011). Tidal corrections are generally applied using data from locally installed tide gauges, or modelled tides, depending on the accuracy required. Finally, since the surveying system is underwater, direct positioning using the Global
Positioning System (GPS) is not possible (Roman and Singh, 2006). The level of error in the data is thus strongly influenced by the accuracy of the different instruments that provide ancillary data to estimate the position of the system's underwater components (Rattray et al., 2014; Lecours and Devillers, 2015).

Figure 4 illustrates different types of errors and artefacts that can be found in bathymetric data of different types and at
different scales, and their propagation to derived terrain attributes. Artefacts commonly found in bathymetric data and that often cannot be corrected using existing methods include gridding and interpolation artefacts (e.g. in the top panels), motion artefacts (e.g. middle and bottom panels), refraction artefacts, and artefacts caused by the temporal or spatial uncertainty associated with ancillary data (e.g. bottom panels). Common errors include spurious soundings (e.g. bottom panels). Artefacts in DBMs are difficult to handle properly as depth generally cannot be ground-truthed, thus preventing verification
of whether or not a feature is natural or the result of an artefact (Li and Wu, 2006). Most marine environments are not easy to access and the collection of ground-truth data is often limited by technological and logistical constraints (Solan et al., 2003; Robinson et al., 2011). Consequently, ground-truthing of DBMs is not standard practice. We note however that ground-truthing is often performed for backscatter data to attempt matching sediment types with acoustic reflectivity characteristics. As illustrated in Fig. 4, artefacts in DBMs may be present at all scales and persist, or are sometimes amplified in derived
terrain attributes. For instance, artefacts in the GEBCO dataset are common (Lecours et al., 2013; GEBCO, 2014; Fig. 4), arising mostly from the merging of datasets of different quality. When the artefacts are large they dominate the surface and cannot be removed with traditional filtering methods (e.g. Gaussian filtering) as this considerably affects the overall quality of the surface (Passalacqua et al., 2015), and the artefacts are also difficult to overcome when deriving terrain attributes even by using multi-scale methods (Sect. 5.1). At a finer scale, Yang et al. (2007) developed an algorithm to correct refraction
artefacts, although this was only partially successful. When the artefacts are smaller, it can be difficult to distinguish them from real fine-scale features such as sandwaves or iceberg scourings (Hughes-Clarke et al., 1996), especially when no underwater video data are available to confirm the geomorphology of an area. This is particularly challenging for marine geomorphometry as analyses are likely to capture both the real features and the artefacts (Wilson et al., 2007). Currently, the

main ways to address artefacts in DBMs are to apply filtering techniques, resample the data to coarser resolutions, manually correct the data based on visual interpretation, and to use algorithms like CUBE that account for errors and uncertainty. Most marine geomorphometry applications simply disregard the presence of the remaining artefacts, excluding them for practical purposes by expert judgement.

# 5 Deriving terrain attributes and terrain features

Bathymetric data, particularly full coverage multibeam, or LiDAR data, are well suited for the generation of quantitative terrain attributes and terrain features. These attributes and feature classifications can be very useful in describing, interpreting and classifying geomorphology in the marine environment, just as their terrestrial equivalents are on land. These derived datasets can also be of further use in many applications (cf. Sect. 6). With bathymetric data now available in many areas at comparable resolutions to terrestrial DEMs, depending on the survey equipment used (cf. Sect. 2), we can extract a similar level of information to that obtainable from terrestrial DEMs. Elsewhere, global (e.g. GEBCO, 2014) and regional (e.g. IBCAO (Jakobsson et al., 2012); EMODnet, 2015) bathymetric datasets combining information from many sources have become an impressive resource and are being used routinely for many marine science applications, not least those including high seas areas which, as yet, have little detailed coverage. This section reviews both the use of general (i.e. terrain attributes) and specific (i.e. terrain features) marine geomorphometry.

## 5.1 General geomorphometry (terrain attributes)

The calculation of terrain attributes (synonymous with terrain/topographic variables) first requires some method for mathematically representing the bathymetric surface. This surface representation is then used to calculate the required terrain attribute, and is typically achieved by either using neighbourhood analysis of raster pixels, or by fitting a polynomial expression to describe the surface. The computations performed on DBMs, and the range of applications that these derived terrain attributes are used for, are common to many of those performed on DEMs for terrestrial applications. Differences in analysis of bathymetric DBMs versus DEMs are often more related to the meaning or application of the information from the analysis. For instance, deriving a watershed network underwater may be useful e.g. for delineating potential sediment pathways on the continental slope, but is a deviation from the original intended purpose. A review of terrain attributes was provided by Wilson et al. (2007) in the context of marine benthic habitat mapping and updated by Dolan et al. (2012). In addition, Brown et al. (2011) offer a useful summary of the extent to which many of these various terrain attributes have been employed within published habitat mapping studies in the period 2000 to 2011. Habitat mapping is currently one of the largest application areas for these techniques. To our knowledge no equivalent reviews on the use of general geomorphometry exist for more general marine geomorphological or other application areas.

Terrain variables can be grouped into four main types describing different properties of the terrain – slope, orientation, curvature/relative position, and terrain variability (Fig. 5). It is beyond the scope of this paper to provide details on all the various options for computation, however, we provide an overview of some of the most commonly used terrain attributes in marine-based studies, as well as an indication of some common calculation approaches (Table 1). Here we note the geomorphological relevance and ecological relevance of the various types of terrain attributes in the context of seabed mapping. Whilst the effects on geomorphology are more direct, the popularity of terrain attributes in benthic habitat mapping is, to a large extent, due to their function as a surrogate (or proxy) in explaining the distribution of benthic fauna. In the absence of better, or alternate information (e.g. gained from high resolution oceanographic data), proxy information such as whether a given location is sheltered or exposed to dominant currents as indicated by its position relative to neighbouring terrain, can be useful in determining suitable habitat for a given species or community. An elevated position for example may be advantageous for suspension feeding organisms and act as a surrogate for the direct need for food supply. Other terrain attributes may capture a proxy for shelter or other ecological advantage. This topic is discussed further by Lecours et al. (2015b) including the all-important effect of scale which is linked both to data resolution and the scale at which geomorphometric analysis is conducted.

For GIS based calculation of terrain attributes, extending the analysis window beyond the basic 3 x 3 neighbourhood is particularly useful in marine geomorphometry as it facilitates the identification of spatial scales that are relevant to benthic communities (Lecours et al., 2015b) or to geomorphological interpretation (Shaw, 1992) and may also help to overcome artefacts in the DBM (Wilson et al., 2007). The multi-scale analysis methods developed by Wood (1996), which built on the work of Evans (1972, 1980), have been fundamental in establishing an appreciation of scale in marine and terrestrial geomorphometry alike. The associated software package Landserf (Wood, 2009) puts multi-scale analysis within easy reach of marine scientists and the use of Landserf for DBM analysis took off following the early applications of the software to bathymetric data (e.g. Wilson et al., 2007). Although Landserf 2.3 is still used by many scientists requiring a standalone programme for geomorphometric analysis, Wood's algorithms are now perhaps more widely used among the marine community through the GRASS module r.param.scale. The newly released ArcGeomorphometry toolbox (Rigol-Sanchez et al., 2015) offers a means to access the Wood-Evans (and other) algorithms for geomophometric analysis, and has the potential to provide a long awaited, convenient multi-scale analysis option for ArcGIS users.

One terrain attribute that is specifically tailored to analysis of bathymetric data is the bathymetric position index (BPI) (Lundblad et al., 2006), which is an adaptation of Weiss' (2001) topographic position index (TPI) and a useful measure of relative position that is simple to calculate over different neighbourhood sizes. Although a relatively simple algorithm to implement (Lunblad's BPI indices can be performed through the raster calculator (e.g. Wilson et al., 2007) or scripting), many marine scientists make use of the Benthic Terrain Modeler (BTM) Toolbox, which was first developed following Lundblad's (2006) study. The current version of BTM (Wright et al., 2012) for ArcGIS 10.1 and later has seen around 4000 downloads in the period 2012-2015, a figure that gives a conservative estimate of how many scientists are actually using the

tool (S. Walbridge, ESRI, pers. comm.). The BTM toolbox relies on ArcGIS Spatial Analyst and includes tools for calculating slope, aspect and terrain variability (rugosity, VRM) as well as methods for combining these into geomorphic zones. It was launched for the scientific community at a time when multibeam data was becoming widely available and modern marine geomorphometry was becoming established. The BTM toolbox quickly became popular as a one-stop shop for terrain analysis and classification of bathymetric data, offering a slightly more tailored solution, and the ability to handle larger datasets, than Landserf, with at least the BPI index being computable at different scales since the first release (now joined by VRM). The utility of the BTM tool has been augmented in recent years through updating of the terrain variability and aspect indices, and by providing the tools as both an AddIn and as a standalone ArcToolbox, providing greater flexibility to users who may wish to benefit from all, or just part of, the functionality.

Several bathymetric data processing software (e.g. CARIS HIPS and SIPS, QPS-Fledermaus) also have built in tools for calculation of basic indices such as slope and rugosity, bringing the functions directly to the bathymetric data user and removing the need to search for and select from the vast array of available methods. This has advantages of convenience for some bathymetric data users, but in most applied projects the computation of terrain attributes and further analysis will be conducted in some generic GIS software. Although many of the commercial software are currently limited to single scale analysis (3 x 3 rectangular neighbourhood) it has become easier to find tools for multi-scale analysis, either directly in open source software (e.g. GRASS), through additional toolboxes (e.g. SEXTANTE for QGIS), or via scripting. Many of these also give alternative choices for computation algorithms, the effects of which are investigated by Dolan and Lucieer (2014) using slope as an example.

Terrain variability has been a particularly popular terrain attribute in relation to benthic habitat mapping. This is largely due to the generally accepted link of terrain variability with biodiversity, which has, however, not yet been fully established with regard to spatial scale (Lecours et al., 2015b). Several measures of terrain variability have been applied to DBMs (Table 1) with some proving suitable for multi-scale analysis and others becoming problematic at larger analysis scales (Wilson, 2006). A rugosity index which is the ratio of surface area to planar area (Jenness, 2004) remains perhaps the most widely applied method in marine studies and this was implemented in early releases of BTM. Both the vector ruggedness measure (VRM; Sappington et al., 2007) now incorporated in BTM, and the more recent Arc_Chord Rugosity measure (Du Preez et al., 2014) offer alternatives that are better decoupled from slope. Where slope and a terrain variability measures are to be used in further analysis e.g. as predictor variables for habitat modelling, it is particularly important that the user is aware of any autocorrelation or covariation between these attributes, so they can be handled appropriately. Du Preez et al. (2014) lists several marine studies among those who have ignored the need for decoupling. However, with methods like VRM and Arc-Chord rugosity, or toolboxes like BTM and TASSE (Lecours, 2015) now readily available, we trust that future studies will make a conscious choice of the best geomorphometric analysis to use for their particular application.

## 5.2 Specific geomorphometry (terrain features/objects)

Compared to general geomorphometry and the use of terrain attributes, applications of computer-based specific geomorphometry are still relatively rare in the marine environment. Calculation of terrain features generally relies on the combined properties of several terrain attributes. For instance, Lecours et al. (2013) used Troeh's landform classification (Shary et al., 2005), which uses different types of curvatures to identify zones of relative deflection or accumulation and transit zones, on bathymetric data. The authors also adapted Weiss' (2001) landform classification, which combines slope with TPI measures at different scales to identify up to 16 landform classes, for application within the marine environment using BPI measures.

Terrain features such as crests and troughs can be extracted through the use of pixel based analysis (e.g. Blaszczynski, 1997; Wood and Dragicevic, 2007), but object-oriented methods for landform classification have recently become increasingly popular and are beginning to make their mark on marine studies (e.g. Lawrence et al., 2015) driven by an opportunity to analyse the DBM in conjunction with acoustic backscatter data (an indicator of seabed sediment type) rather than analysing the DBM alone, which offers several advantages for seabed classification. Geographic Object-Based Image Analysis (GeOBIA, OBIA) has been gaining some traction in the seabed mapping community as the spatial resolution of acoustic backscatter data improves (Diesing et al., 2014). The basic processing units in object based image analysis are objects which are represented by textural changes in the acoustic backscatter image and are constrained by derived topographic variables (Benz et al., 2004). GeOBIA allows for the quantitative extraction of image textures and features to be identified in the backscatter data and the ability to relate these spatially to topographic variability (Costa and Battista, 2013). Multi-resolution segmentation is one of the most popular segmentation algorithms to delineate homogeneous seabed segments (Lucieer, 2008; Lucieer and Lamarche, 2011; Hasan, 2012; Eisank et al., 2014) and in the terrestrial literature stands out as the most successful method to delineate homogeneous terrain segments rather than landforms *per se* (e.g. Drăguţ and Blaschke, 2006; Drăguţ et al., 2011; Blaschke et al., 2014). This has been successfully demonstrated by Ismail et al. (2015) to identify and classify submarine canyons. By combining both the spatial derivatives of the DBM with GeOBIA variables, the authors were able to perform an automated multiple scale terrain analysis to discriminate local and broad scale geomorphic features in the marine landscape. This information was used not only to delineate geomorphic seafloor features but also to identify properties that might influence biodiversity in a complex terrain. Specific geomorphometry is currently not used to its full potential in the marine environment.

# 6 Applications of marine geomorphometry

Pike et al. (2009) listed current and potential applications of marine geomorphometry, including oceanography, coastal geomorphology, geophysical analysis of global tectonics, ocean currents, mineral exploration, fisheries managements,

navigation, and concealment of nuclear submarines. While performing the meta-analysis that enabled the making of Fig. 2, we were able to classify marine geomorphometry articles into four main research areas: geomorphology, geophysics and geohazards, habitat mapping, biogeography and ecology, hydrodynamics and modelling, and others. This section thus introduces the most common applications of geomorphometry in the marine environment, and discusses in Sect. 6.4 the least common uses in addition to potential future applications of marine geomorphometry. A selection of published works that have utilised geomorphometric techniques in their study of seafloor morphology is provided in Table 2.

## 6.1 Marine geomorphology, geophysics and geohazards

Early geomorphometric studies of seafloor morphology in the 1960s were limited by the one-dimensionality and the low resolution of the bathymetric data that were available at the time (e.g. Krause and Menard, 1965; Neidell, 1966). In the last three decades, improvements in seafloor surveying technologies have resulted in a renewed interest in employing geomorphometric techniques to study seafloor geomorphology. Similar techniques have also been utilised in the interpretation of sidescan sonar data (e.g. Blondel et al., 1998; Carmichael et al., 1996; Huvenne et al., 2002; Mitchell and Somers, 1989).

Geomorphometric techniques have generally performed well in submarine environments. The use of specific geomorphometric techniques, where features of interest are identified prior to analysis, has involved examining how different morphological parameters change spatially and with each other. They have been amongst the most successful techniques, particularly with regard to the study of submarine mass movements, canyons and volcanoes. In the study of submarine mass movements, the general approach has been the prior identification of the boundaries of the landslides, the measurements of a series of morphometric parameters and their spatial and statistical analyses. This kind of approach has been applied to slope instability offshore Norway (Haflidason et al., 2005; Issler et al., 2005; Micallef et al., 2008), demonstrating the fractal characteristics of submarine mass movement morphology and statistics, which has important implications for submarine landslide modelling and hazard assessment. It has also been employed on a finer scale (Casalbore et al., 2011; Rovere et al., 2014) and a broader scale (Hühnerbach et al., 2004; McAdoo et al., 2000; Moernaut and De Batist, 2011) to identify tsunamigenic landslides and to provide interesting insights into failure frequency, preconditioning factors, triggers and controls of submarine mass movements in a wide range of environments, including lakes. In submarine canyons, specific geomorphometric analyses of submarine landslides has shown that landslides can be the most efficient process removing material from canyons and that their influence becomes more significant as the canyon matures (Green and Uken, 2008; Micallef et al., 2012). Geomorphometric investigations of submarine canyon form have generally focused on using morphological data to propose model of canyon erosion by turbidity currents (Mitchell, 2004, 2005; Vachtman et al., 2013). More recently, specific geomorphometric techniques have been used to demonstrate how canyons in passive, progradational margins develop into geometrically self-similar systems that approach steady state and higher drainage efficiency (Micallef et al., 2014b), and how canyons in active margins fail to reach steady-state because of continuous adjustment to

perturbations associated with tectonic displacements and base-level change (Micallef et al., 2014a). The geomorphometric study of volcanoes has been useful in determining the key processes constructing and modifying volcano flanks and specifying the conditions that lead to slope instability (Mitchell, 2003; Mitchell et al., 2002; Stretch et al., 2006).

Initially, the techniques of general geomorphometry used in the study of submarine landscapes were less numerous and varied than those used in the study of subaerial landscapes. The majority of studies where geomorphometry was applied to the study of submarine landscapes have involved either spectral analyses of the bathymetric data or the statistical analysis of morphometric attributes (see Table 2 for examples). More recently, general geomorphometric studies have made wider use of morphometric attributes and their statistical analyses and feature-based quantitative representation, most of which were specifically developed for submarine landscapes. Micallef et al. (2007a), for example, developed a methodology for the quantitative analysis of seafloor data, which was shown to exploit the full potential of these data sets and significantly improve the mapping and characterisation of submarine landslides. This methodology was applied to the submarine mass movements offshore Norway to elucidate the evolution dynamics of a multi-phase submarine landslide (Fig. 7), while emphasising the potential role of gas hydrate dissociation and contourite deposition in controlling the location and extent of submarine slope failure (Micallef et al., 2009), and to improve understanding of the mechanics and triggers of spreading, also while using limit-equilibrium and mechanical modelling (Micallef et al., 2007b). The automated and objective mapping of submarine landscapes is indeed an important application of general geomorphometry, and specific techniques have been developed for the characterisation of pockmarks (Harrison et al., 2011; Gafeira et al., 2012), terraces (Passaro et al., 2011) and canyons (Ismail et al., 2015). Others have used general geomorphometric techniques to classify submarine landscapes (e.g. fjords (Moskalik et al., 2014a; Moskalik et al., 2014b), continental shelf and slope (Elvenes, 2013), and global (Harris et al., 2014)), identify the various styles and scales of deformation across submarine landslides (Mountjoy et al., 2009), and infer the evolution of seamounts (Passaro et al., 2010), mid-ocean ridge scarps (Mitchell et al., 2000) and faults in active continental margins (Kukowski et al., 2008).

## 6.2 Marine habitat mapping, biogeography and ecology

Benthic habitat mapping is one of the major applications areas where the use of marine geomorphometry has grown in recent years. Linked to the rise in the use of multibeam data for benthic habitat mapping (Brown et al., 2011; Smith and McConnaughey, 2016) the vast majority of habitat mapping studies with access to good bathymetry data are now using, or at least testing, some form of terrain attribute or feature classification in their habitat mapping activities, even though we note that many of these are not yet reflected in the peer-reviewed literature. Among the habitat mapping community several approaches to habitat mapping are common, many of which directly incorporate biological data, such as modelling species (e.g. Davies et al., 2008) or biotope distributions (e.g. Elvenes et al., 2014) and others which are primarily based on physical attributes deemed relevant for the distribution of benthic fauna (e.g. Micallef et al., 2012; Ismail et al., 2015). Geomorphometry is equally useful for both these approaches and those that combine both aspects (e.g. Tempera et al., 2012)

and this discussion is relevant to an all-encompassing definition of habitat mapping (Lecours et al., 2015b). Figure 6 illustrates how terrain attributes are typically used in the production of predictive seabed habitat maps, providing an invaluable suite of full coverage predictor variables which are used together with point samples of observed habitat as the input data to modelling.

Harris and Baker (2012b) provide a summary of surrogate variables used for habitat mapping studies, including many terrain attributes that have been applied across a multitude of approaches to habitat mapping worldwide. The issue of surrogacy is also discussed in this volume as well as by Lecours et al. (2015b) and McArthur et al. (2010). The case studies presented in the GeoHAB Atlas, and other published studies, vary in the degree to which they have established the ecological relevance of the terrain attributes and/or feature classifications used. For geomorphological variables to really be useful predictors of
seafloor habitat, the relationship between habitat and specific variables first needs to be established. Apart from depth, which all of the geomorphological variables are derived from, different shapes or attributes of the seafloor will be relevant to different species at different scales over different bathymetric and biogeographic zones. Bathymetry is known to have a first order influence on species distribution, largely because many properties that directly affect benthic habitat vary with depth (e.g. light, temperature). A number of recent papers describe the potential of terrain attributes to act as surrogates of species
distribution (Lucieer et al., 2013; Hill et al., 2014). The relationships are validated using several different statistical methods that either test terrain attributes against biological or ecological data, or combine terrain attributes with other environmental data and perform classifications to differentiate between the different habitats (Thiers et al., 2014).

In an example by Rengstorf et al. (2013), habitat suitability models for the cold-water coral *Lophelia pertusa* were developed based on full coverage multibeam bathymetry on the Irish continental margin. Maximum entropy modelling was used to
predict *L. pertusa* reef distribution at a spatial resolution of 0.002° (250 m). Coral occurrences were assembled from public databases, publications and video footage, and filtered for quality. Environmental predictor variables were produced by re-sampling of global oceanographic data sets and a regional ocean circulation model. Multi-scale terrain parameters were computed from multibeam bathymetry at 50 m resolution. In a related study, Rengstorf et al. (2012) examined the effect of bathymetric data resolution on terrain attributes used to predict coral distribution, resampling the original 50 m resolution
bathymetry from the Irish National Seabed Survey at successively coarser intervals up to 1 km. They concluded that terrain attributes derived from higher resolution bathymetry are required to adequately detect the topographic features relevant to corals. In a further related study, Rengstorf et al. (2014) examined the relative importance of terrain attributes and hydrodynamic variables (e.g. current speed, vertical flow, temperature) on models of cold-water coral distribution, concluding that combining the environmental information from these two sources leads to improved predictions over the
spatial scales in question.

At a much finer resolution (~1 m) species−habitat relationships were examined across a marine reserve on the south-eastern coast of Tasmania using boosted regression tree analyses (Cameron et al., 2014). The most important explanatory variables

of community diversity were those describing the degree of reef aspect deviation from east and south (seemingly as a proxy for swell exposure), reef bathymetry (depth), low rugosity and slope. These models could account for up to 30% of the spatial variability in measures of species diversity. As biological data at scales relevant to acoustic or remote sensing data such as that from AUVs, ROVs and diver surveys become available on national or international databases such as the Census for Marine Life and the Ocean Biogeographic Information System (OBIS) the ability to extend species distribution models into the wider ocean at finer scales will enhance the utility and value of marine geomorphology variables for marine biodiversity assessment.

## 6.3 Hydrodynamics and modeling

The interaction of bottom currents with seafloor sediments results in a wide range of erosional and depositional morphologies – e.g. scours, furrows, ripples, dunes, lineations, contouritic drifts – the morphology and dimensions of which depend on flow velocity and sediment grain size (Stow et al., 2009). Detecting change in bedform morphology is of great interest to geologists, physical oceanographers and climatologists, and many others with the applied interest in such features. Bedforms determine basic flow patterns of ocean circulation at coarse and fine scales; even small perturbations in seafloor topography can influence the pathway and velocity of major shallow and deep current flows, heat transport and ultimately climate (Gille et al., 2004; Kunze and Llewellyn Smith, 2004; Metzger and Hurlburt, 2001; Palomino et al., 2011). In turn, bedforms are also excellent archives of current and past bottom flow patterns (Sandwell et al., 2002). Port managers are also interested in bedforms and their evolution, particularly where they constitute a hazard to navigation in coastal waters. Detecting change in bathymetry and its impact on oceanography is therefore important, and local geomorphometric attributes, such as aspect, curvature and rugosity, have been used to develop hydrodynamic models or as proxies for local and regional currents (Lecours et al., 2015a). Seafloor topographic proxies are also fundamental in the predictive mapping of suspension-feeders (Lucieer et al., 2013; Hill et al., 2014), such as cold-water corals (Rengstorf et al., 2012; Tong et al., 2013), because their distribution is inextricably linked to current flow strengths and patterns (Mohn et al., 2014). More recently, understanding the link between seafloor morphometry and currents has been shown to be essential in forecasting the path of floating debris from tsunamis and air disasters and assist in search and rescue operations (Mofield et al., 2004; Normile, 2014; Smith and Marks, 2014).

## 6.4 Emerging and future applications

Other applications of marine geomorphometry can also be found in the literature. We anticipate that the number of application areas will grow substantially over the next few years as awareness of data and analysis techniques expands, and high-resolution data become more widely available.

## 6.4.1 Change detection

A number of studies have described temporal morphological dynamics of the seafloor using acoustic bathymetry (e.g. Duffy & Hughes-Clarke, 2005; Smith et al., 2007). However, assessments of biological change beyond the range of optical sensors have been based primarily on ground sampling methods. Rattray et al. (2013) investigated approaches to quantify temporal

change in benthic habitats from a spatially explicit perspective using acoustic techniques. Their methods (1) quantified change in terms of gains and losses in the extent of habitat at a site on the temperate southeast Australian continental shelf; (2) they could distinguish between systematic and random patterns of habitat change; and (3) were able to assess the applicability of supervised acoustic remote sensing methods for broad-scale habitat change assessment. Change detection in temperate bedrock reefs were identified through morphological characterisation by Storlazzi et al. (2013). They delineated

the classes using a multivariate classification routine (Dartnell and Gardner, 1999) based on acoustic backscatter and rugosity (surface-planar area ratio).

There have also been several examples in the literature of repeat multibeam surveys being used to detect change, many of which are summarised by Schimel et al. (2015). Analysis is generally focussed on differences in depth values detected and often aided by a visual assessment of the changes in morphology. There are fewer studies that have explicitly used terrain

attributes or features in their assessments, but we recognise the potential for gemorphometric techniques to be more widely applied in this type of study. For example, Bøe et al. (2015) used geomorphic feature detection (Wood, 1996) to identify crests and ridges in a sandwave field on the continental slope, and assess movement between surveys based on the change in position of these features. We note also that Schimel et al. (2015) incorporate measures of bathymetric uncertainty in their assessment of volume change and also recommend guidelines on thresholds which can help to improve the confidence of

such assessments.

## 6.4.2 Seismic geomorphometry

Seismic geomorphology is a rapidly evolving discipline. It comprises the application of geomorphological principles and analytical techniques to study palaeo-landscapes as imaged by 3D seismic reflection data (Carter, 2003; Posamentier and Kolla, 2003; Posamentier, 2003). More recently, 3D seismic reflection data have also provided a good alternative source of

bathymetric data when the latter are absent (e.g. broad scale geomorphic mapping in the MAREANO project – http://www.mareano.no/). The development of seismic geomorphometry is a natural consequence of increasing computer power, which enables the rapid manipulation, visualisation and interpretation of 3D seismic reflection data, and the enormous investment in this technology by the oil and gas industry, with academics and government researchers benefitting from access to these data. The integration of seismic geomorphology with seismic stratigraphy currently represents the state-

of-the-art approach to extracting geological information from 3D seismic reflection data to understand large-scale basin evolution. Seismic geomorphological studies have addressed a broad range of geological problems, ranging from

sedimentary to igneous geology, from lithology distribution to large-scale tectonic analysis (e.g. Fachmi and Wood, 2003; Miall, 2003; Wood, 2003).

Up to the present, most studies have focused on the qualitative recognition of broad scale features (e.g. Posamentier et al., 1996; Posamentier et al., 2000; Peyton and Boettcher, 2000; Posamentier, 2003; Zeng and Hentz, 2004). Quantitative
seismic geomorphology, or seismic geomorphometry, is the most recent development of seismic geomorphology (Carter, 2003; Posamentier, 2003; Posamentier and Kolla, 2003). Seismic geomorphometry has been defined as the "*quantitative analysis of the landforms, imaged in 3D seismic data, for the purposes of understanding the history, processes and fill architecture of a basin*" (Wood, 2003). Seismic geomorphometry encompasses techniques that use 3D seismic data to investigate the nature and architecture of reservoirs through extraction and analysis of quantitative morphometric
information.

Great opportunities exist for applying a more quantitative approach in seismic geomorphology. Seismic geomorphometric techniques provide statistical and mathematical insight into the morphological and dimensional characteristics of geologic systems that are difficult to derive through qualitative investigations of outcrop exposures and 2D seismic reflection data. Seismic geomorphometric studies provide a deep and spatially extensive understanding of how morphology develops
through time, providing insight into the historical evolution of a basin and the possibility of developing predictive models. Quantitative relationships derived from seismic geomorphological studies can decrease our uncertainty in predicting the nature and location of reservoirs in deep-water settings by testing cause-and-effect relationships in a variety of settings. Computer-assisted seismic geomorphometry, in particular planform pattern recognition, is a powerful addition to the seismic geomorphological approach. It allows the interpreter to identify geologically significant features in plan view automatically.
The ability to exploit the full potential of large seismic data sets is currently hindered by the lack of tools in existing software packages, coupled with the limited knowledge of how morphometrics can be used in the analytical process. It is the development of such tools that should be a main focus for researchers of marine geomorphometry in the near future.

## 6.4.3 Broad-scale coastal geomorphometry

This paper has shown how geomorphometric techniques developed mostly in terrestrial settings can be applied to the marine
environment or adapted to enable quantification of the seafloor terrain. For a long time however, the boundary between the land and the sea was not easily mapped or delineated and represented a challenge for both marine and terrestrial scientists (Klemas, 2011a). This was due to the inability of satellite remote sensing to collect data in deep waters and the limitations of acoustic systems to collect data in shallow waters, which often creates a gap in terrain data where land meets sea. This littoral gap, sometimes referred to as 'the white zone' because the lack of data in this area between available DBM and DEM
data appear white on maps, often complicates the study of nearshore environments and can have important implications for applications such as navigation and geohazard assessment. For instance, in their attempt to assess the effectiveness of a

marine protected area in a Canadian sub-arctic fjord with habitat maps generated from a combination of terrain attributes and other data, Copeland et al. (2013) were only able to map 32 km$^2$ of the total 82 km$^2$ of the area. They highlighted the laborious nature of shallow water survey (i.e. time and cost-consuming MBES surveys), the need for a continuous coverage because of the large littoral gap, and indicated that interpolation and extrapolation of results in the littoral gap were

inappropriate because of the heterogeneous nature of coastal fjord environments (Copeland et al., 2013).

In the last 15 years, developments in bathymetric LiDAR surveying (cf. Sect. 2.4) slowly helped fill the littoral gap. Consequently, efforts to map the littoral using bathymetric LiDAR have spread across the globe (e.g. the National Coastal Mapping Program of the Joint Airborne Lidar Bathymetry Technical Center of Expertise in the United States), and several examples of investigations of the coastal environment using geomorphometry can now be found in the literature (e.g. Purkis

et al., 2008; Pittman et al., 2009). This body of literature used to be mostly characterized by the study of small areas either above the water (e.g. dunes or emergent features) or submerged (e.g. coral reefs) (Brock and Purkis, 2009), but there are now more and more efforts targeting the collection of topo-bathymetric data that span the coastal environment (e.g. Dunkin et al., 2011; Dunkin and McCormick, 2011). Several fields can benefit from seamless coastal geomorphometric analysis. For instance, inter-tidal rocky shores are known to shelter a lot of biodiversity (Kostylev et al., 2005) and linking quantitative

terrain attributes to measures of biodiversity could improve scientific understanding of ecological patterns and processes in these important areas of the land-sea boundary (e.g. Collin et al., 2012). So far, limitations of LiDAR systems (e.g. inability to collect data in deeper waters, costs associated with airborne surveys) however restricted these efforts to local, sometimes regional scales. To our knowledge, there are no geomorphometric applications that span the terrestrial and underwater landscapes in a continuous way over very large areas, which would require the integration or fusion of datasets from

different sources (e.g. LiDAR, terrestrial DTMs and acoustic surveys).

At a broader scale, a seamless analysis of terrestrial and marine environments requires the combination of terrestrial DTMs, bathymetric data from acoustic systems, and bathymetric LiDAR or optical remotely sensed data to fill the littoral gap and create what has been called in the literature a Coastal Terrain Model (CTM) (Hogrefe et al., 2008; Leon et al., 2013). The challenges encountered with merging datasets from different sources makes such an approach still nascent in the general

literature (Macon et al., 2008; Quadros et al., 2008; Collin et al., 2012), and very rare, if not fully absent, in the marine geomorphometry literature. Data fusion is the process of acquiring, processing and synergistically combining multi-source datasets both geometrically (i.e. in space) and topologically (i.e. in terms of their attributes or information content) (Usery et al., 1995; Samadzadeghan, 2004; Mohammadi et al., 2011). Despite constant developments in data fusion (Pohl and van Genderen, 1998; Dong et al., 2009; Zhang, 2010), it presents particular challenges for geomorphometry. First, despite

improvements in edge matching algorithms, artefacts from merging and surveying can appear when deriving terrain attributes from the fused dataset (Stoker et al., 2009). Data fusion often requires the different datasets to overlap slightly in order to be combined. In theory, the overlapping areas should yield very similar values, within their uncertainty and error ranges. However, important inconsistencies (up to 6.5m) have been reported between depth measurements of the same areas

using bathymetric LiDAR and MBES (Quadros et al., 2008; Costa et al., 2009; Chust et al., 2010; Shih et al., 2014). This has implications for geomorphometry since terrain attributes will capture and classify these mismatches as features, especially as the differences usually occur locally (Chust et al. 2010). Also, coastal environments can be very dynamic; artefacts could appear in the DTM if the multi-source data are not collected at the same time and changes occurred between the data collections. Another issue concerns vertical datums (Hogrefe et al., 2008); terrestrial surveys are usually referenced to a local geoid model based on the GPS, while underwater acoustic surveys are usually referenced to the mean sea level at the time of survey, which is referenced to a local or regional tidal gauge that is itself referenced to a local datum. Calls for a consistent and unified vertical datum have been made but this issue is still unresolved (Hogrefe et al., 2008; Quadros et al., 2008). Finally, data quality and uncertainty may complicate the fusion of the different datasets. For instance, the inability of bathymetric LiDAR systems to collect reliable data in turbid or cloudy waters and in breaking wave conditions, in addition to their difficulties to distinguish the seafloor from the water surface in waters shallower than 30 cm (Quadros et al., 2008) may create a smaller littoral gap called the "dead zone" (Nayegandhi et al., 2009) and prevent proper fusion. Bernstein et al. (2011) recommend a customized survey design to minimise the challenges associated with creating a seamless DTM.

Regular problems of data fusion, for instance related to merging multi-resolution datasets or to software and format compatibility/interoperability, also apply to the development of DTMs for broader-scale coastal geomorphometry. Terrestrial terrain models may have a Digital Elevation Model (.dem) format, while bathymetric LiDAR data can be recorded with a Laser File (.las) format and finally, acoustic data can be save as a Bathymetric Attributed Grid (.bag) format; all these file formats have different structure and characteristics. Impediments to the fusion of multisensor data to build seamless elevation and depth surfaces include, but are not limited to, inconsistent spatial and temporal scales, incompatible formats, and differences in levels of reliability, uncertainty and completeness. Despite these impediments, data fusion has been identified as a promising technique for geomorphometry (Bishop et al., 2012).

Some authors (e.g. Quadros et al., 2008) argue that the different types of datasets cannot be readily integrated, but the main challenges will likely be addressed with improvements in data fusion techniques and ease of implementation of these techniques for non-expert users (Zhang, 2010) for geomorphometry. Current work includes detection and correction of differences in geoid models, consideration of uncertainties, and improvement in edge matching algorithms (Quadros et al., 2008; Dong et al., 2009; Stoker et al., 2009). Recently, Leitão et al. (2016) proposed a new method to merge different DTMs developed specifically for geomorphometry. Future developments in data fusion will likely allow better integration of different data to create seamless coverage for complete geomorphometric analysis and identification of broad-scale overlapping landforms between the different realms. This will be useful for a wide range of coastal applications. For instance, observations of underwater and terrestrial landforms have shed light on how glaciers retreated in Atlantic Canada during the last deglaciation (Shaw et al., 2006); the investigation of landforms that overlap both realms could help refine this type of analysis. Other potential applications include the investigation of coastal morphodynamics and land-sea exchange modelling, dredging, the identification of hazards due to sea-level rise and severe storms, the assessment of consequences of

such storm events, monitoring and shore protection, coastal archaeology, resource management and marine spatial planning, anthropogenic sensitivity and environmental status assessment, and other scientific research.

### 6.4.4 Underwater archaeology

In the last 25 years, terrestrial archaeology has largely benefitted from remote sensing tools and methods (McCoy and
Ladefoged, 2009). Radar and LiDAR data have helped reveal archaeological features of interest in many areas of the world (e.g. Meylemans et al., 2015), or detect anomalies that could be linked to sites of interests that cannot be seen from the ground (e.g. Lin et al., 2011). Geomorphometry has been used on radar and LiDAR data to identify such patterns (Kvamme, 1999) or to describe particular areas (Turrero et al., 2013). Similarly to what happens in marine geomorphometry, its use is often not being recognised as geomorphometry or terrain analysis.

The remote sensing techniques described in Sect. 2 have also been used in underwater archaeology to collect bathymetric and backscatter data that were used for instance in initial investigations of wreck sites location and extent before divers or ROVs deepen the actual archaeological investigation (e.g. Jones et al., 2005; Masetti and Calder, 2012). Similarly to terrestrial archaeology, these types of data enable both the direct identification of the features on the seafloor or anomalies that may indicate potentially buried artefacts (Papatheodorou et al., 2005). Geomorphometry has yet to gain traction in
underwater archaeology, but is not completely absent. Using MBES data, Stieglitz (2012) documented an area of seafloor off Australia that had a conspicuous arrangement of over 1,200 shallow holes, and wide (up to 10 m) and deep (up to 1.5 m) holes. They classified these holes using local slope measurements, and found that the systematic distribution of these seafloor features was related to their distance from a shipwreck and likely caused by bioturbation. In another application, Passaro et al. (2013) extracted archaeological features related to Italian sunken cities using curvatures and the r.param.scale
command from GRASS GIS (cf. Sect. 5.1). Slopes values were used by Solsten and Aitken (2006) to assess the risk of disturbance of archaeological sites by mass movement and marine flooding in Nunavut, Canada. The application of techniques from geomorphometry to underwater archaeology is likely to increase in the future, and we note the potential of seismic geomorphometry to assist in the investigation of buried artefacts.

# 7 The future of marine geomorphometry

## 7.1 Current and future trends in marine geomorphometry

Current developments in the marine geomorphometry literature are primarily focussed on the data acquisition end of the workflow. Technology and equipment for surveying the seafloor are improving in quality, accuracy and cost-effectiveness, which will allow an increase in data availability and quality. In coastal environments, ongoing research is focused on improving the extraction of depth information in the littoral gap in order to create seamless DTMs from the seafloor to land,

and developments in data fusion should soon enable broad-scale geomorphometric analyses of coastal environments. As the pressure on coastal environments increases, such information will become crucial for many applications. From an ecosystem point of view, coastal environments are also very rich in biodiversity. Studies of the topographic structure that can be identified from CTMs using geomorphometric techniques are likely to facilitate a better scientific understanding of these ecosystems. In the deep-sea, extensive use of AUV and ROV-based MBES and other technologies means we are now able to collect high-resolution bathymetric data of environments never explored before at such level of detail. The knowledge that has been gained from using these data in combination with different techniques, including geomorphometry, has revolutionized scientific understanding of many marine environments. It was initially thought that the deep-sea was mostly flat, muddy and lifeless, but the last twenty years of research have proven otherwise. Nevertheless, exploration is far from complete; there are still wide gaps in the scientific knowledge of deep-sea patterns, processes and ecosystems. High quality bathymetric data is fundamental to the success of revealing this knowledge and its limited availability is currently a barrier to effective protection and management of vulnerable species (Vierod et al., 2014; Ross et al., 2015).

As the marine geomorphometry community moves forward, it will rapidly need to start addressing issues other than those associated with data acquisition. The availability of tools that streamline the workflow from data collection to analysis will be key in making a more complete science of marine geomorphometry accessible to marine scientists with a wide range of background and experience. Repositories of comprehensive and freely available datasets and tools, such as Digital Coast that provide free coastal and marine bathymetric data and analytical tools (NOAA, 2016), are the way forward to improve accessibility to the wider scientific community, and this may well mean that bathymetric data gain the attention of those currently engaged in developing geomorphometric methods for terrestrial data. We also acknowledge that easily accessible GIS tools and readily available data can also bring hidden dangers from non-critical use by users with limited appreciation of data collection and processing methods which to the expert clearly reflect the limitations in the utility of particular bathymetric datasets. To prevent this danger of inappropriate use, tools and datasets need to be accompanied by complete metadata that include information on data provenance, survey, scale, error and uncertainty quantification, and any other information relevant to further use of the tools and datasets. Metadata are crucial to create a "quality-aware" community (Devillers et al., 2007; Lecours et al., 2015b). The use of the CUBE algorithm to create BASE surfaces is one way to carry over a measure of quantified uncertainty of the data, but such information is not readily available for the majority of publicly available datasets. This type of information needs to become more accessible to marine scientists with a broad range of scientific backgrounds.

It is becoming critical to raise awareness of geomorphometry in the marine science community. Methods from specific geomorphometry demonstrate a lot of potential for marine applications and should be used more extensively. At the same time, it is opportune to improve practices by setting standards and protocols for the application of geomorphometry. Methods and interpretations need to be standardised, particularly in view of issues specific to the marine environment, or where data and analyses behave differently underwater than on land. Amongst these, the influence of scale and data resolution on the

results, and the consideration of spatial uncertainty should be prioritised. End-users should be explicit about which algorithms or methods they use and at which scale in order to enable proper comparison among studies. Since geomorphometric analyses are more and more performed within GIS environments, devising a GIS-based standard methodology and symbology for marine geomorphological mapping using geomorphometry would be a very useful goal for the marine geomorphometry community. Ultimately, the type of standards and protocols a marine geomorphometry community could develop should encourage wider applications of bathymetric data and allow marine scientists optimise the use of their expensive datasets.

## 7.2 Reuniting efforts in geomorphometry

This manuscript has discussed the current practices in marine geomorphometry, from data collection to the applications. Through all aspects of this discussion it is apparent that the use of modern geomorphometric techniques in the marine realm is relatively nascent, having begun only over a decade ago in application areas outside of marine geomorphology. The dramatic increase in DBM availability, combined with the increasingly accessible and user-friendly GIS tools, is currently fuelling the amount and diversity of applications of marine geomorphometry. However, this availability can become a double-edge sword. As noted by Dolan and Lucieer (2014), "*Although a [DBM] is a model of the seabed surface, it is often not treated as a model but rather is accepted as a true representation of the seabed*". Furthermore, as highlighted in Fig. 2, the end-users of geomorphometric techniques are not always aware that they are actually "doing" geomorphometry, but rather think of the steps they are performing as simply using GIS tools for data analysis. As the amount of applications increases, some of the fundamental issues associated with marine geomorphometry are not being addressed quickly or broadly enough. This can increase the risk of unsuspecting end-users misusing data or techniques, and to the misinterpretation of results. For instance, due to a lack of awareness of the impact of artefacts in DBMs and their propagation to terrain attributes, artefacts are often disregarded, or assumed to be obvious. In habitat mapping, the consequences of artefacts are often apparent to geomorphometry-aware users in the final maps (e.g. Zieger et al., 2009; Lucieer et al., 2012), but this can become problematic if the maps are being used in conservation and management decision making if the effect of the artefacts is not appreciated by the end user. It is thus crucial for end-users, planners, managers and decision-makers to become aware and understand the properties of their data that result from each of the five steps of geomorphometry, and how these properties influence their particular application.

In addition to end-users not being aware of geomorphometry as a science, scientists engaged in more terrestrial and extra-terrestrial geomorphometry are rarely aware of marine geomorphometry, its differences and its similarities with their field of expertise. For example, at the turn of the millennium, Pike (2000) identified the study of seafloor abyssal hills as a prospect topic for the application of geomorphometry. However, many examples can be found of marine geophysics and geomorphology studies dating from the 1980s-1990s that have used geomorphometry in abyssal hills (e.g. Malinverno and Gilbert, 1989; Goff, 1991, 1992; Malinverno and Cowie, 1993; Shaw and Lin, 1993). A decade later, Pike et al. (2009) tried

to suggest using Digital Depth Models (DDM) to characterize surface models of the seafloor, despite the wide acceptance of DBM as an appropriate term in the marine geomorphometry literature.

We recognise a critical need for a dedicated scientific effort in marine geomorphometry that will address, and raise awareness of the fundamental issues related to marine geomorphometry. This effort does not necessarily have to come solely from the marine science community, indeed it may well benefit from the expertise of many of those scientists already engaged in terrestrial geomorphometry. The main objectives of this effort would be to learn from the lessons of terrestrial geomorphometry, ensure that studies of geomorphometry become more widespread in the marine literature, and respond to the challenges and opportunities for a wider adoption of marine geomorphometry as a key tool in marine sciences, whilst improving and upholding scientific standards. Since sub-fields of geomorphometry dealing with different types of environments are ultimately parts of the same science and share more similarities than have differences, these standards should become common to all these sub-fields. For example, geomorphometry is a field recognized for its ambiguous terminology, particularly in terms of terrain attributes (Bishop et al., 2012). The field of geomorphometry should move towards a more uniting terminology and vocabulary across environments that would reduce some of that ambiguity. For instance, the use of the terms DEM, DBM, CTM, and DDM should be abandoned in favour of the more neutral, all-encompassing term DTM. Moving towards a joint terminology is just an example of how we can reunite all sub-fields of geomorphometry together, with common goals and approaches. For instance, many of the issues and future challenges mentioned in this overview (e.g. uncertainty and error propagation and modelling, scale, change detection) have been discussed in recent reviews about terrestrial high-resolution topographic data and Earth surface processes (Tarolli, 2014; Passalacqua et al., 2015), highlighting the similarities in challenges and opportunities that marine and terrestrial geomorphometry are facing. Reuniting efforts in geomorphometry will likely result in more effective research and development and facilitate the coupling with other disciplines, including different fields of marine sciences.

# 8 Conclusions

Relative to the "young" and "still forming" modern terrestrial geomorphometry (Evan and Minár, 2011, p. 105), the use of geomorphometry in the marine realm is still in its infancy. Ever since the first coarse-scale DBMs were generated, marine geomorphometry has helped improve scientific understanding of the oceans, from the relatively thin border where land meets sea to the deepest waters. This paper is timely because it provides an overview of the state of the art in the field and discusses standards for the applications of marine geomorphometry. By following Pike et al.'s (2009) five main steps of geomorphometry we have reviewed marine geomorphometry in a way that can easily be compared with terrestrial geomorphometry. We have provided an overview of the different methods to sample the depth of the seafloor, the interpolation methods and issues of spatial scale associated with the generation of a DBM, as well as discussing the different errors and artefacts that are characteristic of DBMs but different from those common in DEMs. Further, we have discussed

how general and specific geomorphometry are applied underwater, provided applications of marine geomorphometry, and outlined future trends in the field. Clearly there is room in the literature for more detailed reviews of each of these five steps and relating to many of the sub-disciplines, however we hope that this review will serve as a solid foundation for further, more detailed reviews on these sub-topics.

Based on this review, we provide the following recommendations that should help establish more productive practices in marine geomorphometry: (1) errors and spatial uncertainty should be quantified so that they are able to be considered in the geomorphometric analyses and in the interpretation of results; (2) metadata should consistently be associated with datasets to explicitly indicate data provenance, quality (i.e. quantification of uncertainty), and the spatial scale at which the dataset was intended to be used; (3) data, metadata and tools should be made available for a wider applications of bathymetric data; (4)

standardisation of methods and interpretations for each field of application should be documented, particularly in view of the influence of algorithms, scale and data resolution on the results; and (5) a GIS-based standard symbology for marine geomorphological mapping based on geomorphometry should be devised.

Through raised awareness of each other's disciplines, we hope that both marine scientists and geomorphometry practitioners will be better placed to work together in addressing the fundamental issues of marine geomorphometry, whilst upholding

scientific standards in marine spatial analysis. Building a dedicated effort in marine geomorphometry that can draw on lessons learned in terrestrial geomorphometry will not only encourage marine applications and continued scientific development, but will ensure that the science of geomorphometry is used to its full potential for studying the topography of the whole planet.

# Acknowledgements

V. Lecours thanks Emma LeClerc for her valuable comments on sections of this manuscript, Dr. Rodolphe Devillers for the insightful discussions about marine geomorphometry, and the Natural Sciences and Engineering Research Council (NSERC) of Canada for providing funding. A. Micallef is funded by a Marie Curie Career Integration Grant PCIG13-GA-2013-618149.

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

# Tables

Table 1: Summary of the most commonly used terrain attributes in marine-based studies, as well as an indication of some common calculation approaches. Modified after Dolan et al. (2012). The term 'multiple scale' refers to terrain attributes derived in turn using analysis windows of different sizes. 'Multiscale' refers to indices derived simultaneously over a range of window sizes. The more general term 'multi-scale' is used in this paper to refer to both types of analysis as well as geomorphometric analysis using data of different resolutions.

|  | *Slope* | *Orientation* | *Curvature* | *Terrain Variability* |
|---|---|---|---|---|
| *Ecological relevance* | Stability of sediments (ability to live in/on sediments)<br><br>Local acceleration of currents (food supply, exposure, etc.) | Degree of exposure to dominant and/or local currents from a particular direction (food supply, sedimentation, larval dispersion, etc.) | Index of exposure/shelter e.g. on a peak or in a hollow (food supply, sedimentation, predators, etc.) | Index of degree of habitat structure, shelter from exposure/predators (link to life stages)<br><br>Structural diversity linked to biodiversity |
| *Geomorphological relevance* | Stability of sediments (grain size).<br><br>Local acceleration of currents (erosion, movement of sediments, creation of bedforms) | Relation to direction of dominant geomorphic processes | Flow, channelling of sediments/currents, hydrological and glacial processes.<br><br>Useful in the classification of landforms | Terrain variability and structures present reflect dominant geomorphic processes |
| *Commonly computed terrain attribute and example marine-based reference* | Slope (Lundblad et al., 2006; Lanier et al., 2007; Micallef et al., 2012; Dolan and Lucieer, 2014) | Aspect (Galparsoro et al., 2009), northness/northerness and eastness/easterness (Monk et al., 2011) | Mean curvature (Dolan et al., 2008)<br><br>Profile curvature (Guinan et al., 2009)<br><br>Plan/planimetric curvature (Ross et al., 2015)<br><br>Bathymetric Position Index (BPI) (Monk et al., 2010; Pirtle et al., 2015) | Rugosity (Dunn and Halpin, 2009)<br><br>Vector Ruggedness Measure (VRM) (Tempera et al., 2012)<br><br>Relative Relief (Elvenes, 2013)<br><br>Fractal Dimension (Wilson et al., 2007) |
|  |  |  |  |  |

| *Commonly used terrain attribute and software (algorithm reference)* | **Single scale:**<br><br>Slope: ArcGIS Spatial Analyst (Horn, 1981) | **Single scale:**<br><br>Aspect: ArcGIS Spatial Analyst (Horn, 1981)<br><br>Statistical Aspect (northness/eastness) BTM toolbox (Wright et al., 2012) | **Single scale:**<br><br>Mean, Profile and Plan curvature ArcGIS Spatial Analyst (Zevenbergen and Thorne, 1987) | **Single scale:**<br><br>Rugosity (surface area/planar area ratio) (Jenness, 2004) |
|---|---|---|---|---|
| | **Multi-scale:**<br><br>Multiple scale slope: r.param.scale, Landserf (Wood, 1996)<br><br>Multiscale slope: Landserf (Wood, 1996) | **Multi-scale:**<br><br>Multiple scale aspect: r.param.scale, Landserf (Wood, 1996)<br><br>Multiscale aspect: Landserf (Wood, 1996) | **Multi-scale:**<br><br>Several measures of multiple scale curvature: r.param.scale, Landserf (Wood, 1996)<br><br>Multiple scale BPI (Lundblad et al., 2006)<br><br>Multiscale curvature Landserf (Wood, 1996) | **Multi-scale:**<br><br>Multiple scale VRM (Sappington et al, 2007)<br><br>Multiple scale relative relief (Erikstad et al., 2013 and references therein)<br><br>Multiple and multiscale fractal dimension - Landserf (Wood, 1996) |

Table 2: Selection of studies published in the last three decades that applied geomorphometric techniques to the marine environment. A particular focus is given to marine geomorphology studies, as a few other documents (e.g. McArthur et al., 2010; Brown et al., 2011; Harris and Baker, 2012; Rengstorf et al., 2012; Lecours et al., 2015b) already summarized the extent to which many of these techniques have been employed in habitat mapping studies, and many of these techniques have yet to be employed in other contexts.

| Technique | Reference | Spatial Domain | Broad Theme |
|---|---|---|---|
| **General geomorphometry** | | | |
| Morphometric attributes Basic geometrical analysis | (Adams and Schlager, 2000) | Continental slope | Geomorphology and geohazards |
| | (De Moustier and Matsumoto, 1993) | General | |
| | (Teide Group, 1997) | Volcanic islands | |
| | (Coggan and Diesing, 2012) | Continental shelf | Habitat Mapping and ecology |
| | (Ezhova et al., 2012) | Coastal and continental shelf | |
| | (Mofield et al., 2004) | Continental slope and rise, abyssal hills | Hydrodynamics |
| | (Passaro et al., 2013) | Coastal | Others |
| Morphometric attributes and their statistical analyses | (Berkson and Matthews, 1984) | General | Geomorphology and geohazards |
| | (Booth and O'Leary, 1991) | Continental slope and upper rise | |
| | (Chakraborty et al., 2001) | Mid-ocean ridge, abyssal plain, seamounts | |
| | (Goff and Jordan, 1988) | General | |
| | (Kukowski et al., 2008) | Continental slope | |
| | (Micallef et al., 2007a) | Continental slope | |
| | (Mitchell et al., 2000) | Mid-ocean ridge | |
| | (Moskalik et al., 2014a) | Coastal and inner shelf | |
| | (Passaro et al., 2010) | Seamount | |
| | (Passaro et al., 2011) | Seamount, volcanic island | |
| | (Smith and Shaw, 1989) | Abyssal hills | |
| | (Lucieer et al., 2013) | Coastal to inner shelf | Habitat Mapping and ecology |
| | (Hill et al., 2014) | Coastal to inner shelf | |
| | (Rengstorf et al., 2012) | Continental slope | |
| | (Tong et al., 2013) | Outer shelf | |
| | (Micallef et al., 2012) | Coastal to inner shelf | |
| | (Tempera et al., 2012) | Seamounts | |
| | (Rengstorf et al., 2013) | Shelf to abyssal plain | |
| | (Rengstorf et al., 2014) | Continental slope | |
| | (Mohn et al., 2014) | Continental slope | Hydrodynamics |
| | (Tong et al., 2013) | Outer shelf | |
| | (Solsten and Aitken, 2006) | Coastal | Others |
| | (Stieglitz, 2012) | Coastal | |
| Spectral analysis | (Fox and Hayes, 1985) | General | Geomorphology and geohazards |
| | (Fox, 1996) | Mid-ocean ridge | |
| | (Gilbert and Malinverno, 1988) | General | |
| | (Goff and Tucholke, 1997) | Mid-ocean ridge | |
| | (Moskalik et al., 2014b) | Coastal and inner shelf | |
| Geostatistical methods | (Herzfeld, 1989) | Continental slope | Geomorphology and geohazards |
| | (Herzfeld and Higginson, 1996) | Mid-ocean ridge | |
| | (Ismail et al., 2015) | Continental slope | |

| | | | |
|---|---|---|---|
| | (Diesing et al., 2014) | Coastal to inner shelf | Habitat Mapping and ecology |
| | (Hillman et al., 2015) | Continental slope | Hydrodynamics |
| Feature-based quantitative representation | (Harrison et al., 2011) | Outer shelf | Geomorphology and geohazards |
| | (Micallef et al., 2007a) | Continental slope | |
| | (Micallef et al., 2007b) | Continental slope | |
| | (Mitchell and Clarke, 1994) | Continental shelf | |
| | (Pratson and Ryan, 1996) | Continental slope | |
| | (Calvert et al., 2015) | Coastal | Habitat Mapping and ecology |
| | (Bøe et al., 2015) | Continental slope | Others |
| Neural networks | (Jiang et al., 1993) | Mid-ocean ridge | Geomorphology and geohazards |
| | (Marsh and Brown, 2009) | Continental shelf | Habitat Mapping and ecology |
| Other techniques | (Mountjoy et al., 2009) | Continental slope | Geomorphology and geohazards |
| | (Preston et al., 2001) | Coastal to inner shelf | |
| *Specific geomorphometry* | (Casalbore et al., 2011) | Volcanic island | Geomorphology and geohazards |
| | (Gee et al., 2001) | Volcanic island | |
| | (Green and Uken, 2008) | Continental slope | |
| | (Haflidason et al., 2005) | Continental slope | |
| | (Hühnerbach et al., 2004) | Continental slope and volcanic islands | |
| | (Issler et al., 2005) | Continental slope | |
| | (McAdoo et al., 2000) | Continental slope | |
| | (Micallef et al., 2008) | Continental slope | |
| | (Micallef and Mountjoy, 2011) | Continental slope | |
| | (Micallef et al., 2012) | Continental slope | |
| | (Micallef et al., 2014b) | Continental slope | |
| | (Micallef et al., 2014a) | Continental slope | |
| | (Mitchell and Searle, 1998) | Mid-ocean ridge | |
| | (Mitchell et al., 2002) | Volcanic island | |
| | (Mitchell et al., 2003) | Volcanic island | |
| | (Mitchell, 2003) | Volcanic island | |
| | (Mitchell, 2004) | Continental slope | |
| | (Mitchell, 2005) | Continental slope | |
| | (Rovere et al., 2014) | Continental slope | |
| | (Roy et al., 2015) | Coastal to inner shelf | |
| | (Stretch et al., 2006) | Volcanic island | |
| | (Vachtman et al., 2013) | Continental slope | |
| | (Costa and Battista, 2013) | Coastal | Habitat Mapping and ecology |
| | (Diesing et al., 2014) | Coastal to inner shelf | |

# Figures

Fig. 1: Geomorphometry is commonly implemented in five steps (Pike et al., 2009; Bishop et al., 2012), here adapted to the application of geomorphometry to the marine environment (left). The panels on the right describe the structure and elements addressed in Sect. 2 to Sect. 6 of this paper, and list reviews and important discussion papers on these elements.

Fig. 2: Cumulative number of publications (articles or reviews) listed in the Scopus database mentioning specific keywords in their title, abstract of keywords, by the end of June 2016 for land-based (top) and marine (bottom) geomorphometry publications. For the "Geomorphometry" curves, the keywords "geomorphometry" and "geomorphometric" were queried. The keyword "terrain analysis" was researched for the "Terrain Analysis" curves. For the terrestrial applications, the following terms were used to query the database: "topographic variables", "topographic attributes", "topographic derivatives", "terrain variables", "terrain attributes", "terrain derivatives", and "terrain morphology". These terms were also used for marine applications, in addition to "bathymetric variables", "bathymetric attributes", "bathymetric derivatives", "seafloor morphology" and "seabed morphology". The queries were all performed to exclude (for land-based publications) or include (for marine publications) the terms "marine", "ocean" and "underwater". We note that some common terms in the field (e.g. surface parameters, seafloor characterization) were not included because of their parallel use and different meaning in other fields that do not involve geomorphometry.

Fig. 3: Example of elements that can be extracted and visualised when using the CUBE algorithm, using the ROV-based dataset from Fig. 4 (source: Lecours and Devillers, 2015). On top, the components contributing to the horizontal and vertical TPUs can be studied. Other marginal contributions to the vertical TPU included the roll and pitch of the platform, timing of the inertial measurement unit, and uncertainty associated with the sonar system (range and angle). The combination of the GPS and delta draft provides the three-dimensional position of the soundings (x, y, z); in ROV-based research, the positional accuracy decreases with depth (Lecours and Devillers, 2015). On the bottom, it is possible to visualise how the uncertainty and the density of soundings vary spatially.

Fig. 4: Examples of errors and artefacts found in different datasets and their impact on derived terrain attributes. The top panels represent data from GEBCO (2014), which uses radar altimetry to fill in the gaps between higher-resolution, freely available bathymetric data. The main artefacts that can be observed are caused by the interpolation method that was used to combine the different datasets. For instance, a linear artefact following a Southwest to Northeast axis can be observed as a result of the combination of one SBES acoustic survey line with radar altimetry data. Similarly, some "spots" can be seen in the middle of the panels (South to North direction). These artefacts, especially apparent in the curvature, are caused by the merging of punctual lead line measurements with the radar altimetry data. Finally, a slight gridding artefact can be observed in the curvature (i.e. thin vertical and horizontal linear features). The middle panels show ship-based MBES data (Brown et al., 2012). The obvious artefacts follow the surveying pattern of the vessel, and are mainly caused by vessel motion that was not compensated properly by the motion sensor. Finally, the bottom row of panels corresponds to ROV-based MBES collected from 20 m above the seafloor in the deep sea (Lecours et al., 2013). In this case, the artefacts are caused by a combination of heave and other platform motions; the ancillary data collected to account for this motion are too uncertain at

this depth to appropriately correct for the errors (Lecours and Devillers, 2015). The "spots" observed in the bottom and top of the derived terrain attributes are spurious soundings that can be removed in bathymetric software during post-processing of the data. Note differences in spatial resolutions (left axis) and cartographic scales. Depth values of the top left panel range from 60 to 4,275 m deep, those of the middle left panel range from 20 to 105 m deep, and those of the bottom left panel range from 2,345 to 2,425 m deep. Lighter blue is shallower.

Fig. 5: Illustration of the main types of terrain attributes that can be derived from bathymetry data. Modified after Wilson et al. (2007).

Fig. 6: Indicative workflow showing the use of terrain attributes in predictive habitat mapping. Generally following some pre-selection of variables the observed habitat points (response variable) are combined with full coverage predictor variables selected from bathymetry, terrain attributes and other environmental variables as available to form the input to a habitat model which will be used to predict a full-coverage habitat map. The choice of habitat model will depend on the study in question but is typically either a statistical (e.g. GLM) or machine-learning based model (e.g. Random Forest). Observed habitat points are classified from visual or physical samples of the seabed. Terrain attributes are typically multi-scale and may include general and/or specific geomorphometry. Other environmental variables may include, for example, oceanographic data (temperature, salinity, current speed etc.) and geological data (e.g. grain size). A similar workflow applies to modelling of single species or communities, where the output will be a continuous map indicating the probability of occurrence within the study area, rather than a categorical map as shown here.

Fig. 7: Example of the use of marine geomorphometry to semi-automatically map the components of mega-scale submarine landslide offshore Norway (adapted from Micallef et al. (2009)). Figures a-c show the trough depth, ridge length and ridge spacing extracted from a multibeam echosounder map of the north-eastern Storegga Slide using ridge characterisation techniques (Micallef et al., 2007b). Figure d is a classification map generated by using these ridge characteristic maps as input layers in an unsupervised clustering algorithm (ISODATA). Figure e is an interpretative map of the range of spreading events based on figures a-d. Other mass movements and geological processes and structures have been interpreted using geomorphometric mapping (Micallef et al., 2007a).

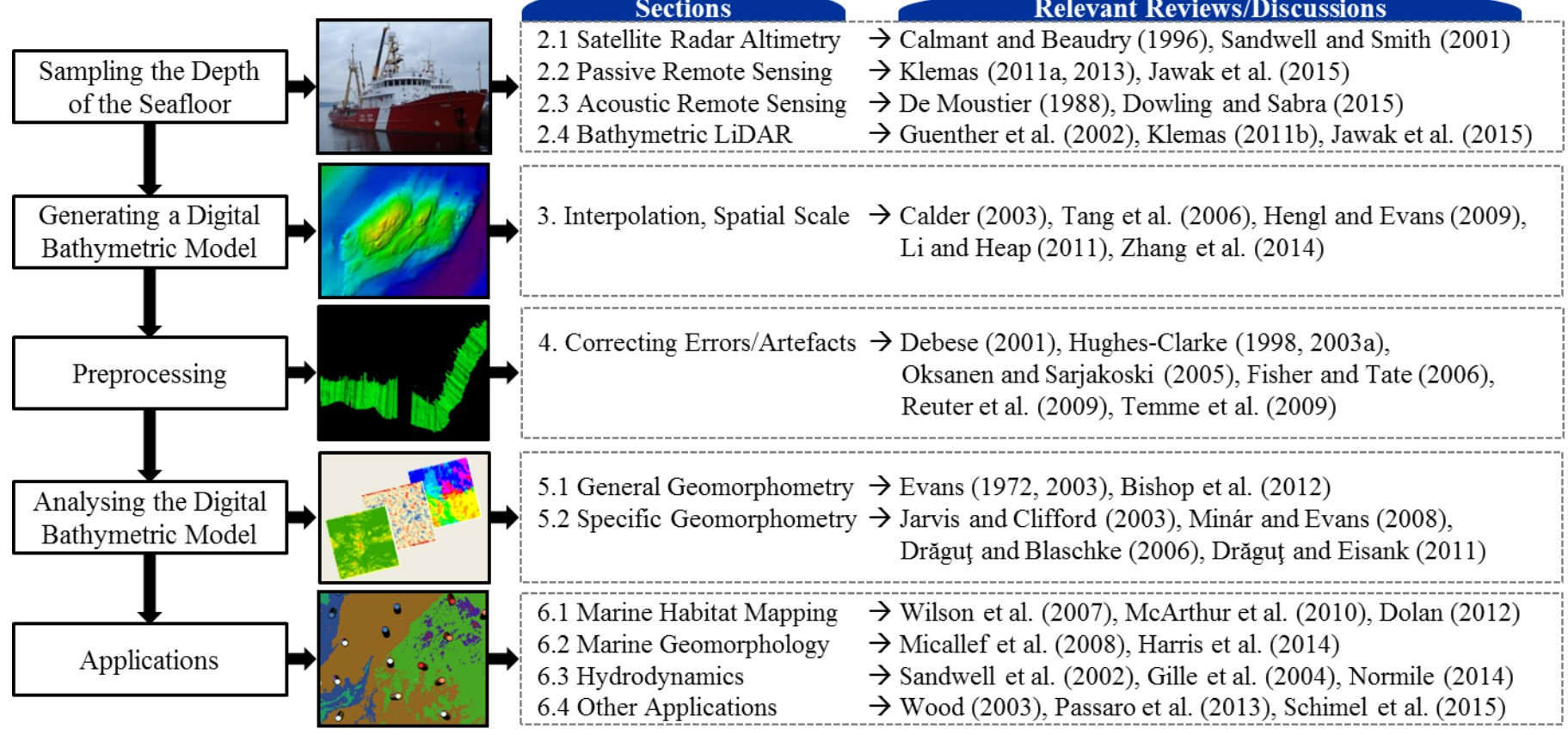

Figure 1

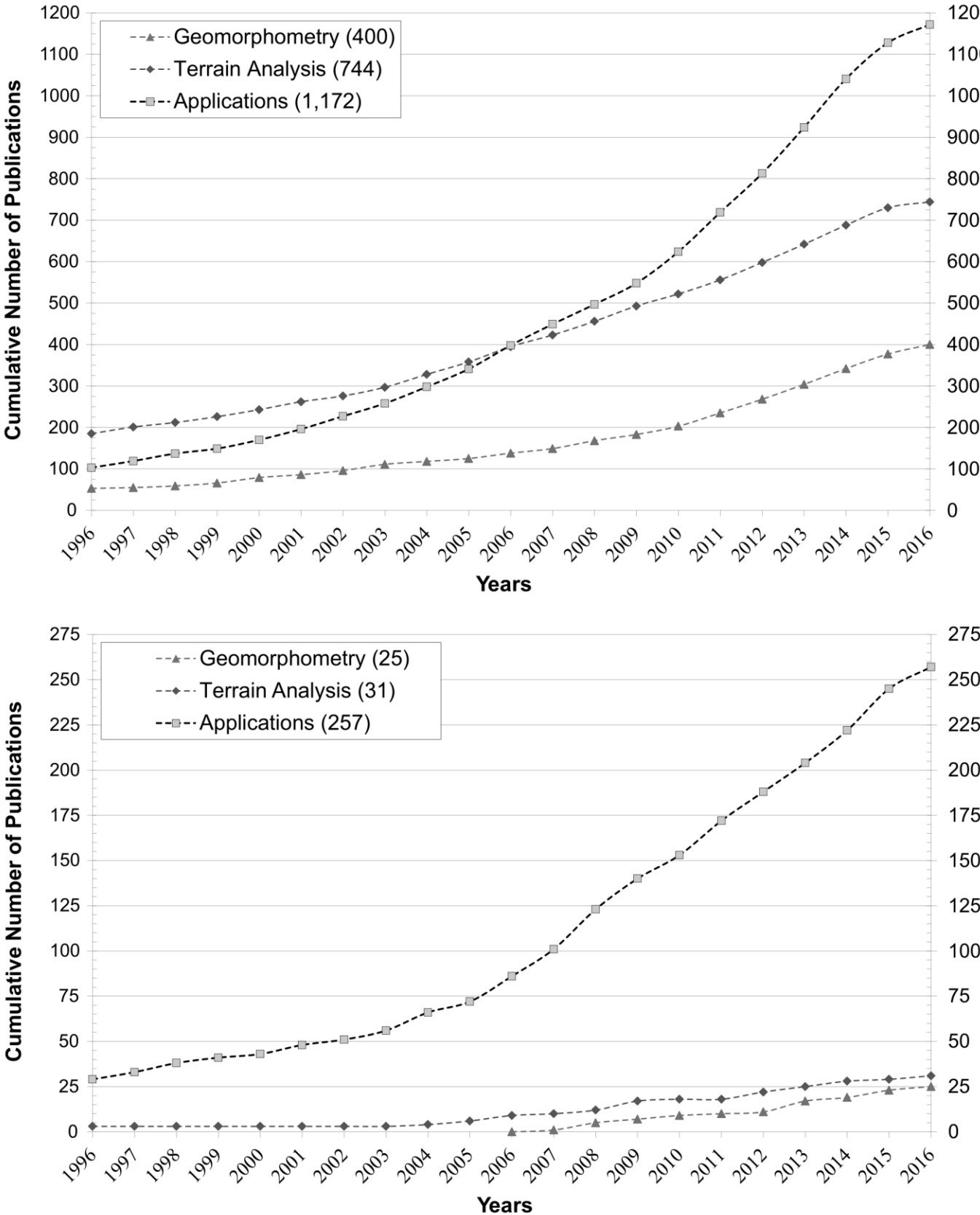

**Figure 2**

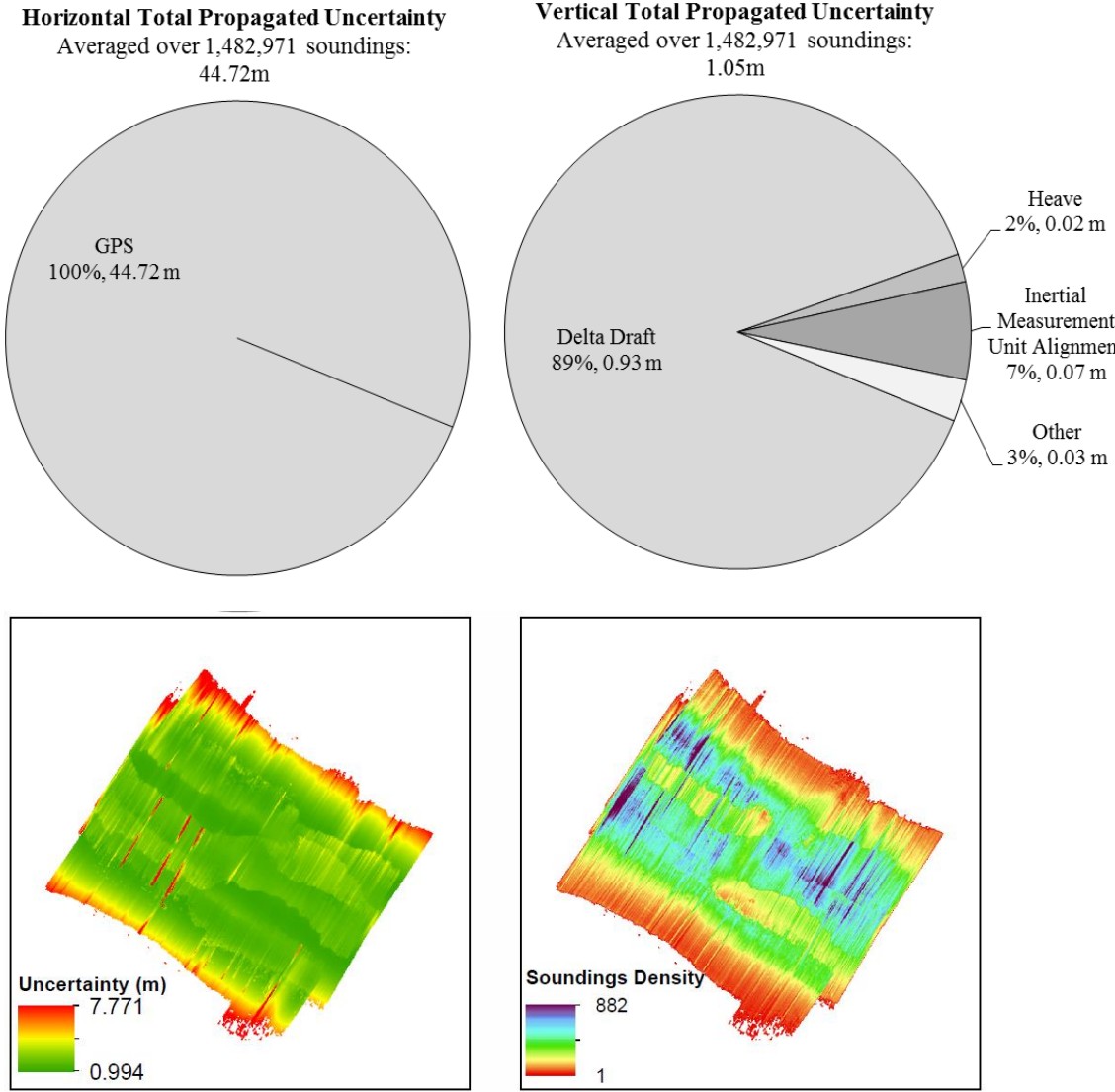

**Figure 3**

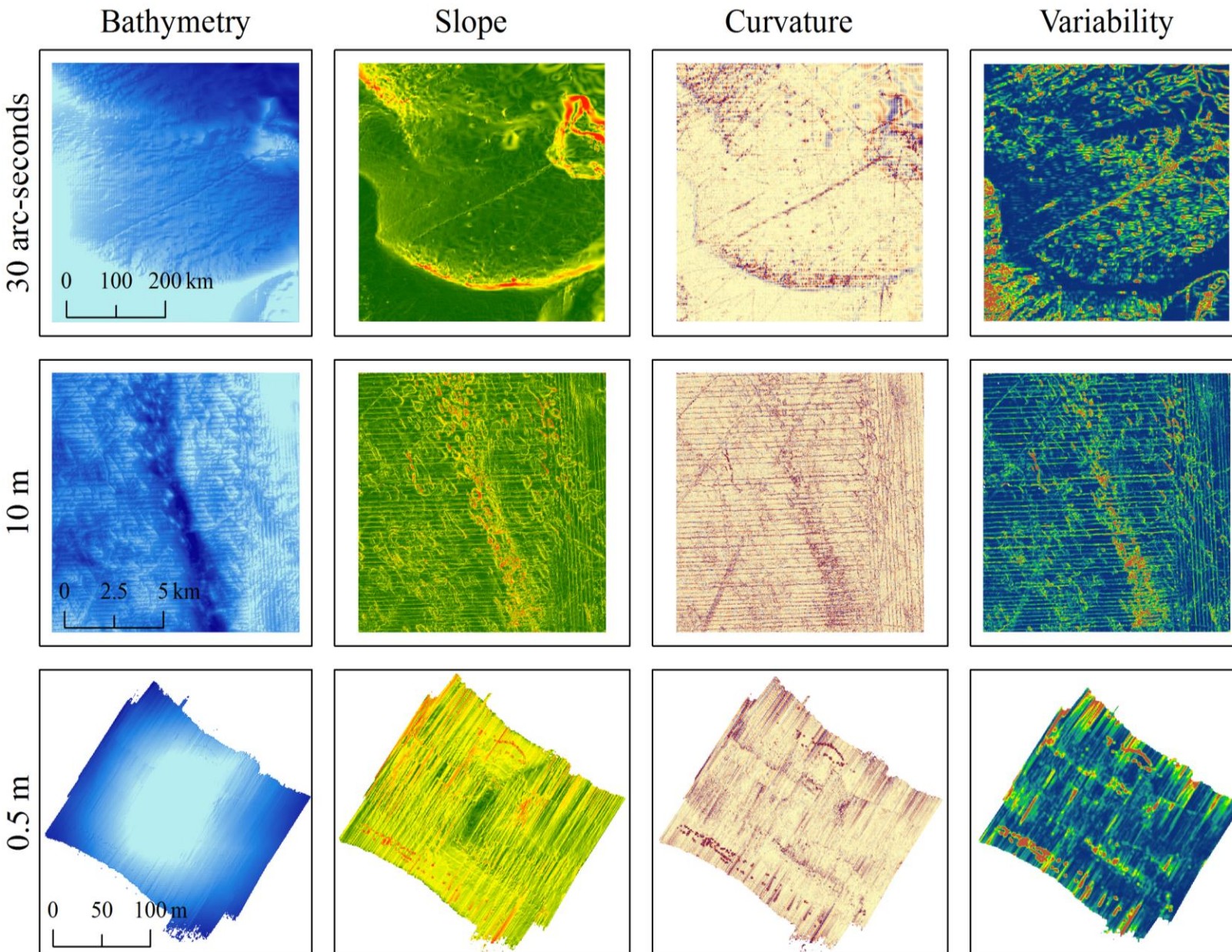

**Figure 4**

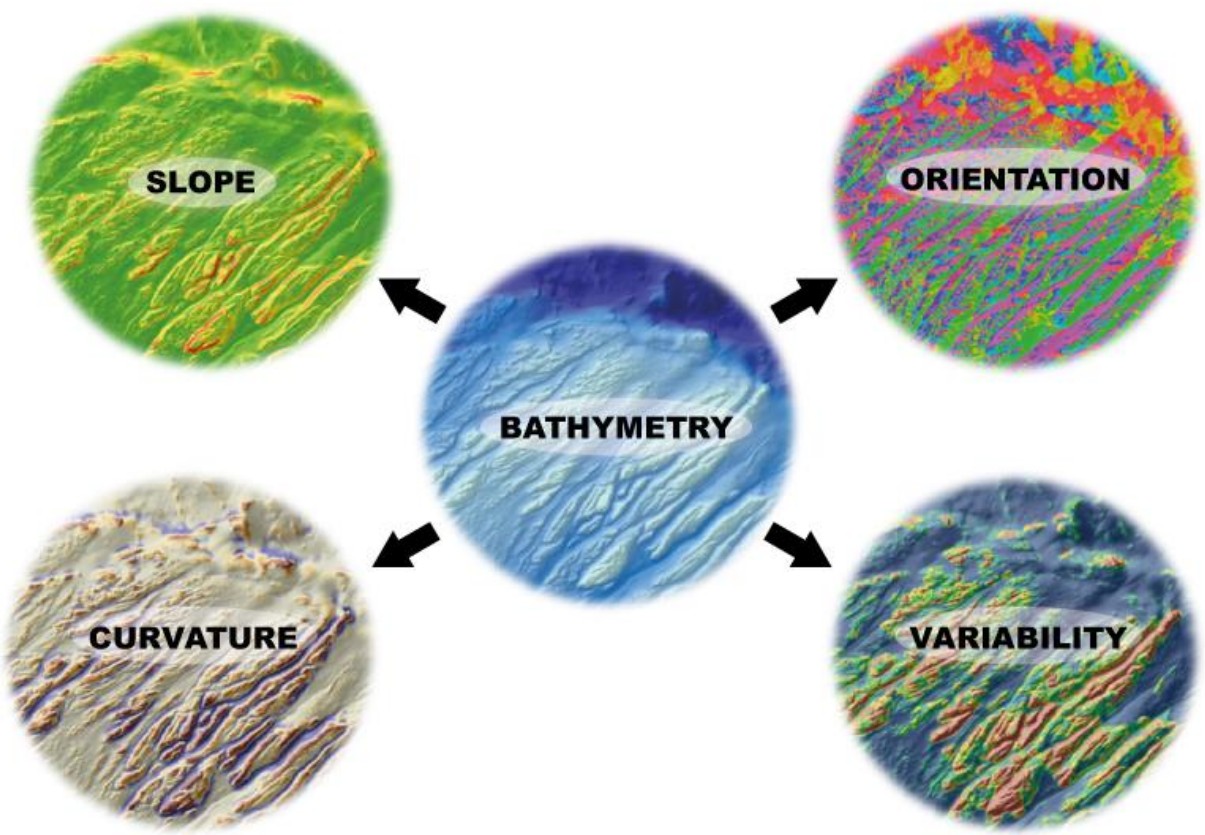

**Figure 5**

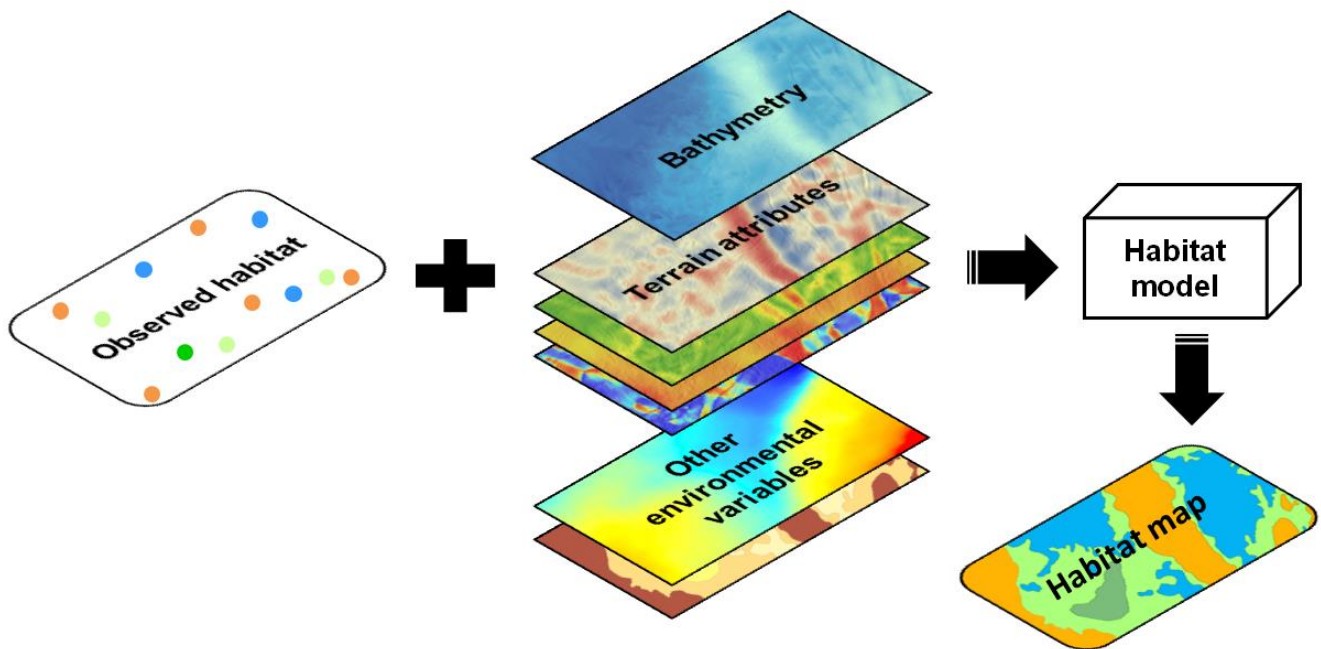

**Figure 6**

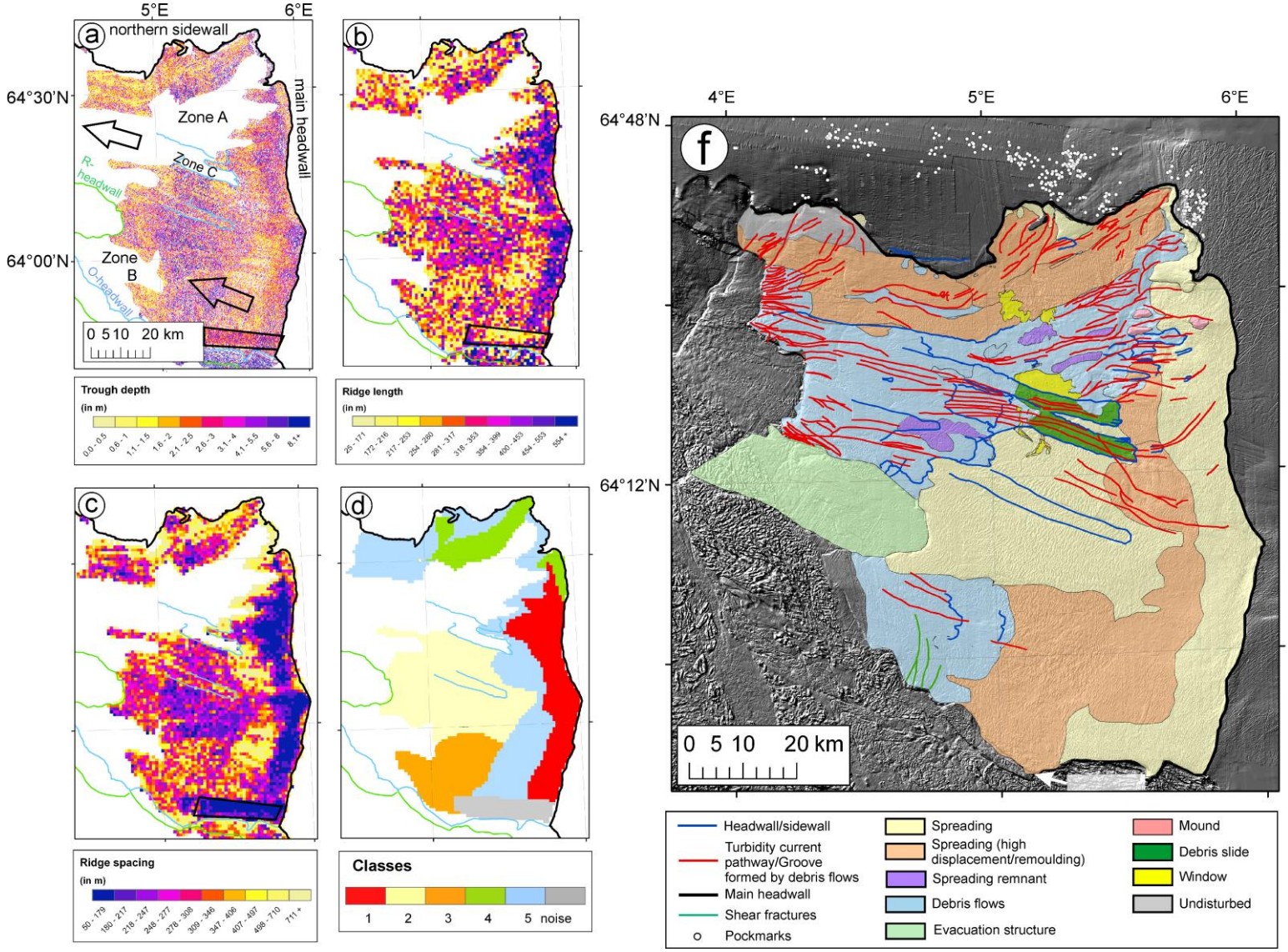

**Figure 7**