# Peer review of "A review of marine geomorphometry, the quantitative study of the seafloor"

_Hydrology and Earth System Sciences, 2016_

## Referee Comment (RC1) · N. Mitchell (Referee) · 18 Mar 2016

This article sets out to review efforts to quantify different aspects of shape of the ocean floors. It has ambitious intent, covering data collection, processing and analysis. A purpose of the article is to highlight the growth of the subject (hence justifying its inclusion in a discussion forum such as HESS).

I found the graphs in Figure 1 interesting. However, the terms 'geomorphometry' and 'geomorphometric analysis' are not widely used in marine geology and geophysics, so the shapes of the graphs in Figure 1b, as the authors acknowledge, strongly reflect how these terms have been adopted, rather than representing the rise in practice in this subject area. Researchers began frequently measuring aspects of ocean floor shape from at least the 1960s onwards if not earlier, e.g., the work in characterizing how

the seabed subsides with crustal age (some of the original articles cited by Parsons and Sclater (1977)). There is a lot of text in this new article devoted to data collection and processing, which is fine (and important), though I thought distracting and left little room for meaningful insight into marine geomorphometry, considering the term in its general sense. The later discussion I therefore found disappointing, focusing on largely practical issues. The authors have already invested significant effort in generating the present version of the article, so they may not wish to invest much further time, though I thought the basic practical steps and data issues (for sonar methods, for example) could be strongly reduced and relevant sources for this information cited, leaving more space for developing insight into how the analysis of bathymetry has evolved.

I would also like to encourage the authors to repeat Figure 1b after attempting to find out how many articles measured shape characteristics from bathymetry in practice. This may take some effort, but the tables presented suggest they have already got part of the way. I have suggested some articles at the end of this review from my own experience. I suspect there are many more, sponsored in part by the US navy (Office of Naval Research). The results would hopefully show how efforts compare with the history of instrument development and number of research vessels. There are at least data on the history of geophysical research cruises to compare against (Wessel & Chandler, 2011) and there may be other information on, for example, the sales of multibeam sonars available from the sonar companies.

As the term "geomorphometry" is not widely used, it may be better to use the title to clarify the meaning for readers not familiar with it (as the aim of the article stated in the conclusions section is to raise awareness). I suggest: "A review of marine geomorphometry, the quantitative study of the shapes of seabed features"

It is of course an author's decision what style to choose to write in. The text to me seems too exaggerated. It is almost in the style of a research grant proposal rather than in the dry style of a serious research article. This is important because the text tends to distract the reader from the facts presented. I recommend Strunk and White

(ref below), which may help in making the text more succinct.

Detailed comments The logic in the text needs to be tightened - some examples below, though I have not captured all problem text. Page/line # 1/10 ... the science of quantitatively describing terrains ... (science doesn't do anything strictly speaking, its the humans that do things in using the scientific method) 1/12 ... (GIS) and other software ... 1/15 ... investigate the characterization of -> characterize 1/19 delete "the science and application of" 1/19 ...we learn from experiences in terrestrial geomorphometry. 1/20 This sentence is too vague. The issues need to be spelt out. 1/23 This paragraph is the only description of the content of the article (the preceding text is background). In my opinion, this needs to be more extensive and the preceding text greatly reduced. 2/2 Geomorphological studies have improved ... 2/3 Morphology - and quantitative measures of topography - are considered .. components ... 2/4 ...because the interpretation of the origins of geomorphological features and their ages necessarily follow it/them 2/5 The shapes of terrestrial landscape are important for ... 2/11 Geomorphology also plays a fundamental role beneath sea level. I'm not sure the following sentence is correct - we always collect samples or data within the water column (depends what we define as the surface exactly). 2/14 .. for many subjects (the following list does not contain questions) 2/18 ... can affect the efficacy of model predictions of marine species dispersion.... (delete "of different elements") 2/20 ..that 90% of the oceans are ... 2/23 ... explaining what is meant by "explored". Delete: "The fact is that " 2/25 an estimate of global bathymetry, which revealed .. 2/29 were all identified as requiring -> require 2/29 Delete the sentence "It is therefore .." and instead specify resolution in previous sentence. 3/3 as above, a subject area does not measure, rather humans measure. Later in the sentence, we use the methods of these disciplines. 3/6 Sentence "Methods in ..." includes citations to recent articles - wouldn't this provide better historical context if the text started out with more original articles? Also, what is meant by "Methods" - this seems a bit too vague to me. 3/10 What "differences"? 3/22 I would not say the field is recent, only the use of the term "geomorphometry" is recent.

4/15. There have been attempts to reconstruct bathymetry from wave refraction, using the effect of water depth on wave celerity. To my knowledge, nobody has analyzed the results in the way considered here, however.

4/23 I wouldn't say the data were used to 'define' the geoid, rather they have been used to derive the marine geoid. The geoid is not 'relative' to sea level, it is sea level in the absence of any ocean currents (dynamic effects). The sea is an equipotential (almost). 4/27 Here 'gravitational field' is mentioned, but previous sentences talk about the geoid. 5/1 I wouldn't say that thick sediments are a problem. Rather, the method relies on using a single density value for the seabed in a given area, but in practice density varies. 5/7-9 This is vague - needs to be more explicit about what the data products contain. The Shuttle radar dataset was collected over land using interferometry, so the other data were needed to fill in marine areas. 5/25 ...in uniformly attenuating water ... 5/30 I would just say that the attenuation rate is obtained by correlating log10(local water depth measurements) with optical amplitude, if that is how it is done. The ground truth measurements don't 'verify' the relationships as such, rather they are used to derive them. 6/13 These methods do not rely on attenuation (please specify)? 6/16 bottom types vary with depth. 6/19 how shallow? 7/1 Please do not use words like 'vector' that have specific meanings in science. The velocity of sound in water is not increased by density (acoustic velocity actually decreases with density if occurring in isolation), rather water has a much higher bulk rigidity and that dominates velocity. 7/7-8 also knowing the speed of sound in the water. 7/13 ...seafloor, water depth can only be measured from the first echo in SSS amplitude data, the remaining parts of the data provide only indirect information on water depth. 7/15 ... by combining data from two receiving ... This is a similar form of bathymetry to that derived using multibeam sonars (e.g., the Kongsberg systems, which use the split beam method do compare phase between pairs of split beams). 7/18 how? This is too vague. This section is missing a couple of methods (unless contained in the cited literature, which is unclear from the text given) - relief of objects from acoustic shadow lengths (which has been used for many years, e.g., in military applications (mine assessments)) and shape-from-shading

(I think illustrated in Blondel and Murton). 7/27 many early systems were not narrow beam. The multibeam sonar is arguably the most important instrument used for marine surveying - it is a shame that it is described in such a limited way here. There is no adequate description of the sounding geometry and no mention of motion sensors and sound velocity issues (although some of these do appear later on). I suggest referring to one of the review articles on these instruments, such as by de Moustier. 8/18 This still doesn't quite say how these systems derive water depth from the data (e.g., time of flight and velocity of light, but time of flight from the laser or through the water (detection of sea surface and seabed)?). 9/7- No system generates truly continuous bathymetry, they all provide samples. Some systems effectively filter the bathymetry by imposing a resolution (e.g., acoustic beam width in the case of multibeams) before sampling. Other systems you might say sub-sample the bathymetry without first filtering, hence 'alias' the bathymetry (widely spaced single beam soundings, for example). 9/26 Also see Smith & Wessel (1990) for interpolation issues. 10/4 Is it the technique that is sensitive to errors or the DBM that is produced using it? 11/6 They are surely not the "principles" of radar altimetry that limit resolution, rather the beam width and noise characteristics of the data? This is too vague. 11/7 not the platform as such, but the distance of the platform from the seabed or sea surface. 11/11 I think why sampling density increases needs to be explained (because of the cycle time of sounding systems limited by the speed of sound in water and beam widths, etc). 11/28 .. and may be amplified in some attributes computed from them. 12/30 This is an obscure sentence. I think what is meant here is that there are no other data to check whether a feature is an artifact or not. 14/5 This is rather trivial and I'm not so sure that grids of bathymetry ought to be negative always (water depth is positive downwards) - a negative depth implies terrain above sea level. 15/29 and 31. I would stick to either 'variability' or 'rugosity' rather than use both terms. 16/1 Is meant here comparing depths with a planar surface fitted to the data? 16/3,4 alternative measures of rugosity that are less affected by gradient. 16/6 between them (what exactly?) 16/11-12 Not in marine geophysics. For example, the subsidence of the seabed with age was first looked at in the 1960s as far as I am aware

(references in Parsons and Sclater, 1977). 18/2,3 Is it bathymetry that is influencing the biology or rather a number of other properties vary with water depth (light because of attenuation, waves, etc)? 19/16 What is meant here exactly by scale invariance in terms of statistics and morphology? 20/1 I think references need to be cited for these, e.g., Fox et al or Malinverno for the spectral method. 24/32 I wouldn't say they are more affordable, rather the oil and gas industry has invested an enormous amount (with often dubious success) to acquire these datasets. Academic and government researchers have benefitted from access to these data. 26/27 Figure 1 only shows the uptake of the word 'geomorphometry', not practice. 27/7 If >50 years can be described as infancy, I am an infant!

p 55. Table 2 contains only a selection, so I would make that clear in the table title.

Figure 2 - surely "pre-processing" should occur before interpolating the data onto a grid? Or am I misunderstanding what pre-processing means here (an iterative stage of binning and gathering statistics before filtering the data and forming a final grid)? A better term might be needed.

Figure 3 caption. The last sentence does not really inform us of what is in the figure. Standard deviation is presumably computed within each cell. It is really not clear what is meant by "hypotheses" - I would not use that kind of term here for what I think is meant a working grid of the data. This kind of language is only going to confuse readers.

Figure 4. I understood the GEBCO charts to contain information from the hydrographic agencies that is not freely available. The caption needs to say where these data were taken from (geographically) for the discussion to have much meaning. I suspect the remaining data were not only obtained from altimetry measurements. axis -> direction We don't know the survey direction, ship track etc, so difficult to read the middle panel. Hence, "are mainly caused by the roll motion" - I suspect this is not roll motion exactly, rather a roll motion that is not compensated by the motion sensor used. Striping can arise because of other errors caused by bubbles passing over transducers in bad weather. Figure 6 is really not very useful, given that the sample set is effectively censored by the limited use of the terms searched.

_____________ NC Mitchell University of Manchester, March 2016

Suggested additional articles within the field of geomorphometry (only a small sub-set from my personal experience):

Czarnecki, M.F., Bergin, J.M., 1986. Characteristics of the two-dimensional spectrum of roughness on a seamount. Naval Research Laboratory. Goff, J.A., 1991. A global and regional stochastic analysis of near-ridge abyssal hill morphology. J. Geophys. Res. 96, 21,713-721,737. Goff, J.A., 1992. Quantative Characterstics of Abyssal Hill Morphology along flow line in the Atlantic Ocean. Jour. Geophys. Res. 97, 9183-9202. Goff, J.A., 2001. Quantitative classification of canyon systems on continental slopes and a possible relationship to slope curvature. Geophys. Res. Lett. 28, 4359-4362. Herzfeld, U.C., 1993. A method for seafloor classification using directional variograms, demonstrated for data from the western flank of the Mid-Atlantic Ridge. Math. Geol. 25, 901-924. Malinverno, A., 1990. A quantitative study of the axial topography of the Mid-Atlantic Ridge. J. Geophys. Res. 95, 2645-2660. Malinverno, A., 1991. Inverse square-root dependence of mid-ocean-ridge flank roughness on spreading rate. Nature 352, 58-60. Malinverno, A., 1993. Transition between a valley and a high at the axis of mid-ocean ridges. Geology 21, 639-642. Malinverno, A., Cowie, P.A., 1993. Normal faulting and the topographic roughness of mid-ocean ridge flanks. J. Geophys. Res. 98, 17921-17939. Malinverno, A., Gilbert, L.E., 1989. A stochastic model for the creation of ocean floor topography at a slow spreading center. J. Geophys. Res. 94, 1665-1675. Menard, H.W., 1984. Origin of guyots: the Beagle to Seabeam. J. Geophys. Res. 89, 11117-11123. Mitchell NC (1995) Diffusion transport model for pelagic sediments on the Mid-Atlantic Ridge. J. Geophys. Res. 100(B10):19,991-920,009 Mitchell NC, Huthnance JM (2007) Comparing the smooth, parabolic shapes of interfluves in continental slopes to predictions of diffusion transport models. Mar.

[Figure]

Geol. 236:189-208 Mitchell, N.C., Stretch, R., Oppenheimer, C., Kay, D., Beier, C., 2012. Cone morphologies associated with shallow marine eruptions: east Pico Island, Azores. Bull. Volcanol. 74, 2289-2300. Shaw, P.R., 1992. Ridge segmentation, faulting and crustal thickness in the Atlantic Ocean. Nature 358, 490-493. Shaw, P.R., Lin, J., 1993. Causes and consequences of variations in faulting style at a Mid-Atlantic Ridge. J. Geophys. Res. 98, 21839-21851. Shaw, P.R., Smith, D.K., 1987. Statistical methods for describing seafloor topography. Geophys. Res. Lett. 14, 1061-1064. Shaw, P.R., Smith, D.K., 1990. Robust description of statistically heterogeneous seafloor topography through its slope distribution. J. Geophys. Res. 95, 8705-8722. Smith, D.K., 1988. Shape analysis of Pacific seamounts. Earth Planet. Sci. Lett. 90, 457-466. Smith, D.K., 1996. Comparison of the shapes and sizes of seafloor volcanoes on Earth and "pancake" domes on Venus. J. Volcanol. Geotherm. Res. 73, 47-64. Smith, D.K., Jordan, T.H., 1988. Seamount statistics in the Pacific Ocean. J. Geophys. Res. 93, 2899-2918.

Other cited references: Blondel, P., Murton, B.J., 1997. Interpretation of sidescan sonar imagery. John Wiley, Chichester. de Moustier, C., 1988. State of the art in swath bathymetry survey systems. Internat. Hydr. Rev., Monaco 65, 25-54. Parsons, B., Sclater, J.G., 1977. An analysis of the variation of ocean floor bathymetry and heat flow with age. J. Geophys. Res. 82, 803-827. Smith, W.H.F., Wessel, P., 1990. Gridding with continuous curvature splines in tension. Geophysics 55, 293-305. Strunk, W., White, E.B., 1972. The elements of style, 2nd Ed. MacMillan Publishing, New York. Wessel P, Chandler MT (2011) The spatial and temporal distribution of marine geophysical surveys. Acta Geophysica 59:55-71

---

## Referee Comment (RC2) · Anonymous Referee #2 · 22 Mar 2016

The paper presents an exhaustive list of research dealing with marine geomorphometry. In this sense, it represents a thorough effort and a very complete picture of what has been done in the topic. I put forward the following suggestions which I think may help to improve the manuscript.

1. I would try to reduce the overall length of the manuscript. As it is now, the paper is not easy to read, it is too long, but especially, sometimes there is a lack of context to the referencing, and all the effort to compile the literature may be partially lost when reporting results of the search.

2. I think that the chapters are fine, but I would create tables with the references, avoiding what may be regarded as a long list of literature. Using tables, where the most important papers are reported, section by section, and using the text to discuss

the importance of those findings may be more convenient, and a context may be built for the references; thus the listing would be avoided, the text would be more readable, and most of the references would be included in tables.

3. Having said this, I think the authors should report on their thinking as derived from lessons learned from the literature. As the paper is now, that task is partly derived to the reader, and it is hinted at in the conclusions with the 5 major points highlighted by the authors.

Here some more detailed comments, as they came across while i read the paper

a. The introduction would benefit to the references to the most recent reviews about high resolution data and earth surface processes, specifically (Passalacqua et al., 2015; Tarolli, 2014). This would also hint to the current limits or merits in marine geomorphometry as related to this wider context of earth surface processes. b. As stated above, the manuscript as of now is a bit on the long side. There is a lot of pages focused on data collection and processing. While this is transversally useful, it hinders the part of the paper which is specific to the theme of the review (marine geo-morphometry). Currently, chapter 2 appears more as a technical report on the different techniques. The basic technical descriptions could be strongly reduced and referred to relevant sources, focusing more on the merits and limits of the different technologies, underlining the areas where there is still room for progress specifically for the field of marine geomorphometry. c. All the softwares reported in the article could be nicely organized and referenced in a table, so that a reader can have quick access to their name and location, and eventually reporting also the works that used one software or the other. d. Chapter 3 feels somehow not linked to the review. The chapter about scale could be merged to the different technique described in chapter 2, again highlighting the difference in scale as merit or limit of each technique, for example. e. Chapter 5 and 6 seems redundant. They could be merged together explaining what technique was used in the different studies. f. I found table 2 very interesting, but surprisingly this is reported only in reference to chapter 6.2, while it could be reorganized grouping the

works also according to the aim, not just the technique.

---

## Referee Comment (RC3) · Anonymous Referee #3 · 24 Apr 2016

**Short summary**

This paper aims at providing a timely review of marine geomorphometry. The intention is to raise awareness of the science of geomorphometry in marine environments, to review the existing literature on marine geomorphometry, to highlight differences from terrestrial geomorphometry, and to outline and recommend future research directions within marine geomorphometry. Overall the paper is structured according to the five steps outlined by Pike et al. (2009) in chapter 1 of the seminal book on geomorphometry edited by Hengl and Reuter.

**General comments**

As I was invited to review the paper after two other reviews were already available, I have tried not to repeat the comments and suggestions of the two other referees which

I both agree with. I instead try to focus on other issues of the paper. Overall I very much agree with the authors that there is a need for a review on marine geomorphometry. It is a discipline which is blooming immensely in these years, and this will for sure be the case in several years to come. However, I do have some comments of overall and general character:

- RE Structure: The five steps in geomorphometry outlined by Pike et al. (2009) are highly pedagogical in relation to teaching and learning geomorphometry as a discipline. However, the choice of structuring the paper according to these five steps and combining this with the aim of attempting an exhaustive review is very challenging, and more or less a Sisyphus task, as each of the five steps would be worthy of its own review. The result is that the review/analysis of each of the five steps tends to become surficial and to some extent rather being a listing of earlier studies. Consider an alternative structure with a clear alignment between the aim and objectives of the paper and the findings and recommendations based on the analysis of the authors. In this regard the aims and objectives could be even more precisely formulated, also in order to highlight the focus of the review. Referee #2 provides many good suggestions of how to alternatively structure the paper.

- RE Technologies: The section on sampling technologies (section 2) could be excluded from the review. Despite the dedication of five pages to the section it remains surficial, as it attempts to encompass a very broad range of technologies, including technologies like SBES which will not play a key role in the future. The data quality is fundamental for the DEM quality, and consequently for all DEM derivatives, and the final interpretations. However, this could be described and explained more generically by focusing on the general characteristics and properties of point data, which in essence is integrated areal information, as all points are related to a footprint. I very much agree with the authors that one of the key dangers in the future application of DEMs and geomorphometry in planning and management is that the planers, managers and decision makers are not aware of the properties of the data foundation of
the DEMs. Hence, consider highlighting data properties and not different technologies.

- RE Terminology: From a reader's point of view, the impression arises that the authors stress the importance of the differences between terrestrial and marine geomorphometry, as if the identification of differences would make the field of marine geomorphometry more relevant. This even leads to the suggestion of different terminologies for DEMs. In my opinion this is contra productive and merely confusing. The acronyms DEM, DSM and DTM are generic and sufficient for all environments. A more uniting approach with suggestions for a joint terminology and vocabulary across environments would be much more meriting. One of the major potentials of geomorphometry, as for geomorphology, is that it interacts with many disciplines. Within the geomorphometry community we should aim at aligning our terminology in order to foster and ease the coupling with other disciplines. Therefore, consider highlighting both similarities and differences between terrestrial and marine geomorphometry, and aim to unite the two disciplines wherever it is possible.

- RE Domain: The exact spatial domain of the review is unclear. There is a strong bias towards deep water environments with practically no review of the shallow water coastal environment. Nevertheless, the authors seem to use the coastline as the perimeter of their domain. Many studies, especially within the last decade, have used high resolution DEMs in shallow water coastal environments to quantify sediment transport, sediment transport pathways, morphology and morphodynamics. These studies are practically absent in the present review. Many of these studies are available in relation to the line of international conferences of MARID, RCEM and ICS, and in the journals JGR Earth Surface, Geomorphology, ESPL and more recently ESurf (I will not mention any specific studies here, as the list is long). Moreover, the authors mention the bridging of terrestrial and marine environments as one of the key future challenges (and actually a paper in the special issue is addressing this), but this without having reviewed the shallow water environments in the first place. Hence, the present analysis is simply to surficial. This was also highlighted by referee #1 although not in relation

HESSD
to shallow water. Consider defining the exact domain of the review, and then either fully include or exclude shallow water coastal environments. Specifically it even leads to erroneous information, e.g. when the authors discuss tidal corrections which are rarely used in high-precision and high-resolution shallow water environments where high-precision positioning is available and applied.

- RE Applications: A range of applications is listed. However, the descriptions of the applications are strongly biased towards habitat mapping, and to some extent also hydrography in relation to safety of navigation. The enormous potential of geomorphometry is that it has a vast amount of applications, and some not yet realised. It would suit the review if the analysis of the authors would lead to suggestions of new areas where geomorphometry has not yet been introduced and tested or highlight areas where present applications could be further developed.

I have not included any specific comments to the separate sections and also no comments for technical corrections. Due to the somehow fundamental character of the general comments from all three referees, it seems more relevant to initially restructure and refocus the paper, before more detailed corrections are suggested.

I sincerely hope that the authors will take up the challenge to revise the paper. Marine geomorphometry is a blooming field and we need an overview within the community, and we need this paper to ease communication with other disciplines that could benefit from marine geomorphometry.

**HESSD**

---

## Author Comment (AC1) · 25 May 2016

Dear Dr. Mitchell,

We thank you very much for your comprehensive and helpful review of our manuscript. We are currently working on addressing the three reviewers' comments while awaiting the decision of the editor.

Please find below a point-to-point summary of your main comments and how these will be addressed. In general, the text will be thoroughly revised to make it more succinct. As can be observed in our answers to the reviewers' comments below, this will entail a reduction of chapter 2 and a revision of chapters 6 and 7.

Specific comments:

1. "I found the graphs in Figure 1 interesting. However, the terms 'geomorphometry' and 'geomorphometric analysis' are not widely used in marine geology and geophysics, so the shapes of the graphs in Figure 1b, as the authors acknowledge, strongly reflect how these terms have been adopted, rather than representing the rise in practice in this subject area. Researchers began frequently measuring aspects of ocean floor shape from at least the 1960s onwards if not earlier, e.g., the work in characterizing how the seabed subsides with crustal age (some of the original articles cited by Parsons and Sclater (1977))."

We will clarify that Figure 1 shows how the terms 'geomorphometry' and 'geomorphometric analysis' have been adopted in the last two decades.

2. "There is a lot of text in this new article devoted to data collection and processing, which is fine (and important), though I thought distracting and left little room for meaningful insight into marine geomorphometry, considering the term in its general sense. The authors have already invested significant effort in generating the present version of the article, so they may not wish to invest much further time, though I thought the basic practical steps and data issues (for sonar methods, for example) could be strongly reduced and relevant sources for this information cited, leaving more space for developing insight into how the analysis of bathymetry has evolved."

We agree to remove the excessive details on data collection and processing and focus on the characteristics of the data collected within each technique, which is still very relevant to geomorphometry.

3. "I would also like to encourage the authors to repeat Figure 1b after attempting to find out how many articles measured shape characteristics from bathymetry in practice. This may take some effort, but the tables presented suggest they have already got part of the way. I have suggested some articles at the end of this review from my own experience. I suspect there are many more, sponsored in part by the US navy (Office of Naval Research). The results would hopefully show how efforts compare with the

history of instrument development and number of research vessels. There are at least data on the history of geophysical research cruises to compare against (Wessel & Chandler, 2011) and there may be other information on, for example, the sales of multibeam sonars available from the sonar companies."

Figure 1b will be revised to include articles measuring shape characteristics. This will be based on data from Table 2 and the literature suggested by the reviewer.

4. "As the term "geomorphometry" is not widely used, it may be better to use the title to clarify the meaning for readers not familiar with it (as the aim of the article stated in the conclusions section is to raise awareness). I suggest: "A review of marine geomorphometry, the quantitative study of the shapes of seabed features""

The title will be modified as suggested by the reviewer.

5. "The logic in the text needs to be tightened - some examples below, though I have not captured all problem text."

We thank the reviewer for all the examples provided (page 1 line 10 to page 27 line 7; figures 2, 3, 4, 6). We will carry out all these modifications.

---

## Author Comment (AC2) · 25 May 2016

We thank you for your time reviewing our manuscript. We are currently working on addressing the three reviewers' comments while awaiting the decision of the editor.

Please find below a point-to-point summary of your main comments and how these will be addressed. In general, the text will be thoroughly revised to make it more succinct. As can be observed in our answers to the reviewers' comments below, this will entail a reduction of chapter 2 and a revision of chapters 6 and 7.

1. I think that the chapters are fine, but I would create tables with the references, avoiding what may be regarded as a long list of literature. Using tables, where the most important papers are reported, section by section, and using the text to discuss the importance of those findings may be more convenient, and a context may be built

for the references; thus the listing would be avoided, the text would be more readable, and most of the references would be included in tables.

We will consider moving the references reported from the text to specific tables to make the text more readable.

2. Having said this, I think the authors should report on their thinking as derived from lessons learned from the literature. As the paper is now, that task is partly derived to the reader, and it is hinted at in the conclusions with the 5 major points highlighted by the authors.

We will include a section in chapter 7 on the lessons learned from past work on marine geomorphometry (particularly the limitations) and how these could be addressed in the future. We also welcome specific suggestions based on the experience of the reviewer.

3. The introduction would benefit to the references to the most recent reviews about high resolution data and earth surface processes, specifically (Passalacqua et al., 2015; Tarolli, 2014). This would also hint to the current limits or merits in marine geomorphometry as related to this wider context of earth surface processes.

A comparison with the more recent work in terrestrial geomorphometry will be made in chapter 7 to highlight the merits/limits of marine geomorphometry. We however note that the work of Passalacqua and colleagues is already included in the manuscript.

4. As stated above, the manuscript as of now is a bit on the long side. There is a lot of pages focused on data collection and processing. While this is transversally useful, it hinders the part of the paper which is specific to the theme of the review (marine geomorphometry). Currently, chapter 2 appears more as a technical report on the different techniques. The basic technical descriptions could be strongly reduced and referred to relevant sources, focusing more on the merits and limits of the different technologies, underlining the areas where there is still room for progress specifically for the field of marine geomorphometry.

[Figure]

We agree to remove the excessive details on data collection and processing and focus on the characteristics of the data collected within each technique, which is still very relevant to geomorphometry.

5. All the softwares reported in the article could be nicely organized and referenced in a table, so that a reader can have quick access to their name and location, and eventually reporting also the works that used one software or the other.

We will move the software reported from the text to specific tables to make the text more readable.

6. Chapter 3 feels somehow not linked to the review. The chapter about scale could be merged to the different technique described in chapter 2, again highlighting the difference in scale as merit or limit of each technique, for example.

Generation of a surface model plays a crucial role in the geomorphometric approach. To address this comment, we will cover the issue of spatial scale in chapter 2.

7. Chapter 5 and 6 seems redundant. They could be merged together explaining what technique was used in the different studies.

We prefer to keep the description of the techniques separate from the discussion of the application of geomorphometry. However, we will revise chapter 6 to ensure there is no needless repetition of the techniques described in chapter 5.

8. I found table 2 very interesting, but surprisingly this is reported only in reference to chapter 6.2, while it could be reorganized grouping the works also according to the aim, not just the technique.

We will modify the table by classifying the references according to the main theme of the study.

---

## Author Comment (AC3) · 25 May 2016

We thank you for your time reviewing our manuscript. We are currently working on addressing the three reviewers' comments while awaiting the decision of the editor.

Please find below a point-to-point summary of your main comments and how these will be addressed. In general, the text will be thoroughly revised to make it more succinct. As can be observed in our answers to the reviewers' comments below, this will entail a reduction of chapter 2 and a revision of chapters 6 and 7.

1. RE Structure: The five steps in geomorphometry outlined by Pike et al. (2009) are highly pedagogical in relation to teaching and learning geomorphometry as a discipline. However, the choice of structuring the paper according to these five steps and combining this with the aim of attempting an exhaustive review is very challenging, and
more or less a Sisyphus task, as each of the five steps would be worthy of its own review. The result is that the review/analysis of each of the five steps tends to become surficial and to some extent rather being a listing of earlier studies. Consider an alternative structure with a clear alignment between the aim and objectives of the paper and the findings and recommendations based on the analysis of the authors. In this regard the aims and objectives could be even more precisely formulated, also in order to highlight the focus of the review. Referee #2 provides many good suggestions of how to alternatively structure the paper.

We agree with the reviewer that each section would be worthy of its own review. Our objective was not to write a review of each of these steps, but to review how they integrate together to form a typical marine geomorphometry workflow. We have discussed the matter and decided to keep the current structure of the manuscript based on Pike et al. (2009). It is often observed in the community of end-users that there is a lack of understanding of the fact that each of the five steps has implications for the derivation of terrain attributes or extraction of terrain features, and consequently implications for the final application. By structuring the review following these steps, we hope that end-users can realize the role of each step in determining their quantitative terrain characteristics. However, we will make some sections, e.g. section 2 on the remote sensing techniques used to collect depth data, shorter and shift their focus towards data characteristics and how they impact the final application. In doing so, we hope to make the review less surficial and more focused on the importance of each step for the final application.

2. RE Technologies: The section on sampling technologies (section 2) could be excluded from the review. Despite the dedication of five pages to the section it remains surficial, as it attempts to encompass a very broad range of technologies, including technologies like SBES which will not play a key role in the future. The data quality is fundamental for the DEM quality, and consequently for all DEM derivatives, and the final interpretations. However, this could be described and explained more gener-
ically by focusing on the general characteristics and properties of point data, which in essence is integrated areal information, as all points are related to a footprint. I very much agree with the authors that one of the key dangers in the future application of DEMs and geomorphometry in planning and management is that the planers, managers and decision makers are not aware of the properties of the data foundation of the DEMs. Hence, consider highlighting data properties and not different technologies.

We agree to remove the excessive details on data collection and processing and focus on the characteristics of the data collected within each technique, which is still very relevant to geomorphometry.

3. RE Terminology: From a reader's point of view, the impression arises that the authors stress the importance of the differences between terrestrial and marine geomorphometry, as if the identification of differences would make the field of marine geomorphometry more relevant. This even leads to the suggestion of different terminologies for DEMs. In my opinion this is contra productive and merely confusing. The acronyms DEM, DSM and DTM are generic and sufficient for all environments. A more uniting approach with suggestions for a joint terminology and vocabulary across environments would be much more meriting. One of the major potentials of geomorphometry, as for geomorphology, is that it interacts with many disciplines. Within the geomorphometry community we should aim at aligning our terminology in order to foster and ease the coupling with other disciplines. Therefore, consider highlighting both similarities and differences between terrestrial and marine geomorphometry, and aim to unite the two disciplines wherever it is possible.

We agree with this comment and will endeavour to highlight the similarities between terrestrial and marine geomorphometry, and to use a uniform terminology. In the interest of precision when using these terms, we note that DSM or DTM are more appropriate terms than DEM when referring to the marine environment.

4. RE Domain: The exact spatial domain of the review is unclear. There is a strong
bias towards deep water environments with practically no review of the shallow water coastal environment. Nevertheless, the authors seem to use the coastline as the perimeter of their domain. Many studies, especially within the last decade, have used high resolution DEMs in shallow water coastal environments to quantify sediment transport, sediment transport pathways, morphology and morphodynamics. These studies are practically absent in the present review. Many of these studies are available in relation to the line of international conferences of MARID, RCEM and ICS, and in the journals JGR Earth Surface, Geomorphology, ESPL and more recently ESurf (I will not mention any specific studies here, as the list is long). Moreover, the authors mention the bridging of terrestrial and marine environments as one of the key future challenges (and actually a paper in the special issue is addressing this), but this without having reviewed the shallow water environments in the first place. Hence, the present analysis is simply to surficial. This was also highlighted by referee #1 although not in relation to shallow water. Consider defining the exact domain of the review, and then either fully include or exclude shallow water coastal environments. Specifically it even leads to erroneous information, e.g. when the authors discuss tidal corrections which are rarely used in high-precision and high-resolution shallow water environments where high-precision positioning is available and applied.

Our spatial domain does extend to the coastline, and our reference list already includes 25 papers focusing on coastal/shallow water environments. We will include a better coverage of these papers (as well as others) in the three applications of marine geomorphometry in section 6. We will also make more explicit the scope of the sub-section related to the littoral gap.

5. RE Applications: A range of applications is listed. However, the descriptions of the applications are strongly biased towards habitat mapping, and to some extent also hydrography in relation to safety of navigation. The enormous potential of geomorphometry is that it has a vast amount of applications, and some not yet realised. It would suit the review if the analysis of the authors would lead to suggestions of new ar-
eas where geomorphometry has not yet been introduced and tested or highlight areas where present applications could be further developed.

The bias towards habitat mapping and geomorphology is in line with the fact that these are the two most common applications of marine geomorphometry, as illustrated in one of the figures. However, we agree with the reviewer that it would suit the review to suggest new areas of application. We will extend chapter 7 to propose new areas where marine geomorphometry may be applied with success in the near future. We also welcome specific examples that the reviewer would like to see included.

**HESSD**

---

## Author Response (AR1)

**Change report**

Manuscript # hess-2016-73

**Reviewer #1 – Dr. Neil Mitchell**

| # | Comment | Revision and Comments |
|---|---------|----------------------|
| 1 | "This article sets out to review efforts to quantify different aspects of shape of the ocean floors. It has ambitious intent, covering data collection, processing and analysis. A purpose of the article is to highlight the growth of the subject (hence justifying its inclusion in a discussion forum such as HESS)." | We thank Dr. Mitchell for acknowledging the relevance of including our manuscript in HESS and for his helpful comments. |
| 2 | "I found the graphs in Figure 1 interesting. However, the terms 'geomorphometry' and 'geomorphometric analysis' are not widely used in marine geology and geophysics, so the shapes of the graphs in Figure 1b, as the authors acknowledge, strongly reflect how these terms have been adopted, rather than representing the rise in practice in this subject area. Researchers began frequently measuring aspects of ocean floor shape from at least the 1960s onwards if not earlier, e.g., the work in characterizing how the seabed subsides with crustal age (some of the original articles cited by Parsons and Sclater (1977))." | With this comment, the reviewer introduced an aspect that we realize we had neglected in the initial version of our paper: the temporal aspect of marine geomorphometry. With the temporal aspect comes the different ways that the field was defined through time. This review addresses what Minar and Evans (2008, Geomorphology 95) called "modern geomorphometry", a science that is comprehensively reviewed in Hengl and Reuter (2009, Developments in Soil Science 33).

To avoid confusing readers in terms of timeline and definitions of geomorphometry, we added a section (1.2) in which we defined our scope more clearly and put our review into historical context. We acknowledged the important contribution that was made to the field by the qualitative study of seafloor morphology from DTMs as early as the 1960s, but our review, like in Hengl and Reuter (2009), addresses geomorphometry as known since the theoretical developments of the 1970s and their implementation in computers in the 1980s. We did not consider in our review works that defined depth values and thus modelled the seafloor, but did not extract any additional quantitative measurements to characterize seafloor morphology. |

| 3 | "There is a lot of text in this new article devoted to data collection and processing, which is fine (and important), though I thought distracting and left little room for meaningful insight into marine geomorphometry, considering the term in its general sense. The later discussion I therefore found disappointing, focusing on largely practical issues. The authors have already invested significant effort in generating the present version of the article, so they may not wish to invest much further time, though I thought the basic practical steps and data issues (for sonar methods, for example) could be strongly reduced and relevant sources for this information cited, leaving more space for developing insight into how the analysis of bathymetry has evolved." | We agree with the reviewer that the details on the different techniques may have been distracting. We thus reduced this section to focus on the elements from these techniques that dictate data characteristics and are thus still very relevant to geomorphometry. The details about the theories behind the different techniques were moved to an appendix in order to leave this information available for readers that may want to deepen their understanding of these techniques. We also added a list of relevant reviews and discussions in Figure 1. |
|---|---|---|
| 4 | "I would also like to encourage the authors to repeat Figure 1b after attempting to find out how many articles measured shape characteristics from bathymetry in practice. This may take some effort, but the tables presented suggest they have already got part of the way. I have suggested some articles at the end of this review from my own experience. I suspect there are many more, sponsored in part by the US navy (Office of Naval Research). The results would hopefully show how efforts compare with the history of instrument development and number of research vessels. There are at least data on the history of geophysical research cruises to compare against (Wessel & Chandler, 2011) and there may be other information on, for example, the sales of multibeam sonars available from the sonar companies." | We have attempted to find a better source of information to revise Figure 1 (Figure 2 in the revised article). It was however difficult to establish a consistent and valid methodology for the meta-analysis. For instance, we felt that including research funded by the US Navy but not from other navies or government agencies (e.g. British, Italian, Spanish and Norwegian) would induce a strong bias in the analysis. We encountered the same issue when looking into instrument development and problems with companies that do not exist anymore or that got bought by others.

In order to still attempt to provide a better figure, we broadened the range of terms queried in the Scopus database, which gave a better estimate of the work that has been done in marine geomorphometry that the previous version. |
| 5 | "As the term "geomorphometry" is not widely used, it may be better to use the title to clarify the meaning for readers not familiar with it (as the aim of the article stated in the conclusions section is to raise awareness). I suggest: "A review of marine geomorphometry, the quantitative study of the shapes of seabed features"" | We modified the title to make it clearer to readers that are not familiar with the topic. |

| | | |
|---|---|---|
| 6 | "It is of course an author's decision what style to choose to write in. The text to me seems too exaggerated. It is almost in the style of a research grant proposal rather than in the dry style of a serious research article. This is important because the text tends to distract the reader from the facts presented. I recommend Strunk and White (ref below), which may help in making the text more succinct. The logic in the text needs to be tightened - some examples below, though I have not captured all problem text." | We thank the reviewer for the examples provided. We made the modifications where appropriate according to the comments below, in addition to have modified the text in other parts. We also tried to limit our use of the passive voice in new sections. |
| 8 | "Page/line # 1/10 …the science of quantitatively describing terrains... (science doesn't do anything strictly speaking, its the humans that do things in using the scientific method)" | We changed it to "the science that helps quantitatively describe terrains", according to the definitions provided by Chorley et al. (1957) and Pike et al. (2009). |
| 9 | "1/12 ... (GIS) and other software" | This was reviewed and now reads "(GIS) and spatial analysis software" |
| 10 | "1/15 ... investigate the characterization of -> characterize" | This was reviewed and now reads "characterize seabed terrain " |
| 11 | "1/19 delete "the science and application of"" | This was deleted. |
| 12 | "1/19 ...we learn from experiences in terrestrial geomorphometry." | This was modified at line 20 to read "we learn from experiences in terrestrial studies. " |
| 13 | "1/20 This sentence is too vague. The issues need to be spelt out." | We added three examples of issues but preferred not to elaborate further considering it is still only the abstract. |
| 14 | "1/23 This paragraph is the only description of the content of the article (the preceding text is background). In my opinion, this needs to be more extensive and the preceding text greatly reduced." | We elaborated on the different sections to give the reader a better idea of the elements that are reviewed and discussed in the paper. We also reduced the preceding text (e.g. lines 1/11-1/14 of the original manuscript). |
| 15 | "2/2 Geomorphological studies have improved" | This has been changed at lines 1/30 to read "To ensure that geomorphometry is used and developed to its full potential….." |
| 16 | "2/3 Morphology - and quantitative measures of topography - are considered... components" | The suggested change was made from line 2/7 This has been changed to read "Morphology and quantitative measures of topography are considered the most important components of geomorphology because they represent the age and origin of the landscape (Speight, 1974; Minár and Evans, 2008; |

| | | Bishop et al., 2012). The shape of the landscape influences many Earth systems across a range of scales." |
|---|---|---|
| **17** | "2/4 ...because the interpretation of the origins of geomorphological features and their ages necessarily follow it/them" | The proposed change was made at lines. "because they represent the age and origin of the landscape". |
| **18** | "2/5 The shapes of terrestrial landscape are important for" | The change was made at line 2/9. |
| **19** | "2/11 Geomorphology also plays a fundamental role beneath sea level. I'm not sure the following sentence is correct - we always collect samples or data within the water column (depends what we define as the surface exactly)." | This sentence was ambiguous and unnecessary, and was thus removed in an attempt to reduce the length of the introduction (cf. comment #3). |
| **20** | "2/14 …for many subjects (the following list does not contain questions)" | This is right; we made the suggested change at line 2/15. |
| **21** | "2/18 ... can affect the efficacy of model predictions of marine species dispersion.... (delete "of different elements")" | This sentence was modified at line 2/18 and "different elements" was deleted. |
| **22** | "2/20 ...that 90% of the oceans are" | Changed at line 2/20 to read "90% of the global ocean is unexplored". |
| **23** | "2/23 ...explaining what is meant by "explored". Delete: "The fact is that "" | These words were deleted. |
| **24** | "2/25 an estimate of global bathymetry, which revealed" | This sentence was modified at line 2/24 to read "The entire ocean floor has been mapped to a resolution of a few kilometres using satellites, which has created an estimated surface of global bathymetry (Smith and Sandwell, 1994)." |
| **25** | "2/29 were all identified as requiring -> require" | The change was made (line 2/27). |
| **26** | "2/29 Delete the sentence "It is therefore .." and instead specify resolution in previous sentence." | The sentence was removed and the resolution was added to the previous sentence (line 2/28). |
| **27** | "3/3 as above, a subject area does not measure, rather humans measure. Later in the sentence, we use the methods of these disciplines." | The sentence was modified to "Geomorphometry is defined as the science on which quantitative measurements of terrain morphology are based, with foundations in geosciences, mathematics, and computer sciences (Chorley et al., 1957; Mark, 1975; Pike et al., 2009)." |

| | | |
|---|---|---|
| **28** | "3/6 Sentence "Methods in ..." includes citations to recent articles - wouldn't this provide better historical context if the text started out with more original articles? Also, what is meant by "Methods" - this seems a bit too vague to me." | Reference to recent articles were added (e.g. Qin et al. 2013, Podobnikar and Székely, 2015, Rigol-Sanchez et al., 2015). The sentence including "methods" was modified to be more specific (line 3/8). |
| **29** | "3/10 What "differences"?" | Two examples were added as an introduction to what is discussed later in the manuscript (lines 3/10-12). |
| **30** | "3/22 I would not say the field is recent, only the use of the term "geomorphometry" is recent" | We set our review in a temporal context in Sect. 1.2 and supported it with appropriate referencing (cf. comment #2). Since modern terrestrial geomorphometry is considered young, we believe that it is appropriate to say that marine geomorphometry is "recent". In order to avoid ambiguity, we elaborated on this in Section 1.2 and modified "recent" for "relatively recent" where appropriate. |
| **31** | "4/15. There have been attempts to reconstruct bathymetry from wave refraction, using the effect of water depth on wave celerity. To my knowledge, nobody has analyzed the results in the way considered here, however." | We shortened and simplified this sentence, simply stating an example and the appropriate reference. |
| **32** | 4/23 I wouldn't say the data were used to 'define' the geoid, rather they have been used to derive the marine geoid. The geoid is not 'relative' to sea level, it is sea level in the absence of any ocean currents (dynamic effects). The sea is an equipotential (almost). | This section was moved to an appendix. We removed "define", kept the previous mention of "derive", and removed "relative to mean sea level". |
| **33** | "4/27 Here 'gravitational field' is mentioned, but previous sentences talk about the geoid." | This section is now in an appendix. The link between geoid and gravitational field was made explicit in line 3 of this section. |
| **34** | "5/1 I wouldn't say that thick sediments are a problem. Rather, the method relies on using a single density value for the seabed in a given area, but in practice density varies." | This was removed from the text in the appendix. |
| **35** | "5/7-9 This is vague - needs to be more explicit about what the data products contain. The Shuttle radar dataset was collected over land using interferometry, so the other data were needed to fill in marine areas." | We specified that these datasets also include terrestrial data (line 6/7). |

| 36 | "5/25 ...in uniformly attenuating water" | The change was made in the appendix as suggested. |
|----|----|----|
| 37 | "5/30 I would just say that the attenuation rate is obtained by correlating log10(local water depth measurements) with optical amplitude, if that is how it is done. The ground truth measurements don't 'verify' the relationships as such, rather they are used to derive them." | This sentence, now moved to the appendix, was modified. |
| 38 | "6/13 These methods do not rely on attenuation (please specify)?" | This comment is no longer relevant since the methods were moved to the appendix but the description of the types of data remained in the main text. Adding details on attenuation would involve bringing back the details on methods from the appendix to the main text. We added references to adequate review papers that describe these principles in detail, both in the appendix and the main text, and in Figure 1. |
| 39 | "6/16 bottom types vary with depth." | This sentence, now in the appendix, was ambiguous and we modified it. Bottom types do not necessarily vary with depth. Bottom reflectance is one of the main sources of error in bathymetric estimation from optical remote sensing. By extracting the spectral signature of the different bottom types, it is possible to correct for these errors by "removing" the effect of bottom reflectance and thus keep only the spectral return relevant to estimating depth. |
| 40 | "6/19 how shallow?" | Specifications were added at lines 6/24-28. |
| 41 | "7/1 Please do not use words like 'vector' that have specific meanings in science. The velocity of sound in water is not increased by density (acoustic velocity actually decreases with density if occurring in isolation), rather water has a much higher bulk rigidity and that dominates velocity." | The word "vector" was removed in this section of the appendix. |
| 42 | "7/7- 8 also knowing the speed of sound in the water." | This was added in the appendix. |
| 43 | "7/13 ...seafloor, water depth can only be measured from the first echo in SSS amplitude data, the remaining parts of the data provide only indirect | This was specified in the appendix. |

| | | |
|---|---|---|
| | information on water depth." | |
| 44 | "7/15 ... by combining data from two receiving ... This is a similar form of bathymetry to that derived using multibeam sonars (e.g., the Kongsberg systems, which use the split beam method do compare phase between pairs of split beams)." | This was modified at lines 7/21-22. |
| 45 | "7/18 how? This is too vague. This section is missing a couple of methods (unless contained in the cited literature, which is unclear from the text given) - relief of objects from acoustic shadow lengths (which has been used for many years, e.g., in military applications (mine assessments)) and shape-from-shading (I think illustrated in Blondel and Murton)." | We added details on this at lines 7/23-26. Since all three reviewers found that section too long and technical, most of its content was moved to the appendix and we did not describe in more details the different methods. |
| 46 | "7/27 many early systems were not narrow beam. The multibeam sonar is arguably the most important instrument used for marine surveying - it is a shame that it is described in such a limited way here. There is no adequate description of the sounding geometry and no mention of motion sensors and sound velocity issues (although some of these do appear later on). I suggest referring to one of the review articles on these instruments, such as by de Moustier." | As the three reviewers agreed that this section was too long and technical, we decided not to elaborate on the description of MBES. However, we added references to Lurton (2010) that describe in details principles of MBES, including sounding geometry, motion sensors and sound velocity, and references to de Moustier (1988). These references were also added to Figure 1 and the appendix. |
| 47 | "8/18 This still doesn't quite say how these systems derive water depth from the data (e.g., time of flight and velocity of light, but time of flight from the laser or through the water (detection of sea surface and seabed)?)" | We added a reference to the appendix and elaborated on this in it to avoid getting too long and technical in the main manuscript. |
| 48 | "9/7- No system generates truly continuous bathymetry, they all provide samples. Some systems effectively filter the bathymetry by imposing a resolution (e.g., acoustic beam width in the case of multibeams) before sampling. Other systems you might say sub-sample the bathymetry without first filtering, hence 'alias' the bathymetry (widely spaced single beam soundings, for example)." | The reviewer is right. We corrected this at line 7/13 to read- swath coverage. |
| 49 | "9/26 Also see Smith & Wessel (1990) for interpolation issues." | A sentence referring to the conclusions of Smith and Wessel was added at lines 10/2-4. |
| 50 | "10/4 Is it the technique that is sensitive to errors or the DBM that is produced using it?" | It is the technique; we specified it at lines 10/7-8. |

| 51 | "11/6 They are surely not the "principles" of radar altimetry that limit resolution, rather the beam width and noise characteristics of the data? This is too vague." | This section was removed (cf. comment #8 of Reviewer #2), but this particular sentence was moved and modified at lines 5/11-13. |
|---|---|---|
| 52 | "11/7 not the platform as such, but the distance of the platform from the seabed or sea surface." | This was clarified at line 5/13. |
| 53 | "11/11 I think why sampling density increases needs to be explained (because of the cycle time of sounding systems limited by the speed of sound in water and beam widths, etc)." | This is now explained at lines 11/5-7. |
| 54 | "11/28 .. and may be amplified in some attributes computed from them." | This was changed as suggested. |
| 55 | "12/30 This is an obscure sentence. I think what is meant here is that there are no other data to check whether a feature is an artifact or not." | This was clarified at line 12/26-29. |
| 56 | "14/5 This is rather trivial and I'm not so sure that grids of bathymetry ought to be negative always (water depth is positive downwards) - a negative depth implies terrain above sea level." | We agree with this comment and have removed the sentence. |
| 57 | "15/29 and 31. I would stick to either 'variability' or 'rugosity' rather than use both terms." | As summarised in Table 1 there are several measures of terrain variability, including rugosity, VRM etc. It is incorrect to replace the collective term variability with any one of these sub-terms throughout this paragraph. We have, however replaced 'rugosity' with 'terrain variability' in the generic sentence at line 15/31. The following sentences give detail on sub-methods where individual terms need to be retained for clarity. |
| 58 | "16/1 Is meant here comparing depths with a planar surface fitted to the data?" | No, the rugosity measure compares the surface area and planar area. We have revised the wording to make this clearer. |
| 59 | "16/3,4 alternative measures of rugosity that are less affected by gradient." | We did not implement the suggested change here because we felt that it would have changed the meaning of the claim that is made. |
| 60 | "16/6 between them (what exactly?)" | 'them' was changed to 'these attributes' – refers to slope and variability at the start of the sentence. |

| | | |
|---|---|---|
| **61** | "16/11-12 Not in marine geophysics. For example, the subsidence of the seabed with age was first looked at in the 1960s as far as I am aware (references in Parsons and Sclater, 1977)." | See responses to comments #2 and #30, and Section 1.2. Specific geomorphometry is used here in terms of (semi)automatic extraction rather than qualitative description of landforms based on terrain characteristics. |
| **62** | "18/2,3 Is it bathymetry that is influencing the biology or rather a number of other properties vary with water depth (light because of attenuation, waves, etc)?" | Yes it is the bathymetry influencing the biology- line clarified to read "Bathymetry is known to have a first order influence on species distribution, largely because many properties that directly affect benthic habitat vary with depth (e.g. light, temperature)." |
| **63** | "19/16 What is meant here exactly by scale invariance in terms of statistics and morphology?" | This sentence was changed to "This kind of approach has been applied to slope instability offshore Norway, demonstrating the fractal characteristics of submarine mass movement morphology and statistics." |
| **64** | "20/1 I think references need to be cited for these, e.g., Fox et al or Malinverno for the spectral method." | We agree with the reviewer. However, in order to address comment #3 of Reviewer #2, we added a reference to Table 2 in which the reader can find the relevant citations. |
| **65** | "24/32 I wouldn't say they are more affordable, rather the oil and gas industry has invested an enormous amount (with often dubious success) to acquire these datasets. Academic and government researchers have benefitted from access to these data." | This sentence has been changed to "The development of seismic geomorphometry is a natural consequence of increasing computer power, which enables the rapid manipulation, visualisation and interpretation of 3D seismic reflection data, and the enormous investment in this technology by the oil and gas industry, with academics and government researchers benefitting from access to these data." |
| **66** | "26/27 Figure 1 only shows the uptake of the word 'geomorphometry', not practice." | This reference to Figure 1 was removed. |
| **67** | "27/7 If >50 years can be described as infancy, I am an infant!" | This was addressed in responses to comments #2, #30 and #61. |
| **68** | "p 55. Table 2 contains only a selection, so I would make that clear in the table title." | The title of the table was modified accordingly. |
| **69** | "Figure 2 - surely "pre-processing" should occur before interpolating the data onto a grid? Or am I misunderstanding what pre-processing means here (an iterative stage of binning and gathering statistics before filtering the data and forming a final | The term "pre-processing" correspond to the term that is accepted within the geomorphometry literature (cf. Hengl and Reuter, 2009; Bishop et al., 2012), thus making us reluctant to use a different term. We however understand that there may be |

| | | |
|---|---|---|
| | "grid)? A better term might be needed." | confusion for readers that for instance are used to bathymetric data processing. To prevent his, we define and clarified the meaning of "pre-processing" in this context in the introduction of the paper, when referring to this figure (lines 3/26-28). |
| 70 | "Figure 3 caption. The last sentence does not really inform us of what is in the figure. Standard deviation is presumably computed within each cell. It is really not clear what is meant by "hypotheses" - I would not use that kind of term here for what I think is meant a working grid of the data. This kind of language is only going to confuse readers." | This sentence was removed. |
| 71 | "Figure 4. I understood the GEBCO charts to contain information from the hydrographic agencies that is not freely available. The caption needs to say where these data were taken from (geographically) for the discussion to have much meaning. I suspect the remaining data were not only obtained from altimetry measurements." | The GEBCO digital dataset is now freely available, including a software to extract sub-areas of the global dataset, online at www.gebco.net. It is characterized on the website as "the most authoritative, publicly-available bathymetry data set for the world's oceans." |
| 72 | "axis -> direction We don't know the survey direction, ship track etc, so difficult to read the middle panel. Hence, "are mainly caused by the roll motion" - I suspect this is not roll motion exactly, rather a roll motion that is not compensated by the motion sensor used. Striping can arise because of other errors caused by bubbles passing over transducers in bad weather." | The mention of roll was removed. |
| 73 | "Figure 6 is really not very useful, given that the sample set is effectively censored by the limited use of the terms searched." | Figure 6 was removed. |
| 74 | "Suggested additional articles within the field of geomorphometry (only a small sub-set from my personal experience):

 Czarnecki, M.F., Bergin, J.M., 1986. Characteristics of the two-dimensional spectrum of roughness on a seamount. Naval Research Laboratory.

 Goff, J.A., 1991. A global and regional stochastic analysis of near-ridge abyssal hill morphology. J. Geophys. Res. 96, 21,713-721,737.

 Goff, J.A., 1992. Quantative Characterstics of | We thank the reviewer for these references. The following ones were added in the text:

 Blondel and Murton (1997) →lines 7/15 and 7/26

 De Moustier (1988)→line 7/16 and Figure 1

 Czarnecki and Bergin (1986) → line 4/14

 Goff (1991) →line 27/30

 Goff (1992) →lines 4/15 and 27/30 |

Abyssal Hill Morphology along flow line in the Atlantic Ocean. Jour. Geophys. Res. 97, 9183-9202.

Goff, J.A., 2001. Quantitative classification of canyon systems on continental slopes and a possible relationship to slope curvature. Geophys. Res. Lett. 28, 4359-4362.

Herzfeld, U.C., 1993. A method for seafloor classification using directional variograms, demonstrated for data from the western flank of the Mid-Atlantic Ridge. Math. Geol. 25, 901-924.

Malinverno, A., 1990. A quantitative study of the axial topography of the Mid-Atlantic Ridge. J. Geophys. Res. 95, 2645-2660.

Malinverno, A., 1991. Inverse square-root dependence of mid-ocean-ridge flank roughness on spreading rate. Nature 352, 58-60.

Malinverno, A., 1993. Transition between a valley and a high at the axis of mid-ocean ridges. Geology 21, 639-642.

Malinverno, A., Cowie, P.A., 1993. Normal faulting and the topographic roughness of mid-ocean ridge flanks. J. Geophys. Res. 98, 17921-17939.

Malinverno, A., Gilbert, L.E., 1989. A stochastic model for the creation of ocean floor topography at a slow spreading center. J. Geophys. Res. 94, 1665-1675.

Menard, H.W., 1984. Origin of guyots: the Beagle to Seabeam. J. Geophys. Res. 89, 11117-11123.

Mitchell NC (1995) Diffusion transport model for pelagic sediments on the Mid-Atlantic Ridge. J. Geophys. Res. 100(B10):19,991- 920,009

Mitchell NC, Huthnance JM (2007) Comparing the smooth, parabolic shapes of interfluves in continental slopes to predictions of diffusion transport models. Mar. Geol. 236:189-208

Mitchell, N.C., Stretch, R., Oppenheimer, C., Kay, D., Beier, C., 2012. Cone morphologies associated with shallow marine eruptions: east Pico Island, Azores. Bull. Volcanol. 74, 2289-2300.

Goff (2001) →line 4/15

Herzfeld (1993) → line 4/16

Malinverno (1990) → line 4/15

Malinverno and Cowie (1993) → line 27/32

Malinverno and Gilbert (1989) → line 27/31

Parsons and Scalter (1977) →line 4/12

Shaw (1992) → line 14/17

Shaw and Lin (1993) → line 27/32

Shaw and Smith (1987) → line 4/14

Shaw and Smith (1990) → line 4/15

Smith and Wessel (1990) →line 10/4

Wessel and Chandler (2011) → line 4/12

Shaw, P.R., 1992. Ridge segmentation, faulting and crustal thickness in the Atlantic Ocean. Nature 358, 490-493.

Shaw, P.R., Lin, J., 1993. Causes and consequences of variations in faulting style at a Mid-Atlantic Ridge. J. Geophys. Res. 98, 21839-21851.

Shaw, P.R., Smith, D.K., 1987. Statistical methods for describing seafloor topography. Geophys. Res. Lett. 14, 1061-1064.

Shaw, P.R., Smith, D.K., 1990. Robust description of statistically heterogeneous seafloor topogra- phy through its slope distribution. J. Geophys. Res. 95, 8705-8722.

Smith, D.K., 1988. Shape analysis of Pacific seamounts. Earth Planet. Sci. Lett. 90, 457-466.

Smith, D.K., 1996. Comparison of the shapes and sizes of seafloor volcanoes on Earth and "pancake" domes on Venus. J. Volcanol. Geotherm. Res. 73, 47-64.

Smith, D.K., Jordan, T.H., 1988. Seamount statistics in the Pacific Ocean. J. Geophys. Res. 93, 2899-2918.

Other cited references:

Blondel, P., Murton, B.J., 1997. Interpretation of sidescan sonar imagery. John Wiley, Chichester.

de Moustier, C., 1988. State of the art in swath bathymetry survey systems. Internat. Hydr. Rev., Monaco 65, 25-54.

Parsons, B., Sclater, J.G., 1977. An analysis of the variation of ocean floor bathymetry and heat flow with age. J. Geophys. Res. 82, 803-827.

Smith, W.H.F., Wessel, P., 1990. Gridding with continuous curvature splines in tension. Geophysics 55, 293-305.

Strunk, W., White, E.B., 1972. The elements of style, 2nd Ed. MacMillan Publishing, New York.

Wessel P, Chandler MT (2011) The spatial and temporal distribution of marine geophysical surveys.

**Reviewer #2 – Anonymous**

| # | Comment | Revision and Comments |
|---|---------|----------------------|
| 1 | The paper presents an exhaustive list of research dealing with marine geomorphometry. In this sense, it represents a thorough effort and a very complete picture of what has been done in the topic. I put forward the following suggestions which I think may help to improve the manuscript. | We thank the reviewer for recognising the effort that was put in reviewing the existing literature, and for its helpful comments. |
| 2 | I would try to reduce the overall length of the manuscript. As it is now, the paper is not easy to read, it is too long, but especially, sometimes there is a lack of context to the referencing, and all the effort to compile the literature may be partially lost when reporting results of the search. | We tried to reduce the overall length of the manuscript. For instance, the second section was significantly reduced to keep the focus on data characteristics that may impact marine geomorphometry. We also removed the section on spatial scale. However, the other reviewers also asked to add some sections, e.g. in terms of emerging applications and the temporal context of marine geomorphometry. We thus do not think that it resulted in a net length reduction, although we hope to have greatly improved the flow of the manuscript, which should make it easier to read. |
| 3 | I think that the chapters are fine, but I would create tables with the references, avoiding what may be regarded as a long list of literature. Using tables, where the most important papers are reported, section by section, and using the text to discuss the importance of those findings may be more convenient, and a context may be built for the references; thus the listing would be avoided, the text would be more readable, and most of the references would be included in tables. | To address this comment, we expanded Table 2 to re-organize the references used as examples of work performed in marine geomorphometry, and modified Figure 1 to include relevant reviews or discussions related to the different topics. We also tried to reduce the list of literature by removing some references in places where more than two references were given to support a claim. We however preferred not to remove all the references from the text: since each reference usually supports a claim or an argument, removing them from the text would prevent the association of the reference with the appropriate claim. |
| 4 | Having said this, I think the authors should report on their thinking as derived from lessons learned from the literature. As the paper is now, that task is partly derived to the reader, and it is hinted at in the conclusions with the 5 | We included some insights on the lessons learned from past work on marine geomorphometry (particularly the limitations) and how these could be addressed in the future in Section 7. We also welcome specific suggestions based on the experience of the reviewer. |

| | major points highlighted by the authors. | |
|---|---|---|
| 5 | The introduction would benefit to the references to the most recent reviews about high resolution data and earth surface processes, specifically (Passalacqua et al., 2015; Tarolli, 2014). This would also hint to the current limits or merits in marine geomorphometry as related to this wider context of earth surface processes. | A comparison with the more recent work in terrestrial geomorphometry was added in Section 7 to highlight the merits/limits of marine geomorphometry. |
| 6 | As stated above, the manuscript as of now is a bit on the long side. There is a lot of pages focused on data collection and processing. While this is transversally useful, it hinders the part of the paper which is specific to the theme of the review (marine geomorphometry). Currently, chapter 2 appears more as a technical report on the different techniques. The basic technical descriptions could be strongly reduced and referred to relevant sources, focusing more on the merits and limits of the different technologies, underlining the areas where there is still room for progress specifically for the field of marine geomorphometry. | We agree with the reviewer that the technicalities associated with the remote sensing techniques shifted the focus away from the main theme of the review. We thus reduced this section to focus on how each technique dictates data characteristics that are relevant to geomorphometric analyses. We moved the technical parts to an appendix in order to still inform readers that may have limited understanding of the theories behind the different techniques. |
| 7 | All the softwares reported in the article could be nicely organized and referenced in a table, so that a reader can have quick access to their name and location, and eventually reporting also the works that used one software or the other. | We agree that a list of software relevant to each step of geomorphometry may be a useful addition to a manuscript focusing just one or two steps of geomorphometry, where the emphasis is on methods/algorithms. However, we feel that a list of software is not appropriate for this review for the following reasons:

-A list of software would not add anything to the paper unless we made it a comprehensive list of all available software for each of the 5 steps of geomorphometry, not just the few examples of commonly used software we have mentioned to date in certain sections. Such a list would quickly approach 100 software/scripts/tools covering diverse aspects of data processing and analysis.
-Such a list would be overwhelming in the present manuscript which only provides an overview of all 5 steps of geomorphometry. We feel it would detract from the focus the |

| | paper and existing tables.
-Listing of software relevant in 2016 would also quickly give grounds for the paper to become outdated since software are under continuous development both in the commercial and open-source sector.
We appreciate that software reviews have been useful in publications such as Hengl & Reuter (2009) but these were in-depth chapters on software where the functionality could be described in detail. Such detail is beyond the scope of the current review, but would be worth considering for future papers focussing on more technical aspects of just one, or possibly two steps of geomorphometry. |
|---|---|
| **8** | Chapter 3 feels somehow not linked to the review. The chapter about scale could be merged to the different technique described in chapter 2, again highlighting the difference in scale as merit or limit of each technique, for example. | We removed the section on spatial scale and included the relevant parts in Section 2. We however kept Section 3 as the generation of a surface model plays a crucial role in the geomorphometric approach and involves decisions from the user regarding spatial resolution, data quality, interpolation characteristics, sampling, and intended use. We modified this section to link it better to the rest of the review. |
| **9** | Chapter 5 and 6 seems redundant. They could be merged together explaining what technique was used in the different studies. | We kept the description of the techniques separated from the discussion of the application of geomorphometry, partly to enable addressing a comment from Reviewer #3 regarding the addition of a sub-section in the applications section that would not have fit well if Sections 5 and 6 were merged. |
| **10** | I found table 2 very interesting, but surprisingly this is reported only in reference to chapter 6.2, while it could be reorganized grouping the works also according to the aim, not just the technique. | We agree with the reviewer that this table is more useful when used in reference to many applications. We modified the table by classifying the references according to the main theme of the study. |

**Reviewer #3 – Anonymous**

| # | Comment | Revision and Comments |
|---|---|---|
| **1** | This paper aims at providing a timely review of marine geomorphometry. The intention is to raise awareness of the science of geomorphometry in marine environments, to review the existing literature on marine geomorphometry, to highlight differences from terrestrial geomorphometry, and to outline and recommend future research directions | N/A |

| | | |
|---|---|---|
| | within marine geomorphometry. Overall the paper is structured according to the five steps outlined by Pike et al. (2009) in chapter 1 of the seminal book on geomorphometry edited by Hengl and Reuter. | |
| 2 | As I was invited to review the paper after two other reviews were already available, I have tried not to repeat the comments and suggestions of the two other referees which both agree with. I instead try to focus on other issues of the paper. Overall I very much agree with the authors that there is a need for a review on marine geomorphometry. It is a discipline which is blooming immensely in these years, and this will for sure be the case in several years to come. | We appreciate that the reviewer finds this review timely. |
| 3 | RE Structure: The five steps in geomorphometry outlined by Pike et al. (2009) are highly pedagogical in relation to teaching and learning geomorphometry as a discipline. However, the choice of structuring the paper according to these five steps and combining this with the aim of attempting an exhaustive review is very challenging, and more or less a Sisyphus task, as each of the five steps would be worthy of its own review. The result is that the review/analysis of each of the five steps tends to become surficial and to some extent rather being a listing of earlier studies. Consider an alternative structure with a clear alignment between the aim and objectives of the paper and the findings and recommendations based on the analysis of the authors. In this regard the aims and objectives could be even more precisely formulated, also in order to highlight the focus of the review. Referee #2 provides many good suggestions of how to alternatively structure the paper. | We agree with the reviewer that each section would be worthy of its own review. Our objective was not to write a review of each of these steps, but to review how they integrate together to form a typical marine geomorphometry workflow.

Reviewer #2 thought that "the chapters are fine". We have discussed the matter and decided to keep the current structure of the manuscript based on Pike et al. (2009). It is often observed in the community of end-users that there is a lack of understanding of the fact that each of the five steps has implications for the derivation of terrain attributes or extraction of terrain features, and consequently implications for the final application. By structuring the review following these steps, we hope that end-users can realize the role of each step in determining their quantitative terrain characteristics, something that we emphasized on in lines 24-26 of page 27.

To address this comment, we made some sections, e.g. section 2 on the remote sensing techniques and section 3, shorter and shifted their focus towards data characteristics and how they impact the final application. We believe that this make the manuscript flow better. We also made it clearer in the introduction why we picked that structure, and why it is important to consider each of these steps when using geomorphometry. In doing so, we hope that it made the review less surficial and |

<table>
<tr><td></td><td></td><td>more focused on the importance of each step for the final application. We also added specifications throughout the manuscript that it is meant to be an overview and not a comprehensive review or each steps. For instance: "Clearly there is room in the literature for more detailed reviews of each of these five steps and relating to many of the sub-disciplines, however we hope that this review will serve as a solid foundation for further, more detailed reviews on these sub-topics." (lines 3-5 of page 29)</td></tr>
<tr><td>4</td><td>RE Technologies: The section on sampling technologies (section 2) could be excluded from the review. Despite the dedication of five pages to the section it remains surficial, as it attempts to encompass a very broad range of technologies, including technologies like SBES which will not play a key role in the future. The data quality is fundamental for the DEM quality, and consequently for all DEM derivatives, and the final interpretations. However, this could be described and explained more generically by focusing on the general characteristics and properties of point data, which in essence is integrated areal information, as all points are related to a footprint. I very much agree with the authors that one of the key dangers in the future application of DEMs and geomorphometry in planning and management is that the planers, managers and decision makers are not aware of the properties of the data foundation of the DEMs. Hence, consider highlighting data properties and not different technologies.</td><td>We did not fully exclude this section from the review. As mentioned in the responses to other comments, we found important to individually address each of the five steps that impact geomorphometric applications. In addition, Reviewer #1 had many comments (#31 to 48) that suggested specifications and extension of this section, thus challenging the idea of excluding this section.

We however agree with the reviewer that this section was too technical and too detailed, so we shifted the focus away from the technologies and towards the data. Regarding SBES, we indicated that its use is likely to dwindle, but we also mention a recent study that demonstrated that previously collected SBES data can be combined together and offer much potential for applications.

We highlighted the reviewer's last thought on awareness of data foundation in lines 24-26 of page 27.</td></tr>
<tr><td>5</td><td>RE Terminology: From a reader's point of view, the impression arises that the authors stress the importance of the differences between terrestrial and marine geomorphometry, as if the identification of differences would make the field of marine geomorphometry more relevant. This even leads to the suggestion of different terminologies for DEMs. In my opinion this is contra productive and merely confusing. The acronyms DEM, DSM and DTM are generic and sufficient for all environments. A more uniting approach with suggestions for a joint terminology and vocabulary across environments</td><td>We agree with this comment and consequently tried to highlight the similarities between terrestrial and marine geomorphometry rather than solely the differences. We added some elements on this, for instance in Section 7.

Regarding terminology, the reviewer seems to suggest that we came up with the different terms for DTMs. Since this is review, the different terms that are discussed in the manuscript are those found in the literature. The difference between DEM and DBM was clearly defined in Section 3,</td></tr>
</table>

| | | |
|---|---|---|
| | would be much more meriting. One of the major potentials of geomorphometry, as for geomorphology, is that it interacts with many disciplines. Within the geomorphometry community we should aim at aligning our terminology in order to foster and ease the coupling with other disciplines. Therefore, consider highlighting both similarities and differences between terrestrial and marine geomorphometry, and aim to unite the two disciplines wherever it is possible. | thus avoiding confusion. We agree that the generic terms DSM and DTM are sufficient for all environments (although they mean different things), and as suggested by the reviewer, we clarified this issue and added to our recommendations the adoption of a joint terminology across environments (cf. Section 7.2). |
| 6 | RE Domain: The exact spatial domain of the review is unclear. There is a strong bias towards deep water environments with practically no review of the shallow water coastal environment. Nevertheless, the authors seem to use the coastline as the perimeter of their domain. Many studies, especially within the last decade, have used high resolution DEMs in shallow water coastal environments to quantify sediment transport, sediment transport pathways, morphology and morphodynamics. These studies are practically absent in the present review. Many of these studies are available in relation to the line of international conferences of MARID, RCEM and ICS, and in the journals JGR Earth Surface, Geomorphology, ESPL and more recently ESurf (I will not mention any specific studies here, as the list is long). Moreover, the authors mention the bridging of terrestrial and marine environments as one of the key future challenges (and actually a paper in the special issue is addressing this), but this without having reviewed the shallow water environments in the first place. Hence, the present analysis is simply to surficial. This was also highlighted by referee #1 although not in relation to shallow water. Consider defining the exact domain of the review, and then either fully include or exclude shallow water coastal environments. Specifically it even leads to erroneous information, e.g. when the authors discuss tidal corrections which are rarely used in high-precision and high-resolution shallow water environments where high-precision positioning is available and applied. | Our spatial domain does extend to the coastline. Our reference list already included many articles focusing on the coastal waters and the continental shelf/shallow waters. To make this clearer, we added the spatial domain to the list of articles cited in Table 2.

We also do not feel the need to specifically review the shallow water environment as suggested by the reviewer since we do not specifically review the deep water environment. We define our spatial domain as the marine environment, which encompasses shallow and deep waters. We believe, as demonstrated in Table 2, that we cover both depth ranges in a reasonable fashion that do not justify excluding one or the other.

Bridging terrestrial and marine environments is a challenge in terms of integrating and fusing datasets from multiple sources for regional to global-scale analysis, rather than actually bridging the environments at a local-scale with LiDAR. We agree that this may not have been clear, and thus modifications were made to Sect. 6.4.3 to review the current literature on LiDAR, and made more explicit that the challenge for the future is in combining different datasets for geomorphometry at a broader scale.

We also note that we did not provide erroneous information regarding tidal corrections. Acoustic surveys do require tidal corrections, and references are made to published peer-reviewed literature to support this claim. The reviewer also mentioned the paper in this special issue that |

| | | |
|---|---|---|
| | | addresses the littoral gap using LiDAR; a quick look at it confirmed that the authors of this paper did account for tides (-1m DVR90), using a tidal gauge 20km away from the study site (which confirms a claim made in our paper), and that they needed to correct for the "continually changing water level in the study area due to tides". This highlight the fact that many applications still do not have access to the latest positioning technologies mentioned by the reviewer and still require proper calibration of their system. |
| 7 | RE Applications: A range of applications is listed. However, the descriptions of the applications are strongly biased towards habitat mapping, and to some extent also hydrography in relation to safety of navigation. The enormous potential of geomorphometry is that it has a vast amount of applications, and some not yet realised. It would suit the review if the analysis of the authors would lead to suggestions of new areas where geomorphometry has not yet been introduced and tested or highlight areas where present applications could be further developed. | The bias towards habitat mapping and geomorphology is in line with the fact that these are the two most common applications of marine geomorphometry, as illustrated in one of the figures that was removed from the initial submission. However, we agree with the reviewer that it would suit the review to suggest new areas of application. We thus extended Section 6 to suggest new areas where marine geomorphometry may be applied with success in the near future. We also welcome specific examples that the reviewer would like to see included. |
| 8 | I have not included any specific comments to the separate sections and also no comments for technical corrections. Due to the somehow fundamental character of the general comments from all three referees, it seems more relevant to initially restructure and refocus the paper, before more detailed corrections are suggested. | N/A. |
| 9 | I sincerely hope that the authors will take up the challenge to revise the paper. Marine geomorphometry is a blooming field and we need an overview within the community, and we need this paper to ease communication with other disciplines that could benefit from marine geomorphometry. | We thank the reviewer for recognising that this review paper is timely. |

**A review of marine geomorphometry, the quantitative study of the seafloor**

[revised manuscript text omitted]

 Likewise, the oceans  play a fundamental role in the Earth system at multiple scales. ~~An important part of ocean-
20   related work addresses the physical, chemical and biological patterns and processes of the water surface and, to a lesser extent, the water column. . By contrast, seafloor research has been less spatially extensive, except for research on global, broad-scale geomorphology and processes (e.g. Ma et al., 1998).questions~~subjects (Smith, 2004). For example, seafloor topography, or bathymetry, influences surface currents (Gille et al., 2004), near-bottom currents (White et al., 2007), and ocean mixing rates (Kunze and Llewellyn Smith, 2004). Lack of
25   knowledge on factors influenced by bathymetry can affect the efficacy of model predictions, for example models of marine species distributions (McArthur et al., 2010), climate (Jayne et al., 2004), or the paths of floating objects like marine debris (Smith and Marks, 2014).

It is commonly stated that 90% of the global ocean is unexplored (e.g. Gjerde, 2006) and that more is known about the surface of Earth's Moon , , Mars , Mercury

(e.g. Zuber et al., 2012) or Venus (e.g. Ford and Pettengill, 1992) than about the ocean floor (Sandwell et al., 2002; Smith, 2004; Smith and Marks, 2014). However, such statements mean little without further specification or elaboration on their real meaning. The facts are that the in relation to objectives, data types and spatial resolution. The entire ocean floor has been mapped to a resolution of a few kilometres using satellites, which generatedhas created an estimationestimated surface of the underwater landscape and revealed features of the Earth's crust beneath seafloor sedimentsglobal bathymetry (Smith and Sandwell, 1994; Smith, 1998). However, these coarse-resolution data are often inadequate for many scientific, economic, public safety and management purposes. Applications such as tsunami hazard assessment, submarine cable and pipeline route planning, resource exploration, habitat mapping, territorial claims, navigation, and ocean circulation and climate studies, and navigation were all identified as requiringrequire more reliable, fine-scale bathymetric data (i.e. finer than 5 km) (Sandwell et al. 2002). It is therefore imperative to extract the most useful information that can be retrieved from the seafloor data available at resolutions finer than 5 km.

[revised manuscript text omitted]
. The aims of this contribution are therefore to raise awareness of this relatively recent field and to lay the basis of marine geomorphometry practices by reviewing the relevant literature to date. As illustrated in Fig. 2, this manuscript addresses the five main steps of geomorphometry identified by Pike et al. (2009) with a focus on how these steps are relevant to marine geomorphometry and different from traditional, terrestrial geomorphometry. The five steps are: sample the surface (Sect. 2), generate a surface model from the sampled heights (Sect. 3), preprocessing, i.e. correct for errors and artefacts in the surface model (Sect. 4), surface analysis, i.e. derive terrain attributes and terrain features (or objects) (Sect. 5), and apply the terrain attributes and features to a specific problem (Sect. 6). We conclude with recommendations and reflections on the future of marine geomorphometry.~~

**2 Sampling the depth of the seafloor**

For centuries, the lead line was the main instrument used to determine the depth of the seafloor, until remote sensing technologies revolutionized the way we could measure bathymetry. This section introduces  four types of remote sensing technology that are currently used to collect depth information and found in the geomorphometry literature: satellite radar altimetry, optical remote sensing, acoustic remote sensing, and bathymetric LiDAR . From an application perspective, the survey methodology

or methodologies dictate the spatial scale (i.e. resolution and extent) of the final surface model. First, the fundamental technical limitations of the remote sensing technique that is used to collect bathymetric data will define the scale (resolution) of the surface model (Kenny et al., 2003; Van Rein et al., 2009). both collect depth information in an indirect wayFor instance, radar altimetry data limits models constructed with them to coarse, usually kilometre-scale resolution, while acoustic remote sensingother methods can achieve up to centimetre-scale resolution models. Second, by defining the distance between the platform and bathymetric LiDAR measure the seafloor directly. Radar the target, the characteristics of the latter may also influence the scale (extent) of the final model; a remotely sensed image collected from a satellite will usually have a coarser resolution and cover a larger area than an image collected from an aircraft or an unmanned aerial vehicle. While radar altimetry and acoustic remote sensing are limited to deeper waters whileand optical remotely sensed and bathymetric LiDAR data are limited to shallower waters, although there is some degree of overlap between the depths in which the various methods can be applied. A global scaleIn terms of effort, systematic bathymetric survey could be performed with satellite-based methods within a few years at a global scale (Sandwell et al., 2002), compared to the estimated 600 years (Carron et al. 2001) that it would take using acoustic remote sensing technologies. The different techniques are discussed in the perspective of using the information they collect to generate Digital Terrain Models (DTM)DTMs and perform geomorphometric analyses. DTMs using bathymetric data are hereafter referred to as Digital Bathymetric Models (DBM) to distinguish them from Digital Elevation Models (DEM), a term usually reserved for terrestrial elevation data though sometimes including bathymetric data for global datasets.. Other techniques can be used to measure depth but are less common in the literature. For instance, (e.g. ground-penetrating radar can be used to detect the thickness of the water by measuring the difference between the radar echoes from the air-water and water-seafloor interfaces (, Feurer et al., 2008). More details on the underlying theories of these four techniques can be found in Appendix 1.

**2.1 Satellite radar altimetry**

In the 1970s, satellite-based radar altimeters were developed (cf. Appendix 1) as a method to study the oceans on a global scale (Douglas et al., 1987), which was a significant improvement over the extent covered by very narrow ship tracks. Radar altimeters emit microwaves that bounce on the sea surface and return to the receiver, giving the altitude of the satellite over the sea surface. The topography of the surface can then be deduced and used to derive ocean circulation patterns (Fu, 1983) or define geoid models (e.g. Fernandes et al., 2000). The geoid represents the gravitational equipotential surface of the Earth relative to the mean sea level; gravity varies in space, and the anomalies in its distribution were found to be correlated with bathymetry (McKenzie and Bowin, 1976; Watts 1979). Despite initial reports stating that it was impossible to derive reliable bathymetry from satellite altimeter (Keating et al., 1984; Watts and Ribe, 1984), Dixon et al. (1983) were the first to demonstrate its feasibility using real data. Several algorithms and methods to estimate and predict bathymetry from the gravitational field have since been developed (reviewed in Calmant and Beaudry, 1996 and Sandwell and Smith, 2001).

However, it remains a complex process (Calmant and Beaudry, 1996) that still requires acoustic data for calibration (Smith and Sandwell, 1997).

TheConsequently, the applications of altimetry-derived bathymetric data are limited to the study of broad-scale patterns, processes and features as they only provide low resolution estimates of the bathymetry (Goff et al., 2004); ocean waves create a lot of noise that prevents the collection of fine-resolution data (Smith, 1998), and rough seafloor geology and thick sediments affect data accuracy (Smith and Sandwell, 1994). Technological constraints and satellite orbits also prevent data collection close to the poles and the coastline (Sandwell et al., 2002). Some authors identified weaknesses in the method and warned that predicted depths from altimetry may not be reliable and should not be used for geodynamics studies (Smith, 1993), navigation, or hazard identification (Smith and Sandwell, 1994). The main advantages of altimetry-derived bathymetry are speed of collection and uniformity of coverage (Mackenzie, 1997).

Two main altimetry-derived datasets are currently used in applications of marine geomorphometry: the General Bathymetric Chart of the Oceans (GEBCO, 2014) and the Shuttle Radar Topography Mapping 30-arc second database (SRTM30_PLUS, Becker et al., 2009). They are both free datasets that combine together elevation and bathymetric data. The bathymetric parts were created by filling the gaps between publicly available datasets from different sources with radar altimetry (Smith and Sandwell, 1994, 1997; Becker et al., 2009). These datasets have been used for instance in habitat mapping and predictive modelling (e.g. Davies et al., 2008; Knudby et al., 2013), conservation (e.g. Ross and Howell, 2013), search and rescue operations (Smith and Marks, 2014), and geomorphology (e.g. Harris et al., 2014). Many works have found these datasets to be too coarse for their purposes (e.g. Davies et al., 2008; McNutt, 2014). For instance, Vierod et al. (2014) stated: "At present, the availability of bathymetric data at a resolution sufficient to inform reliable terrain attribute predictors is a major limitation to the ability of deep-sea species distribution models to make accurate predictions of the distributions of benthic organisms." For many applications, quality can also be just as important as resolution (e.f.cf. Sect. 4 and Sect. 6).

**2.2 Optical remote sensing**

Of the four remote sensing methods presented in this section, optical passive remote sensing is the least common in the marine geomorphometry literature. However, it presents a cheaper alternative to LiDAR data for collecting depth information in very shallow coastal areas (Su et al., 2014), as satellites can cover large areas in less time (Lafon et al., 2002; Wang and Philpot, 2007). Two main passiveoptical remote sensing techniquesgroups of methods are presentedusually used to estimate bathymetry from optical remote sensing: one based on the interactions of electromagnetic radiations with water and one based on principles from stereoscopy. (see Appendix 1).

Methods based on electromagnetic radiations are limited to shallow waters because of light attenuation within the water column (Jawak et al., 2015). In theory, based on light penetration in coastal waters, depths down to 50 m could be retrieved (Speight and Henderson, 2010). However, the practical limit varies with local sea conditions. A maximum of 30 m deep is

usually achieved when local conditions are exceptionally good (Collet et al., 2000; Jawak et al., 2015), and most often a depth of 15 m is reported as being the performance limit of optical remote sensing for bathymetry retrieval (e.g. Stumpf et al., The ability to derive depth estimates from imagery comes from the optical Beer Lambert law of light absorbance, which describes light absorption as it goes through a transparent medium (Serway and Beichner, 1983). In water, light gets

5 absorbed exponentially as depth increases (Lyzenga, 1978). The Beer Lambert law allows the mathematical derivation of depth estimates from the brightness values of pixels in an image, when the absorption characteristics of an area are known (Mobley et al., 2005; Carbonneau et al., 2006). Since the absorption rate of an area is dependent on water turbidity and the characteristics of the incoming energy (e.g. the intensity, angle and wavelength of sunlight), calibration with ground truth data verifying depth colour relationships (i.e. local light absorption characteristics) is a key step in the application of this

10 method. However, calibration is made difficult by temporal variations in the illumination characteristics of an area (Carbonneau et al., 2006); the calibration data would ideally need to be collected at the same time as the remotely sensed data to ensure identical environmental conditions. Since reliable calibration data are particularly challenging to obtain in marine waters (Lafon et al., 2002; Dekker et al., 2011), some methods have been proposed to estimate bathymetry without ground truth data (e.g. Fonstad and Marcus, 2005), however these are not yet widely adopted (Feurer et al., 2008). Rather

15 than using the level of absorbed energy to derive bathymetry from imagery, some authors, e.g. Maritorena et al. (1994), have used bottom reflectance, which is the level of reflected energy.

Photogrammetry applied to pairs of stereo images can also be used to build DBMs in a similar method to the technique applied on land.2003). Optical methods Although possible (e.g. Stojic et al., 1998), through water photogrammetry is challenging due to the need to correct for the air-water interface (Feurer et al., 2008). Underwater photogrammetry (i.e.

20 active remote sensing) has, however, been successfully applied at a fine scale to reconstruct the digital terrain (e.g. Johnson-Roberson et al., 2010; Kwasnitschka et al., 2013). The work by Friedman et al. (2012) is noteworthy as they derived multi-scale measures of rugosity, slope and aspect from underwater stereo image reconstructions.

All types of imagery have been used to derive marine bathymetry: hyperspectral (e.g. Mobley et al., 2005; Ma et al., 2014), multispectral (e.g. Lyzenga et al., 2006; Pacheco et al., 2015), broadband colour (e.g. Westaway et al., 2003) and grayscale

25 images (e.g. Winterbottom and Gilvear, 1997). Multispectral images enable refined depth estimates by extracting information on the bottom types from non-visible spectral bands and linking the bottom types with depth (Winterbottom and Gilvear, 1997). The red band of the electromagnetic spectrum is particularly successful in detecting depth variations (Legleiter et al., 2004; Carbonneau et al., 2006).

All methods from optical passive remote sensing are limited to shallow waters and are sensitive to errors caused by waves,

[revised manuscript text omitted]

**2.4 Bathymetric LiDAR**

Bathymetric LiDAR is an adaptation of the more traditional airborne topographic LiDAR (Irish and Lillycrop, 1999; Guenther et al., 2002, see Appendix 1) and has become increasingly common in the literature in the last two decades (Brock and Purkis, 2009). The main difference between the two types of systems is the wavelengths used; the laser from Recently, they have been combined into topo-bathymetric LiDAR uses a wavelength in the green spectrum, compared to the red/infrared wavelength of the topographic LiDAR. Recent systems, sometimes called topo-bathy LiDAR, which are multispectral LiDAR and combinesystems that enable data collection both types of laser, which enables the surveying of bothabove land and water in one flight; when flying over the water, thea green laser – characteristic of bathymetric LiDAR – penetrates the sea surface and collects information on the water column and the seafloor, while the red/infrared laser – characteristic of topographic LiDAR – collects information on the sea surface. LiDAR can also collect intensity values that, like acoustic backscatter, provide information on the characteristics of the seafloor (Costa et al., 2009; Kashani et al., 2015).

Bathymetric LiDAR is the only technique that can collect high-resolution data in very shallow waters, which makes it especially relevant for coastal applications requiring fine-scale data (<1 m resolution) (Brock and Purkis, 2009). The efficiency of bathymetric LiDAR systems is greatly limited by turbidity, wave action, depth (up to 50-70 m in veryexceptionally good conditions), steep slopes, and rocky substrate (Costa et al., 2009; Chust et al., 2010; Jalali et al., 2015). Current geomorphometric applications on bathymetric LiDAR data are mainly related to the exploration of coastal ecosystems (e.g. Wedding et al., 2008; Zavalas et al., 2014) and geomorphology (e.g. Arifin and Kennedy, 2011; Kennedy et al., 2014), but are likely to extend to other applications such as marine archaeology and natural hazards assessment (e.g. Solsten and Aitken, 2006). In 2015, LiDAR data represented 4.5% of the coastal data collected for the Continually Updated Shoreline Product (CUSP) compiled by the National Oceanic and Atmospheric Administration (NOAA) and the National Geodetic Survey (NGS) of the United States (Graham et al., 2015).

**3 Generating a surface model from sampled depths**

~~By nature, geomorphometric analyses necessitate spatially continuous data, but not all remote sensing techniques used to collect depth samples create continuous surfaces. For instance, bathymetric LiDAR and MBES collect point data that need to be cleaned and then interpolated in order to create a full coverage, and SBES collect data in narrow lines that sometimes need to be interpolated to fill in between the survey lines. This section describes the different interpolators that are available for DBM generation and the question of spatial scale of DBMs.We do, however,in Sect. 3.2 where~~when we present a method where uncertainty algorithms are used to aid data cleaning, and where the interpolation of data is intrinsically linked to the calculation of uncertainty of the bathymetric surface.

**3.1 Interpolation**

[revised manuscript text omitted]

Harris and Baker (2012b) provide a summary of surrogate variables used for habitat mapping studies in the volume 'Seafloor Geomorphology as Benthic Habitat: GeoHAB Atlas of Seafloor Geomorphic Features and Benthic Habitats' including many terrain attributes that have been applied across a multitude of approaches to habitat mapping worldwide. The issue of surrogacy is also discussed in this volume as well as by Lecours et al. (2015b) and McArthur et al. (2010). The case studies presented in the GeoHAB Atlas, and other published studies, vary in the degree to which they have established the ecological relevance of the terrain attributes and/or feature classifications used. For geomorphological variables to really be useful predictors of seafloor habitat, the relationship between habitat and specific variables first needs to be established. Apart from depth, which all of the geomorphological variables are derived from, different shapes or attributes of the seafloor will be relevant to different species at different scales over different bathymetric and biogeographic zones. Bathymetry is known to be a first order influence on species distribution. There are a number of recent papers describing 
[revised manuscript text omitted]

Fernandes, M. J., Bastos, L., and Catalão, J.: The role of multi mission ERS altimetry in the determination of the marine geoid in the Azores. Mar. Geod., 23, 1-16, 2000.

Feurer, D., Bailly, J.-S., Puech, C., Le Coarer, Y., and Viau, A. A.: Very-high-resolution mapping of river-immersed topography by remote sensing. Prog. Phys. Geog., 32, 403-419, 2008.

Fisher, P. F., and Tate, N. J.: Causes and consequences of error in digital elevation models. Prog. Phys. Geog., 30, 467-489, 2006.

Florinsky, I. V.: Accuracy of local topographic variables derived from digital elevation models. Int. J. Geogr. Inf. Sci., 12, 47-61, 1998.

Florinsky, I. V.: Digital terrain analysis in soil science and geology. Elsevier/Academic Press, The Netherlands, 379 p., 2012.

Fonstad, M. A., and Marcus, W. A.: Remote sensing of stream depths with hydraulically assisted bathymetry (HAB) models. Geomorphology, 72, 320-339, 2005.

Ford, P. G., and Pettengill, G. H.: Venus topography and kilometre-scale slopes. J. Geophys, Res., 97, 13103-13114, 1992.

Fosså, J. H., Lindberg, B., Christensen, O., Lundälv, T., Svellingen, I., Mortensen, P. B.., and Alvsvag, J.: Mapping of *Lophelia* reefs in Norway: experiences and survey methods, in: Cold-water corals and ecosytems, Springer-Verlag, Berlin, Germany, 359-391, 2005.

Foster, G., Walker, B. K., and Riegl, B. M.: Interpretation of single-beam acoustic backscatter using LiDAR-derived topographic complexity and benthic habitat classifications in a coral reef environment. J. Coastal Res., 53, 16-26, 2009.

Fox, C. G.: Objective classification of oceanic ridge-crest terrains using two-dimensional spectral models of bathymetry: Application to the Juan de Fuca Ridge. Mar. Geophys. Res., 18, 707-728, 1996.

Fox, C. G., and Hayes, D. E.: Quantitative methods for analyzing the roughness of the seafloor. Rev. Geophys., 23, 1-48, 1985.

Friedman, A., Pizarro, O., Williams, S. B., and Johnson-Roberson, M.: Multi-scale measures of rugosity, slope and aspect from benthic stereo image reconstructions. PLOS One, 7, e50440, 2012.

Fu, L. L.: Recent progress in the application of satellite altimetry to observing the mesoscale variability and the general circulation of the oceans. Rev. Geophys. Space Phys., 21, 1657-1666, 1983.

[revised manuscript text omitted]

Fig.

Fig. 6: Indicative workflow showing the use of terrain attributes in predictive habitat mapping. Generally following some pre-selection of variables the observed habitat points (response variable) are combined with full coverage predictor variables selected from bathymetry, terrain attributes and other environmental variables as available to form the input to a habitat model which will be used to predict a full-coverage habitat map. The choice of habitat model will depend on the study in question but is typically either a statistical (e.g. GLM) or machine-learning based model (e.g. Random Forest). Observed habitat points are classified from visual or physical samples of the seabed. Terrain attributes are typically multi-scale and may include general and/or specific geomorphometry. Other environmental variables may include, for example, oceanographic data (temperature, salinity, current speed etc geological data (e.g. grain size). A similar workflow applies to modelling of single species or communities, where the output will be a continuous map indicating the probability of occurrence within the study area, rather than a categorical map as shown here.

Fig. 7: Example of the use of marine geomorphometry to semi-automatically map the components of mega-scale submarine landslide offshore Norway (adapted from Micallef et al. (2009)).  Figures a-c show the trough depth, ridge length and ridge spacing extracted from a multibeam echosounder map of the north-eastern Storegga Slide using ridge characterisation techniques (Micallef et al., 2007b).  Figure d is a classification map generated by using these ridge characteristic maps as input layers in an unsupervised clustering algorithm (ISODATA). Figure e is an interpretative map of the range of spreading events based on figures a-d. Other mass movements and geological processes and structures have been interpreted using geomorphometric mapping (Micallef et al., 2007a).

[Figure]

[Figure]

| | Sections | Relevant Reviews/Discussions |
|---|---|---|
| Sampling the Depth of the Seafloor | 2.1 Satellite Radar Altimetry
2.2 Passive Remote Sensing
2.3 Acoustic Remote Sensing
2.4 Bathymetric LiDAR | → Calmant and Beaudry (1996), Sandwell and Smith (2001)
→ Klemas (2011a, 2013), Jawak et al. (2015)
→ De Moustier (1988), Dowling and Sabra (2015)
→ Guenther et al. (2002), Klemas (2011b), Jawak et al. (2015) |
| Generating a Digital Bathymetric Model | 3. Interpolation, Spatial Scale | → Calder (2003), Tang et al. (2006), Hengl and Evans (2009),
Li and Heap (2011), Zhang et al. (2014) |
| Preprocessing | 4. Correcting Errors/Artefacts | → Debese (2001), Hughes-Clarke (1998, 2003a),
Oksanen and Sarjakoski (2005), Fisher and Tate (2006),
Reuter et al. (2009), Temme et al. (2009) |
| Analysing the Digital Bathymetric Model | 5.1 General Geomorphometry
5.2 Specific Geomorphometry | → Evans (1972, 2003), Bishop et al. (2012)
→ Jarvis and Clifford (2003), Minár and Evans (2008),
Drăguţ and Blaschke (2006), Drăguţ and Eisank (2011) |
| Applications | 6.1 Marine Habitat Mapping
6.2 Marine Geomorphology
6.3 Hydrodynamics
6.4 Other Applications | → Wilson et al. (2007), McArthur et al. (2010), Dolan (2012)
→ Micallef et al. (2008), Harris et al. (2014)
→ Sandwell et al. (2002), Gille et al. (2004), Normile (2014)
→ Wood (2003), Passaro et al. (2013), Schimel et al. (2015) |

**Figure 1**

[Figure]

[Figure]

**Figure 2**

[Figure]

[Figure]

**Figure 3**

[Figure]

[Figure]

**Figure 4**

[Figure]

[Figure]

**Figure 5**

[Figure]

**Figure 6**

[Figure]

[Figure]

[Figure]

**Figure 7**

[Figure]

**Figure 8**

---

## Author Response (AR2)

**Change report**

Manuscript # hess-2016-73

We thank the editor for her positive feedback. We made the six suggested changes and corrected some more

5    typos that were found upon rereading the manuscript. The proposed reference was also added in the coastal geomorphometry section.

[revised manuscript text omitted]

**Sampling the Depth of the Seafloor**

2.1 Satellite Radar Altimetry → Calmant and Beaudry (1996), Sandwell and Smith (2001)
2.2 Passive Remote Sensing → Klemas (2011a, 2013), Jawak et al. (2015)
2.3 Acoustic Remote Sensing → De Moustier (1988), Dowling and Sabra (2015)
2.4 Bathymetric LiDAR → Guenther et al. (2002), Klemas (2011b), Jawak et al. (2015)

**Generating a Digital Bathymetric Model**

3. Interpolation, Spatial Scale → Calder (2003), Tang et al. (2006), Hengl and Evans (2009), Li and Heap (2011), Zhang et al. (2014)

**Preprocessing**

4. Correcting Errors/Artefacts → Debese (2001), Hughes-Clarke (1998, 2003a), Oksanen and Sarjakoski (2005), Fisher and Tate (2006), Reuter et al. (2009), Temme et al. (2009)

**Analysing the Digital Bathymetric Model**

5.1 General Geomorphometry → Evans (1972, 2003), Bishop et al. (2012)
5.2 Specific Geomorphometry → Jarvis and Clifford (2003), Minár and Evans (2008), Drăguţ and Blaschke (2006), Drăguţ and Eisank (2011)

**Applications**

6.1 Marine Habitat Mapping → Wilson et al. (2007), McArthur et al. (2010), Dolan (2012)
6.2 Marine Geomorphology → Micallef et al. (2008), Harris et al. (2014)
6.3 Hydrodynamics → Sandwell et al. (2002), Gille et al. (2004), Normile (2014)
6.4 Other Applications → Wood (2003), Passaro et al. (2013), Schimel et al. (2015)

**Figure 1**

[Figure]

**Figure 2**

[Figure]

**Figure 3**

|            | Bathymetry | Slope | Curvature | Variability |
| :--------: | :--------: | :---: | :-------: | :---------: |

[Figure]

**Figure 4**

[Figure]

**Figure 5**

[Figure]

**Figure 6**

[Figure]

**Figure 7**